# Variable particle size distributions reduce the sensitivity of global export flux to climate change

Shirley W. Leung[1], Thomas Weber[1,2], Jacob A. Cram[1,3], Curtis Deutsch[1]

[1]School of Oceanography, University of Washington, Seattle, 98195, US

[2]School of Arts and Sciences, University of Rochester, Rochester, 14627, US

[3]Horn Point Laboratory, University of Maryland Center for Environmental Science, Cambridge, 21613, US

*Correspondence to*: Shirley W. Leung (shirlleu@uw.edu)

**Abstract.** Recent Earth System Models predict a 10-20% decrease in particulate organic carbon export from the surface ocean by the end of the 21st century due to global climate change. This decline is mainly caused by increased stratification of the upper ocean, resulting in reduced shallow subsurface nutrient concentrations and a slower supply of nutrients to the surface euphotic zone in low latitudes. These predictions, however, do not typically account for associated changes in remineralization depths driven by sinking particle size. Here we combine satellite-derived export and particle size maps with a simple 3-D global biogeochemical model that resolves dynamic particle size distributions to investigate how shifts in particle size may buffer or amplify predicted changes in surface nutrient supply and therefore export production. We show that higher export rates are empirically correlated with larger sinking particles and presumably larger phytoplankton, particularly in tropical and subtropical regions. Incorporating these empirical relationships into our global model shows that as circulation slows, a decrease in export is associated with a shift toward smaller particles, which sink more slowly and are thus remineralized shallower. This shift towards shallower remineralization in turn leads to greater recycling of nutrients in the upper water column and thus faster nutrient recirculation into the euphotic zone. The end result is a boost in productivity and export that counteracts the initial circulation-driven decreases. This negative feedback mechanism (termed the particle size-remineralization feedback) slows export decline over the next century by ~14% globally (from -0.29 GtC/year to -0.25 GtC/year) and by ~20% in the tropical and subtropical oceans, where export decreases are currently predicted to be greatest. Our findings suggest that to more accurately predict changes in biological pump strength under a warming climate, Earth System Models should include dynamic particle size-dependent remineralization depths.

## 1 Introduction

A key mechanism that controls the partitioning of carbon dioxide ($CO_2$) between the atmosphere and ocean is the biological pump, in which $CO_2$ is fixed into phytoplankton organic matter via photosynthesis, and then exported from the surface to the deep ocean as sinking particles (e.g., Ducklow et al., 2001). Decomposition of this particulate organic carbon (POC) in the ocean interior maintains a reservoir of respired $CO_2$ that is sequestered out of contact with the atmosphere, thus exerting an important control on long-term atmospheric $CO_2$ concentrations and global

climate (e.g., Martínez-García et al., 2014; Passow & Carlson, 2012; Sarmiento & Siegenthaler, 1992). Carbon
exported out of the surface euphotic zone also fuels the metabolism of organisms in the mesopelagic zone, sustaining
economically and socially important fisheries, as well as ecologically important zooplankton and micronekton
communities (e.g., Boyd et al., 2019; Friedland et al., 2012). POC export is also an important driver of dissolved
oxygen concentrations in the water column. Where sinking POC fluxes are particularly high and supply of oxygen via
physical transport is low, enhanced bacterial degradation of particles can deplete available oxygen and create hypoxic
or even suboxic conditions in which many organisms cannot survive (e.g., Deutsch et al., 2015; Deutsch et al., 2020;
Hofmann and Schellnhuber, 2009). Given the critical role of POC export in driving ocean carbon sequestration, the
global climate system, fisheries productivity, and dissolved oxygen availability, there is a growing need to better
understand how export will respond to future climate warming.
Recent Earth System Models (ESMs) that are part of the Coupled Model Intercomparison Project 5 (CMIP5)
predict decreases in global export production (defined as the sinking POC flux at 100m) of ~10-20% by 2100 (Bopp
et al., 2013; Cabré et al., 2015a) and ~30% by 2300 (Moore et al., 2018). In many of these models, primary production
and subsequent carbon export are largely limited by the physical supply of nutrients to the surface ocean, which is
predicted to slow with future warming (Cabré et al., 2015a; Fu et al., 2016; Laufkötter et al., 2015; Moore et al., 2018).
Mechanisms driving this nutrient supply slowdown include: (i) surface warming-induced stratification of the water
column, which will shoal winter mixed layers, limit vertical exchange, and "trap" nutrients in the ocean interior (Bopp
et al., 2013; Cabré et al., 2015a; Capotondi et al., 2012; Moore et al., 2018), and (ii) a weakening of the trade winds,
which will reduce upwelling rates and vertical nutrient supply in tropical oceans (Bopp et al., 2001; Collins et al.,
2010), as well as lateral Ekman-driven nutrient supply into the subtropics (Letscher et al., 2016).
Changes in the POC flux itself, however, also have the potential to modulate nutrient supply to the surface
ocean and therefore impact export. Because particles release nutrients when they decompose, the depth scale of
particle remineralization determines the proximity of these nutrients to the surface and their resupply rate to the
euphotic zone (Kwon et al., 2009; Yamanaka & Tajika, 1996). Shallow remineralization in mesopelagic waters,
especially above the permanent pycnocline, drives rapid nutrient recirculation to the surface; nutrients remineralized
in deeper waters, on the other hand, can take hundreds of years to re-emerge at the surface (Martin et al., 1987;
Matsumoto, 2007b). This raises the possibility of feedback loops in which changes in particle remineralization depth
might either dampen (negative feedback) or enhance (positive feedback) circulation-driven decreases in primary
production and export. For instance, increasing ocean temperatures may speed up bacterial remineralization rates
(Cavan et al., 2019; Cram et al., 2018; John et al., 2014; Laufkötter et al., 2017; Marsay et al., 2015; Matsumoto,
2007a) and enhance recycling of nutrients near the surface, which would dampen physically-driven decreases in
surface nutrient concentrations and result in a negative feedback on export. Oxygen concentrations, on the other hand,
are predicted to decrease with future warming (Bopp et al., 2002; Cabré et al., 2015b; Keeling et al., 2010; Long et
al., 2016; Matear & Hirst, 2003; Schmidtko et al., 2017) and slow bacterial remineralization and zooplankton-mediated
particle disaggregation rates (Cavan et al., 2017; Devol & Hartnett, 2001; Hartnett & Devol, 2003; Laufkötter et al.,
2017; Van Mooy et al., 2002). This would result in deeper particle remineralization and further exacerbate circulation-
driven nutrient supply decreases, leading to a positive feedback on export production. A decrease in mineral ballasting
and protection of particles with ocean acidification may also feedback negatively on export decreases by shoaling
remineralization depths (Hofmann and Schellnhuber, 2009).

Future changes in sinking particle size may also lead to strong feedbacks on export. Recent work has shown

that particle size, through its influence on sinking speed (Alldredge and Gotschalk, 1988; Smayda, 1971), plays a
paramount role in determining remineralization length scales and carbon transfer efficiency to depth (Cram et al.,
2018; Kriest & Oschlies, 2008; Weber et al., 2016). Potential mechanisms that could drive changing particle sizes
include changes in underlying phytoplankton community structure and organic matter packaging processes at higher
trophic levels. Whatever the mechanism, the direction and magnitude of the particle size-remineralization feedback in
a warming ocean will depend on how particle sizes change as export declines in the future. If the export decline is
associated with a shift towards larger organic particles that sink more quickly, remineralization depths will deepen
and further reduce surface nutrient supply and export in a positive feedback. If, on the other hand, export decreases
are associated with a shift towards smaller sinking particles, shallower remineralization will allow faster nutrient
recirculation to the surface and dampen stratification-driven decreases in nutrient supply and export in a negative
feedback.

Despite the potential importance of particle size, CMIP5 models do not resolve dynamic particle size

distributions and so cannot fully capture biological feedbacks driven by particle size (Laufkötter et al., 2016; Le Quéré
et al., 2005; Séférian et al, 2020). More complex models that resolve aggregation-disaggregation transformations
and/or particle size distributions have been developed (Gehlen et al., 2006; Jokulsdottir and Archer, 2016; Kriest &
Oschlies, 2008; Niemeyer et al., 2019; Schwinger et al., 2016), but have not been used to examine the interactions
between climate change, particle size, and export production.  Furthermore, parameters and processes in most previous
models are not constrained by observations of particle size distributions or the relationships between particle size and
export.

Here we combine new data analyses and idealized model experiments to assess the potential impact of

feedbacks induced by dynamic particle size-dependent remineralization depths on future export changes. We use
remotely-sensed datasets to empirically constrain the relationship between export rates and sinking particle size, then
implement this relationship in a 3-D global biogeochemical model that resolves particle size distributions. Together,
these analyses reveal a *negative* particle size-remineralization (PSR) feedback effect on export, suggesting that ESMs
lacking these interactions may overestimate the decrease in ocean carbon export during the 21$^{st}$ century.

**2 Methods**

**2.1 Ocean biogeochemical and particle remineralization model**

**2.1.1 Model setup**

We quantified the PSR feedback using an idealized ocean biogeochemical model, which comprises a simple

nutrient cycle (DeVries et al., 2014) embedded within the observationally-constrained Ocean Circulation Inverse
Model (OCIM) (DeVries, 2014). OCIM assimilates passive and transient tracer data to generate an annual-mean
circulation that realistically reproduces water mass distributions and ventilation rates at 2-degree horizontal resolution
on 24 vertical layers. The circulation rates are stored in a transport matrix ($A$), that quantifies physical exchanges
between every grid cell in our model. Thus, all physical (advective and diffusive) fluxes of tracer $X$ in our model are
represented by the matrix-vector product $A*[X]$. OCIM has previously been used successfully for high-fidelity
simulation of nutrients (DeVries, 2014) and oxygen (DeVries and Weber, 2017) and does not suffer from the equatorial
biases often evident in dynamical models with the same resolution. Nutrient cycling comprises phytoplankton
phosphate ($PO_4^{3-}$) uptake and export as sinking organic particles out of the surface ocean (<75m), particle
remineralization in the subsurface (>75m), and production and decomposition of dissolved organic phosphorus
(DeVries et al 2014). Nutrient concentrations in the ocean interior represent the sum of preformed nutrients,
transported from regions of incomplete utilization in the ocean surface, and the accumulated product of particulate
and dissolved organic matter remineralization.

Vertical particle fluxes are simulated by the 1-D mechanistic Particle Remineralization and Sinking Model
(PRiSM) (DeVries et al., 2014). PRiSM computes particle flux profiles as a function of particle size distribution at the
surface, microbial remineralization rate, and empirical relationships between particle size, mass, and sinking velocity.
These empirical relationships are in some cases derived from measurements of sinking phytoplankton and in other
cases from those of sinking particles or porous aggregates. PRiSM therefore implicitly assumes that phytoplankton
and smaller particles behave similarly as they sink down the water column.

Particle abundances in the ocean tend to follow a power-law distribution, with many more small particles
than large ones (Boss et al., 2001; Buonassissi and Dierssen, 2010; Cael and White, 2020; Sheldon et al., 1972; White
et al., 2015). Thus, PRiSM produces particles in the surface euphotic zone (<75m) following a power-law size
spectrum, in which the log of the particle number density declines linearly with the log of the particle diameter,
between the sizes of 20 and 2000 um in diameter. Accordingly, the relative abundance of small and large particles is
controlled by the slope of the spectrum on a log-log scale ($\beta$): a shallower slope (small $\beta$) indicates a greater proportion
of large particles relative to small ones, while a steeper slope (large $\beta$) indicates a smaller proportion of large particles.
This surface particle size distribution slope is defined via specification of a global $\beta$ map. Previous work with PRiSM
has demonstrated that spatial variations in annual mean $\beta$ of the magnitude observed by satellite can lead to large
differences in particle fluxes at depth (Fig. S1; Fig. 1a-b; DeVries et al 2014).

Following export, the simulated particle size spectrum evolves through the water column due to
remineralization and size-dependent sinking. Remineralization is represented by first-order mass loss from particles,
such that each individual particle shrinks and sinks more slowly with depth. Because smaller, slower-sinking particles
reside for longer within any given depth interval and therefore have more time to remineralize, they are preferentially
lost from the particle population over depth. A constant rate of microbial respiration is used, optimized to fit global in
situ phosphate distributions (DeVries et al., 2014). There are therefore no temporal changes in bacterial respiration
due to warming, for example, which allows us to isolate changes in export that stem from the PSR feedback alone.
While PRiSM has recently been expanded to include temperature and oxygen effects on bacterial respiration and

remineralization (Cram et al., 2018), as well as to represent particle disaggregation (Bianchi & Weber et al., 2018), here we use the original version described in DeVries et al. (2014), which can be solved analytically and has previously undergone parameter optimization to best fit global phosphate distributions.

The model configuration and parameter values used here are outlined in Table S1. Further model details and validation are described in DeVries et al. (2014). Here we extend the original PRiSM-enabled biogeochemical model in DeVries et al. (2014) in two important ways:

*1.)* The original diagnostic nutrient uptake term (i.e., nutrient-restoring production) is replaced by the prognostic organic matter production scheme developed by Weber and Deutsch (2012) with minor parameter updates (see Table S2). This scheme calculates phytoplankton growth, in terms of $PO_4^{3-}$ uptake, as a function of observed annual-mean temperatures (Locarnini et al., 2010) and solar radiation levels (Rossow & Schiffer, 1999), along with modeled $[PO_4^{3-}]$. This formulation successfully reproduces the broad spatial patterns of surface $[PO_4^{3-}]$ (Weber and Deutsch, 2012), suggesting that our model accurately captures the balance between preformed and remineralized nutrients in the ocean interior. Of the organic phosphorus produced by uptake in the euphotic zone, 10% is routed to dissolved organic matter, which circulates and degrades over time, with the remainder being routed to particulate organic matter (Thornton, 2013). An empirical, spatially variable relationship between particulate C-to-P ratios and phosphate concentrations (Galbraith & Martiny, 2015) is then used to convert particulate organic phosphorus fluxes into POC fluxes.

*2.)* We add the ability to enable or disable the PSR feedback by implementing an empirical relationship that links changes in particle size spectrum slope ($\beta$) to changes in carbon export out of the surface ocean (<75 m) (see Section 2.1.2).

**2.1.2 Model representation of the PSR feedback**

When the PSR feedback is disabled within our model, circulation-driven changes in the nutrient supply to the euphotic zone (see Section 2.3) will lead to changes in POC export, but $\beta$ (and therefore particle remineralization depths) remains constant over time. With the PSR feedback enabled, any change in POC export is accompanied by a change in $\beta$, the direction and magnitude of which is specified using the empirical relationships discussed in Section 2.2. We note that by design, this modeling approach makes no assumptions about the mechanisms driving shifts in the particle size distribution, only that $\beta$ changes in tandem with POC export, in a manner that is consistent with observations. Mathematically, $\beta$ is updated at a given grid point as follows between timesteps *t* and *t+1*:

$$\beta_{t+1} = \beta_t + \frac{d\beta_{sat}}{dE_{n,sat}} \frac{E_{t+1} - E_t}{E_t},\tag{1}$$

where $E$ is the modeled export rate and $\frac{d\beta_{sat}}{dE_{n,sat}}$ is the empirical, time-independent fractional change in satellite-derived $\beta$ ($\beta_{sat}$) per change in satellite-derived, time-mean normalized export ($E_{n,sat}$, defined as absolute export divided by time-mean export calculated between 1997 and 2010 at a given grid point–see Section 2.2.2 for details).

To disable the feedback, $\frac{d\beta_{sat}}{dE_{n,sat}}$ is set equal to zero so that modeled $\beta$ remains constant over time. To enable

the feedback, $\frac{d\beta_{sat}}{dE_{n,sat}}$ is set equal to the linear temporal regression coefficient between $\beta_{sat}$ and $E_{n,sat}$, which is computed
from remotely-sensed time series of the two variables at each grid cell over the global ocean (Section 2.2). Thus, when
the feedback is enabled, changes in modeled $\beta$ over time are dictated by the magnitude of modeled export change as
well as the strength and direction of the relationship between observed $\beta$ and export, which can vary spatially.

**2.2 Empirical analyses of phytoplankton size, $\beta$, and export from satellite data**

Because the strength and direction of our modeled PSR feedback depends strongly on the observed

relationship between $\beta$ and export ($\frac{d\beta_{sat}}{dE_{n,sat}}$ in Eq. (1)), we sought a robust empirical constraint on this relationship.
Sections 2.2.1 and 2.2.2 respectively describe the global satellite-derived time series maps of $\beta$ and export used here.
Section 2.2.3 then describes how these monthly-mean $\beta$ and export maps are used to compute a range of possible
global $\frac{d\beta_{sat}}{dE_{n,sat}}$ relationships.

**2.2.1 Global satellite-derived particle size distribution map**

Global 1/12°-by-1/12° monthly maps of $\beta$ observed by the satellite Sea-viewing Wide Field-of-view Sensor

(SeaWiFS, in operation from September 1997 – December 2010) were downloaded from
ftp://ftp.oceancolor.ucsb.edu//pub/org/oceancolor/MEaSUREs/PSD/. These $\beta$ maps were derived from remotely-
sensed particulate backscattering measurements, which were previously validated with in situ near-surface Coulter
counter measurements (Kostadinov et al., 2009). To enable more efficient computation, we reduced the resolution of
the original monthly $\beta$ maps to 1°-by-1° degree via spatial averaging. At this resolution, time-mean $\beta$ ranges from
~3.3 in coastal high-latitude regions (where high nutrient conditions favor larger phytoplankton) to ~5.3 in the
subtropics (where low macronutrient concentrations favor small phytoplankton) (Fig. 1a). Although $\beta$ from
Kostadinov et al. (2009) is computed only over particle sizes ranging from 0.002 to 63 um, we assume that the same
$\beta$ continues to hold for larger particles up to 2000 um (the largest particle size in PRiSM), as supported by prior
research (e.g., Durkin et al., 2015). Ideally, measurements of $\beta$ would be computed over the same particle size range
as simulated in PRiSM (20–2000 um); however, such a dataset was not readily available. Indeed, the Kostadinov et
al. (2009) observations of $\beta$ were the only readily available measurements spanning long enough timescales, with high
enough spatiotemporal resolution to compute the relationships between $\beta$ and POC export needed for this study.

**2.2.2 Global satellite-derived export maps**

POC export was computed as the product of net primary production (NPP) and the particle export ratio

(export/NPP, or e-ratio), both of which can be derived from satellite data. To create a range of plausible global monthly
export maps, we multiplied all possible permutations of three monthly NPP estimates and e-ratio algorithms, yielding
nine distinct monthly datasets of global export spanning >10 years (Fig. S2). All three sets of monthly satellite NPP
maps were downloaded from http://sites.science.oregonstate.edu/ocean.productivity/ and derived from SeaWiFS
observations processed through the following algorithms: (i) the chlorophyll-based Vertically Generalized Production
Model (VGPM) (Behrenfeld & Falkowski, 1997); (ii) the Eppley-VGPM model (VGPME), containing a modified
relationship between temperature and production compared to the original VGPM (Carr et al., 2006); and (iii) the
Carbon-based Production Model (CbPM), which uses particulate backscatter-derived carbon rather than chlorophyll
to measure phytoplankton biomass (Behrenfeld et al., 2005). The three e-ratio algorithms we used were the ecosystem-
model based relationship of Laws et al., 2000, and the empirical relationships of Dunne et al. (2005) and Laws et al.
(2011). All three of these algorithms link e-ratio to SST and NPP. The in situ, statistically interpolated SST dataset
used here was NOAA's Extended Reconstructed Sea Surface Temperature (ERSST) v3b, downloaded from
https://www1.ncdc.noaa.gov/pub/data/cmb/ersst/v3b/netcdf/ (Smith et al., 2008). Euphotic zone depths needed to
compute D2005 e-ratios were derived from SeaWiFS-sensed chlorophyll concentrations (downloaded from the same
website as NPP) according to Equation 10 in Lee et al. (2007). As with $\beta$, all variables were computed and stored on
a 1°-by-1° degree grid over the entirety of the SeaWiFS period (September 1997 – December 2010, 160 months long).

In the following computations of $\frac{d\beta_{sat}}{dE_{n,sat}}$ (Section 2.2.3), we employed all nine sets of global monthly export

maps to propagate uncertainty into our assessment of the PSR feedback. When computing most-likely $\frac{d\beta_{sat}}{dE_{n,sat}}$ values,
we weighted the nine export map sets according to how well each map set's annual mean export matches in situ
oxygen and mass balance-based observations (Emerson, 2014; Reuer et al., 2007) within each region defined here
(Table S3; see Weber et al. (2016) for derivation of weighting factors). Fig. 2 shows the weighted annual mean carbon
export flux over the nine map sets, as well as the regions used for weighting, which are delineated based on
biogeochemical characteristics such as sea surface temperature and surface phosphate concentrations (Weber et al.,
2016). The Atlantic and Pacific Oceans are divided into warm subtropics dominated by smaller picophytoplankton
(STA, STP), cold subarctic regions dominated by blooms of larger microphytoplankton in the north (NA, NP), and
cool tropical upwelling zones dominated by larger phytoplankton in the east (ETA, ETP). The Indian Ocean is kept
intact (IND), while the Southern Ocean is divided into the productive, diatom-dominated Subantarctic Zone (SAZ)
and the high-nutrient, low-chlorophyll Antarctic Zone (AAZ). The Indian Ocean region (IND) did not contain a
sufficient number of in situ observations of export to enable comparison to the satellite export maps, so all nine maps
are weighted equally there.
**2.2.3 Regionally variable empirical $\beta$ versus export relationships ($\frac{d\beta_{sat}}{dE_{n,sat}}$)**

We quantified the empirical relationship between $\beta$ and export individually for each grid cell by extracting

the monthly timeseries (September 1997 – December 2010) of $\beta$ and normalized export ($E_n$) from the satellite products
described above, and then applying linear regression. This process produced a spatially variable, 1°-by-1° degree
global map of the best-fit linear slopes ($\frac{d\beta_{sat}}{dE_{n,sat}}$) relating $\beta$ and $E_n$. To capture the range of plausible $\frac{d\beta_{sat}}{dE_{n,sat}}$ maps, we
repeated this process for each of the nine export products to generate nine distinct global $\frac{d\beta_{sat}}{dE_{n,sat}}$ maps (Fig. S3). To
smooth out small-scale noise and illuminate large-scale patterns in the $\beta$ vs. export relationship, we spatially averaged
the $\frac{d\beta_{sat}}{dE_{n,sat}}$ slopes over the ocean biogeochemical regions defined in Fig. 2 (Fig. 3a). Finally, we set all grid points
within a given region equal to that region's weighted (Table S3; Section 2.2.2) mean value (Fig. 3b) to generate the
final $\frac{d\beta_{sat}}{dE_{n,sat}}$ map used in our PSR feedback-on runs (Fig. 3c).

To quantify the sensitivity of $\frac{d\beta_{sat}}{dE_{n,sat}}$ to the choice of export map used, we computed upper and lower-bound

$\frac{d\beta_{sat}}{dE_{n,sat}}$ maps by adding and subtracting one standard deviation (error bars in Fig. 3b) to the weighted regional mean
$\frac{d\beta_{sat}}{dE_{n,sat}}$ values. Conducting PSR feedback-on runs using upper and lower-bound $\frac{d\beta_{sat}}{dE_{n,sat}}$ maps establishes the range of
PSR feedback strengths we can reasonably expect from our model forced with empirically-derived relationships.

**2.3 Model runs to simulate future ocean warming and quantify the PSR feedback effect**

To represent present-day conditions, we run a baseline simulation with modern-day circulation rates to steady

state. To simulate increased water column stratification and reduced vertical exchange due to warming in an idealized
way, we uniformly and instantaneously reduce circulation and diffusion rates by 10% throughout the ocean (i.e. we
multiply the tracer transport matrix $A$ by 0.9, such that circulation patterns remain unchanged but the absolute
exchange rates between all grid cells are scaled down by 10%). For comparison, observations show that the Atlantic
meridional overturning circulation (AMOC) has weakened by about 15% since the mid-20[th] century due to
anthropogenic warming (Caesar et al., 2018), while ESMs project that AMOC will weaken by 11-34% over the 21[st]
century, depending on the chosen radiative forcing scenario (11% assumes the "high mitigation" RCP2.6 scenario,
while 34% assumes the "business-as-usual" RCP8.5 scenario) (Collins et al., 2019). A 10% decrease in circulation
rates is therefore a relatively conservative estimate of the effects of anthropogenic warming. Although modulation of
ocean circulation rates in response to climate change will be more complicated and variable than the uniform 10%
decrease applied here (e.g., Toggweiler and Russell, 2008), we seek only a simple, idealized way to approximate the
reduced surface nutrient supply that is expected in a warmer future ocean. Although we use a simplified representation
of future changes in ocean circulation, the exact same simplified representation is implemented in both PSR feedback-
on and -off simulations. We are thus isolating the effects of the PSR feedback from the effects of the circulation
change. It is therefore not unreasonable to assume that our calculated PSR feedback strength would be comparable to
that computed from a physical model with a more complex representation of future circulation changes, as long as
that model also applied identical circulation changes in PSR feedback-on and -off scenarios.

To quantify the impact of the global PSR feedback on export changes with future warming, we run the slower-

circulation rate simulation with and without the PSR feedback effect enabled. In feedback-off runs, $\beta$ is set equal to
annual mean values (Fig. 1a) for the entire duration of the run. In feedback-on runs, $\beta$ is initially set equal to annual
mean values, but is allowed to change according to Eq. (1), with $\frac{d\beta_{sat}}{dE_{n,sat}}$ defined as in Fig. 3c for the entire duration of
the run. Additional feedback-on runs were conducted using the upper and lower-bound $\frac{d\beta_{sat}}{dE_{n,sat}}$ maps (described in
Section 2.2.3).

All of the above described runs were also repeated with 10% faster circulation rates to determine whether the

PSR feedback strength is symmetrical with regard to the direction of circulation change. Within all runs, $\beta$ is
constrained to realistically remain between 2 and 6.5 at all grid points, though these extremes are rarely reached. We
run all experimental simulations for 100 years (initializing with conditions from the end of the present-day spin-up)
to study near-future changes in export production and nutrient distributions, and to facilitate comparison with 100-
year changes projected by the state-of-the-art Earth System Models discussed above.
**3 Results and Discussion**
**3.1 Empirically-derived, spatially-resolved $\beta$ versus export relationships ($\frac{d\beta_{sat}}{dE_{n,sat}}$)**

No matter which export datasets are used (Section 2.2), satellite-derived $\beta$ and export are strongly negatively

correlated (Fig. 3; Fig. S3-4). The vast majority of variance in both $\beta$ and export occurs over seasonal (rather than
interannual or longer) timescales, and therefore the coincident seasonal cycles of $\beta$ and export account for much of
the relationship between the two variables (Fig. S3-4). Because $\beta$ and export are negatively correlated, export tends to
be high when $\beta$ is small (particles are large) and low when $\beta$ is large (particles are small). These empirical findings
are in agreement with Cram et al. (2018), who observed that large particles tend to comprise a larger fraction of the
sinking flux where productivity and carbon export are high.

While our analysis does not provide mechanistic insights into the roots of the negative correlation between $\beta$

and export, a plausible explanation for the direction of this relationship is as follows. Low-nutrient conditions select
for small phytoplankton with high surface area-to-volume ratios, such that smaller phytoplankton are more abundant
in low-nutrient conditions (Litchman et al., 2007). In these nutrient-limited regions of the ocean, productivity and
export are also suppressed. Thus, nutrient availability controls both the export rate and the size structure of the
phytoplankton community over much of the ocean. Assuming that phytoplankton size in turn controls the size of
sinking particles, as suggested by past research (e.g., Guidi et al., 2007; Guidi et al., 2008; Guidi et al., 2009), the
availability of nutrients then ultimately controls sinking particle size as well. This potentially explains why small
particles (large $\beta$) are associated with reduced export rates and low-nutrient conditions, while large particles (small $\beta$)
are associated with increased export rates and high-nutrient conditions.

This line of reasoning may also explain why regions that are more nutrient-limited (i.e., the subtropics)

exhibit especially strong negative relationships between $\beta$ and export (Fig. 3; Fig. S3-4); both $\beta$ and export are likely
predominantly driven (in opposite directions) by surface nutrient supply in these areas. Where light or temperature
take over as the dominant factors limiting phytoplankton productivity, the relationships between $\beta$ and export are
weakened, as in the higher-latitude regions (Fig. 3; Fig. S3-4). The counterintuitive weakly positive relationship
between $\beta$ and export in the Subantarctic Zone (SAZ) of the Southern Ocean (Fig S5) is in line with findings from

Lam and Bishop (2007), who showed that in the Southern Ocean, areas with higher biomass and larger particles at the surface were actually associated with lower rates of export out of the euphotic zone. In these diatom-dominated regions, zooplankton may be more active and have higher particle grazing efficiencies, leading to faster attenuation of particulate carbon fluxes with depth. The unique relationship between β and export in the SAZ is worth further exploration, and may be further elucidated by NPP datasets that are specifically calibrated for the Southern Ocean (e.g. Johnson et al., 2013), but is beyond the scope of the current study.

**3.2 An empirical negative global particle size-remineralization (PSR) feedback**

To determine how the empirically-derived relationships between β and export ultimately affect the direction and strength of the PSR feedback effect on a global scale, we must first understand the effects of β on sinking particle speeds and remineralization depths. Past work has broadly established a positive relationship between particle size and sinking speed in the ocean (Alldredge and Gotschalk, 1988; Smayda, 1971; Iversen and Ploug, 2010)—although there are exceptions to these rules (Cael and White, 2020; Laurenceau-Cornec et al., 2019), particularly in the Southern Ocean (McDonnell and Buesseler, 2010). The characteristic depth scale of particle remineralization is proportional to this sinking speed divided by a microbially-mediated remineralization rate (Kwon et al., 2009; McDonnell et al., 2015). Here we define remineralization depth as the depth at which POC flux out of the euphotic zone is reduced by a factor of $e$ or 63% (i.e., the $e$-folding depth of the flux) (Fig. 1b). The dominance of smaller sinking particles in the water column results in a shallower remineralization depth, as bacteria have more time to decompose these slow-sinking particles into nutrients and $CO_2$ as they pass through the upper layers of the water column (Bach et al., 2016). Together with the empirical relationships we found between POC export and particle size (Section 3.1), this points towards a predominantly *negative* feedback loop that would dampen the response of POC export to physically-induced changes in nutrient supply. In a warming and stratifying ocean, this hypothesized particle size-remineralization feedback would theoretically proceed through the following steps at any given location, which are illustrated schematically in Fig. 4:

*1.) Slower Circulation (SC)* – First, stratification of the water column and slowing trade winds with climate warming will reduce shallow subsurface nutrient concentrations and vertical exchange/upwelling rates. This slows nutrient supply into the euphotic zone, which in turn decreases phytoplankton productivity and resultant export production (Fig. 4a, b, green arrows).

*2.) Ecological Effect (EE)* – A decrease in surface nutrient supply also selects for smaller phytoplankton, which presumably leads to a larger proportion of small particles in the export flux. The net result of this ecological effect (EE) (Fig. 4a, red arrow) is captured in a predominantly negative relationship between export and β (Fig. 4c, red line). Constrained by this empirical relationship, changes in export and β under slowed circulation (SC) must fall along the red line in Fig. 4c ("SC with EE" point). In the absence of the ecological effect (i.e., phytoplankton/particle sizes are not affected by changes in the nutrient supply), there is no such requirement and β would remain unchanged under a slowed circulation scenario ("SC without EE" point in Fig. 4c).

*3.) Sinking Speed Effect (SSE)* – Smaller particles resulting from the ecological response to a reduced nutrient
supply would sink more slowly and therefore remineralize shallower in the water column. More regenerated
nutrients would then accumulate within shallower waters and thus recirculate more quickly to the surface. In
isolation, a shift to smaller particles would therefore ultimately lead to greater surface nutrient availability
and larger export rates (Fig. 4a, blue arrow), represented by the positive slope of the blue export-versus-$\beta$
line in Fig. 4d. In the presence of this sinking speed effect (SSE), changes in export and $\beta$ under slowed
circulation must fall along the blue sinking speed-related export-versus-$\beta$ line (Fig. 4d). In the absence of this
sinking speed effect (i.e., particle size does not affect sinking rates/remineralization depths), there is no such
requirement, and the initial stratification-induced export decrease would remain unaltered ("SC without SSE,
with EE" point in Fig. 4d).

Only in the presence of both the ecological and sinking speed effects does the PSR feedback function in full;
in this case, after circulation is slowed, export and $\beta$ must reach a new steady-state at the intersection of the red and
blue lines ("SC with SSE *and* EE" yellow star in Fig. 4d). Thus, the overall decrease in POC export would be smaller
than predicted from decreased circulation rates and surface nutrient supply alone. That is, the net effect of
phytoplankton selection and particle size-dependent remineralization depths provide a negative feedback on, or
dampening of, changes in export, due to the empirically-derived negative relationship between $\beta$ and export. While
we have assumed that phytoplankton community structure is the underlying mechanism linking POC export and
particle size, the PSR feedback would operate in the same direction discussed here if another mechanism were
ultimately responsible for the empirical negative relationship between these two factors. Though the above description
focuses on export decreases under decreased circulation rates, the PSR feedback would result in an analogous
dampening of export increases under increased circulation rates and surface nutrient supply.

### 354 3.3 Predicted export changes in the presence of the global negative PSR feedback effect

In this section, we discuss how predicted future changes in export production and mesopelagic POC flux
differ between biogeochemical model simulations with and without the PSR feedback effect applied globally. Sections
3.3.1 and 3.3.2/3.3.3 respectively examine resultant global and zonal/regional mean changes in export. Section 3.3.4
examines resultant global and zonal/regional mean changes in deeper POC fluxes.

### 359 3.3.1 Predicted global mean export changes with and without the global PSR feedback

To examine the global strength of the PSR feedback within our model under an idealized climate change
scenario, we compare global mean export changes over time in the PSR feedback-on and off runs after a 10% decrease
in circulation rates (Fig. 5, comparing slower circulation dashed and solid lines). In both the feedback-on and off
cases, instantaneously decreasing circulation rates reduces surface nutrient supply and immediately leads to a sharp
decrease in global mean export of ~0.2 molC m$^{-2}$ yr$^{-1}$ from 3.54 molC m$^{-2}$ yr$^{-1}$. After this initial plunge, global mean
export declines by an additional 0.09 molC m$^{-2}$ yr$^{-1}$ over the next 100 years with the feedback off (for a total decrease
of 0.29 molC m$^{-2}$ yr$^{-1}$ or 8.1%), versus an additional 0.05 molC m$^{-2}$ yr$^{-1}$ with the feedback on (for a total decrease of
0.25 molC m$^{-2}$ yr$^{-1}$ or 7.0%) (Fig. 5, slower circulation lines and bars).
Turning the PSR feedback on in our model reduced the total 100-year predicted decrease in export by ~14%
relative to the PSR feedback-off scenario (the ratio of the solid-colored bar length to the full bar length below zero in
Fig. 5. At equilibrium (when global mean export stabilizes ~500 years after decreasing circulation rates), this feedback
effect increases to ~16%. With the feedback turned on, particle sizes shrink and remineralization depths shoal in
response to an initial circulation-driven decrease in surface nutrient supply, thereby moderating this initial decrease
by keeping more recycled nutrients at the surface. In particular, global mean $\beta$ increases by 0.03 (from 4.34 to 4.37)
under 10% decreased circulation rates after 100 years with the PSR feedback on (Fig. 6a,b), corresponding to a 17 m
global mean shoaling (from 595 to 578 m) of e-folding remineralization depths (Fig. 6b). The greatest regional mean
$\beta$ increase of 0.06 occurs in the Indian Ocean (IND), resulting in a 41 m shoaling of remineralization depths there
(Fig. 6b). Results from runs employing upper and lower-bound $\frac{d\beta_{sat}}{dE_{n,sat}}$ maps (defined in Section 2.2.3, represented by
the error bars in Fig. 4b) lend further support to our findings and indicate that the modeled global PSR feedback effect
size is relatively insensitive to the choice of export maps used to compute $\frac{d\beta_{sat}}{dE_{n,sat}}$ (Fig 5, black error bars).
The PSR feedback also dampens the response of global-mean carbon export to an instantaneous *increase* in
ocean circulation rates (Fig. 5). One hundred years after circulation rates are increased by 10%, global mean carbon
export increases from 3.54 molC m$^{-2}$ yr$^{-1}$ by 0.28 molC m$^{-2}$ yr$^{-1}$ (8.0%) with the feedback off, whereas it increases by
~0.23 molC m$^{-2}$ yr$^{-1}$ (6.6%) with the feedback on (Fig. 5, faster circulation lines and bars). Thus, increasing circulation
rates by 10% with the PSR feedback on reduces the 100-year increase in export production by ~18% (the ratio of the
solid-colored bar length to the full bar length above zero in Fig. 5). At equilibrium, this feedback effect increases to
~20%. With the feedback turned on, particle sizes grow and remineralization depths deepen in response to an initial
circulation-driven increase in surface nutrient supply, thereby moderating this initial increase by transferring more
nutrients to deeper waters where they recirculate more slowly to the surface. In particular, global mean $\beta$ decreases by
0.03 units (from 4.34 to 4.31) under 10% increased circulation rates, corresponding to a 20 m global mean deepening
(from 595 to 615 m) of e-folding remineralization depths (not shown). The greatest regional mean $\beta$ decrease of 0.07
occurs in the Indian Ocean (IND), resulting in a 54 m shoaling of remineralization depths there. Compared with the
decreased circulation case, absolute changes in remineralization depths are slightly larger under increased circulation
rates because remineralization depth changes are more sensitive to variations in $\beta$ when particles are larger (that is, at
smaller values of $\beta$). Because remineralization depth changes are greater under increased circulation rates, so too is
the global PSR feedback strength (14% with decreased circulation rates versus 18% with increased circulation rates).
Again, results from PSR feedback-on runs constrained by upper and lower-bound $\frac{d\beta_{sat}}{dE_{n,sat}}$ maps further support the
notion that the PSR feedback size is relatively insensitive to the choice of export maps used to compute $\frac{d\beta_{sat}}{dE_{n,sat}}$ (Fig 5,
error bars). Thus, the effect of the PSR feedback is to buffer changes in export production in response to any physical
perturbation in nutrient supply, regardless of the direction.
The strength of the PSR feedback also does not depend on the size of circulation rate changes. Indeed, we
observed that PSR feedback strength remains constant whether circulation rates are increased/decreased by 10% or
50%. Thus, the percentage difference in projected export change between PSR feedback on and off cases is relatively
uniform even under quite different changes in circulation rates.
**3.3.2 Predicted zonal and regional mean export changes *without* the global PSR feedback**
In our baseline simulation under current-day circulation rates, POC export covaries tightly throughout the
low to mid-latitudes with nutrient concentrations in shallow subsurface waters beneath the euphotic zone, quantified
here as [PO4] at 200m depth ($P_{200m}$) (Fig. 7a,b; Fig. 8a). South of ~40°S and north of ~40°N, other factors such as light
and/or temperature become limiting; as a result, export does not vary as tightly with $P_{200m}$ in these higher-latitude
regions. The spatial structure of the relationship between export and $P_{200m}$ confirms that nutrient supply from
subsurface layers is the primary driver of export rates throughout the nutrient-limited low- to mid-latitudes. Therefore,
in these regions, the following balance approximately holds:
$Export = E \approx wP_{200m},$                       (2)
where $w$ is the local upwards nutrient supply velocity, which represents the net effect of all vertical exchange
processes, including diffusion, upwelling, entrainment, and mixing. This relationship between export, upwelling, and
subsurface nutrient concentrations reflects the common assumption that at steady-state, export flux out of the euphotic
zone must approximately balance the supply of nutrients into the euphotic zone by upwelling (e.g. Ducklow et al.,
2001; Passow and Carlson, 2012). This balance can in turn be used to derive (via perturbation analysis) a simple,
approximate diagnostic for understanding changes in export under altered circulation rates at any given location:
$\Delta E = \Delta w * P_{200m,baseline} + w_{baseline} * \Delta P_{200m},$             (3)
where *baseline* denotes variables from the baseline simulation ran to steady-state with current-day circulation rates
and $\Delta$ denotes change from the baseline simulation under altered circulation rates. (Note that we ignore the
"perturbation product" term, $\Delta w * \Delta P_{200m}$, because it is negligible.)
Though Eqs. (2-3) are not mathematically equivalent to the full model solution, they explain much of the
full model's behavior and provide us with a tool to simplify, deconvolve, and better understand the different
mechanisms leading to export changes. In particular, when ocean circulation is slowed, Eq. (3) allows us to identify
two different contributions to the resultant reduction in export through the low to mid-latitudes. First, and most
intuitively, when circulation rates are uniformly decreased, $w$ is reduced across the entire ocean ($\Delta w < 0$) and the
supply of "baseline" nutrients is curtailed. Second, a decrease in circulation rates also reduces phosphate
concentrations throughout the shallow subsurface layer in the low to mid-latitudes ($\Delta P_{200m} < 0$) (solid lines and
bars in Fig 7c,d; Fig. 8b). This decrease in $P_{200m}$ is likely largely driven by enhanced biological nutrient utilization in
the surface of the Southern Ocean in response to slower circulation, which is then propagated into the low to mid-
latitude interior through Antarctic Intermediate and Subantarctic Mode Waters (e.g., Sarmiento et al., 2004; Marinov
et al., 2006), as observed in future climate simulations by more complex ESMs (e.g., Moore et al., 2018).
Together, the decreases in shallow subsurface nutrient concentrations ($P_{200m}$) and vertical exchange rates ($w$)
result in substantial reductions in export throughout most of the ocean under our decreased circulation simulations as
dictated by Eq. (3), with the greatest reductions occurring in nutrient-limited areas. In the absence of the PSR feedback,
the 10% decrease we imposed on circulation rates leads to 100-year zonal mean export decreases of >15% at 35°N
and S and ~10% between 35°N and S (solid line in Fig. 7e). Regionally, the oligotrophic subtropics (especially the
STP) exhibit the largest relative decreases in export (~10-13%), followed closely by the tropics (ETA, ETP) with
export decreases around 8-10% (solid bars in Fig. 7f). As expected, the decrease in export mirrors the pattern of $\Delta P_{200m}$
in low to mid-latitude regions due to a strong dependence of export on nutrient supply from the shallow subsurface
here.

### 3.3.3 Predicted zonal and regional mean export changes *with* the global PSR feedback

As with the global mean (Section 3.3.1), we quantify zonal and regional mean PSR feedback strength as the
difference in circulation-driven export change from baseline between the feedback-on and -off runs, normalized by
the change from baseline in the feedback-off run. In other words, the PSR feedback strength is the percentage by
which turning on the PSR feedback reduces (dampens) the response of carbon export to changes in ocean circulation
(blue line and bars in Fig. 7g,h). Thus, the zonal mean feedback strength (blue line in Fig. 7g) is equal to the difference
between the dashed and solid lines divided by the solid line in Fig. 7e, while the regional mean PSR feedback strength
(blue bars in Fig. 7h) is equal to the length of the solid-colored portion of the bars divided by the entire length of the
bars in Fig. 7f. The PSR feedback strength is greatest (most strongly damping) in the low to mid-latitudes and in the
tropics (ETA, ETP) and subtropics (STA, STP, IND), with the feedback able to reduce zonal and regional mean export
changes by up to 20% in these regions (blue lines and bars in Fig. 7g,h). To understand this spatial pattern, we combine
Eq. (3) with our definition of PSR feedback strength to yield the following diagnostic, which can help separate out the
various determinants of PSR feedback strength:
$$PSR\ feedback\ strength = \frac{\Delta E_{on} - \Delta E_{off}}{\Delta E_{off}} \approx \frac{\frac{\Delta P_{200m,on} - \Delta P_{200m,off}}{P_{200m,baseline}}}{\frac{\Delta w}{w_{baseline}} + \frac{\Delta P_{200m,off}}{P_{200m,baseline}}}, \qquad (4)$$
where *on/off* denotes whether the PSR feedback was turned on or off under the altered circulation rates. This
expression reveals that the PSR feedback effect is strongest wherever activating the feedback leads to the greatest
dampening of changes in $P_{200m}$, compared to the changes that occur in the feedback-off case. In the decreased
circulation simulations ($\frac{\Delta w}{w_{baseline}}$ = -10% everywhere), the low to mid-latitude regions display the greatest differences
in $P_{200m}$ changes between feedback-on and off runs (Fig. 7c,d; Fig. 8b-d); these regions undergo the greatest reductions
in circulation-driven export change due to the PSR feedback (Fig. 7e,f) and thus exhibit the largest PSR feedback
effects (blue lines and bars in Fig. 7g,h).
The degree to which the PSR feedback dampens $P_{200m}$ changes is in turn driven by the strength of the
relationship between $\beta$ and export. The low to mid-latitudes exhibit the most negative $\frac{d\beta_{sat}}{dE_{n,sat}}$ values and therefore, the
tightest coupling between $\beta$ and export (Fig. 3c). In these regions, where macronutrient limitation is the dominant
constraint on productivity, a given circulation-driven decrease in surface nutrient supply causes a relatively large drop
in both export and phytoplankton/particle size (leading to an increase in $\beta$) in the presence of the PSR feedback. This
then allows significantly more nutrients to be recycled at the surface, resulting in greatly dampened decreases in $P_{200m}$
and subsequent export production.
Our simple diagnostic (Eq. (4), derived from Eqs. (2-3)) can explain PSR feedback strengths quite well over
much of the global ocean, as can be seen by comparing total feedback strengths (blue lines/bars in Fig. 7g,h) with our
diagnostic-derived feedback strengths (right-hand side of Eq. (4), represented by orange lines/bars in Fig. 7g,h), which
were calculated under the assumption that export is nearly equal to the supply of nutrients into the euphotic zone via
upwelling (Eq. (2)). In other words, Eqs. (2-4) are a good approximation to the full model solution where the orange
lines/bars lie relatively close to the blue lines/bars in Fig. 7g-h. However, new production can be fed by local upwelling
as well as lateral advection, such that changes in $P_{200m}$ and vertical exchange rates alone (orange lines/bars in Fig.
7g,h) cannot perfectly predict all changes in export (blue lines/bars in Fig. 7g,h), especially in regions where lateral
advection plays a relatively large role in supplying nutrients to the surface.
**3.3.4 Vertical reorganization of POC fluxes induced by the PSR feedback**
Up to this point, the effect of the PSR feedback effect has been analyzed solely for carbon exported out of
the surface euphotic zone (<75 m depth here). We largely focus on export out of the bottom of the euphotic zone
rather than on POC fluxes at greater depths because export, by definition, is a measure of the total organic carbon
supply that feeds subsurface heterotrophic communities. Thus, the PSR feedback effect buffers the productivity of
mesopelagic communities as a whole by damping changes in export.
This buffering of the food supply does not occur uniformly through the water column, however. While
shallower particle remineralization helps maintain the nutrient supply to the surface and buffers the POC export rate,
it also means that fewer particles persist at depth. Indeed, the PSR feedback on export has the opposite effect on
POC fluxes in the lower mesopelagic zone (at 800 m depth, equivalent to 725 m below the bottom of the euphotic
zone, for example – Fig. 9a-b). This results in a vertical re-organization of particle fluxes, with more food available
in the upper mesopelagic and less in the lower mesopelagic when the PSR feedback is activated. In particular, by
350 m depth (225 m below the bottom of the euphotic zone), the PSR feedback effect on POC flux has changed sign
from negative to positive across all ocean regions except for the SAZ, AAZ, and NA (Fig. 9c). In regions where the
negative PSR feedback effect is strongest, vertical reorganization is more extreme and the sign of the PSR feedback
flips from negative to positive shallower in the water column. The PSR feedback effect becomes positive below
depths as shallow as 125, 148, 158, and 165 m (50, 73, 83, and 90 m below the euphotic zone) in the NP, ETP, STA,
and IND regions, respectively, for example (Fig. 9c).
A negative PSR feedback effect dampens predicted circulation-driven changes in global export out of the
euphotic zone, while a positive PSR feedback effect amplifies predicted changes in lower mesopelagic global POC
fluxes. It follows that global models without the PSR feedback effect would *overestimate* changes in export, but
*underestimate* changes in POC fluxes at deeper depths in a future warming/slowed circulation scenario. This vertical
reorganization of POC flux changes brought about by the PSR feedback effect, leading to greater-than-expected
fluxes in the upper mesopelagic and lower-than-expected fluxes in the lower mesopelagic under future warming, has
the potential to alter follow-on predictions of vertical ecological community organization as well. Importantly, the
net effect of the PSR feedback is still a buffering of the total food supply to subsurface heterotrophic communities,
however.

### 3.4 Predicted export changes in the presence of regional PSR feedback effects

In this section, we discuss each individual ocean region's contribution to the global PSR feedback effect. To

isolate the PSR feedback effect coming from each region, we conduct a set of model runs in which we decrease the
circulation rate globally, but only activate the PSR feedback within one region at a time. In these feedback-on runs,
we set $\frac{d\beta_{sat}}{dE_{n,sat}}$ in Eq. (1) equal to zero at all grid points outside of the region we are isolating; within the isolated region,
we set $\frac{d\beta_{sat}}{dE_{n,sat}}$ equal to the corresponding empirically-derived value (as shown in Fig. 3b). These simulations are then
compared to the same feedback-off run discussed in Section 3.3 (i.e., no changes in $\beta$ anywhere) to determine the
impact of enabling the feedback within one region at a time. Sections 3.4.1. and 3.4.2 respectively describe the global
and regional mean export changes resulting from this set of experiments.

### 3.4.1 Predicted global mean export changes with and without regional PSR feedbacks

Analysis of the regional feedback-on runs show that tropical (ETA, ETP) and subtropical (STA, STP, IND)

regions contribute most significantly to the global PSR feedback (Fig. 10). Turning the feedback on in the ETP alone,
for instance, leads to a 3.9% reduction in global mean export change compared to the feedback-off case (Fig. 10a –
row 7, last column); the ETP alone thus accounts for 38.6% of the global PSR feedback strength (Fig. 10b – row 7,
last column), while spanning only 10.3% of total ocean area. Turning the feedback on in the subtropical (STA, STP,
IND) and tropical (ETA, ETP) regions one at a time and then summing their individual contributions (11.7%, 11.6%,
22.3%, 13.3%, 38.6% respectively; Fig. 10b – last column) accounts for 97.5% of the global PSR feedback effect,
while all other regions (AAZ, SAZ, NA, NP) account for only a negligible fraction of the effect (or even act to decrease
the overall effect in the case of the SAZ) (Fig. 10b – last column). The dominant contributions of the
tropical/subtropical regions to the global PSR feedback can once again be understood via spatial patterns in $\frac{d\beta_{sat}}{dE_{n,sat}}$
(Fig. 3c), with large changes in $\beta$ and remineralization depth associated with relatively small changes in export in the
nutrient-limited tropical/subtropical regions.

### 3.4.2 Predicted regional mean export changes with and without regional PSR feedbacks

The significant tropical/subtropical contribution to the PSR feedback can also be seen by examining export

changes within individual regions. Activating the PSR feedback in the STA, for example, dampens regional mean
export decreases within the STA, the ETA, and the NA by 7.7%, 3.1%, and 2.2%, respectively (Fig. 10a – row 3).
Turning on the feedback in the STP (Fig. 10a – row 4), ETA (Fig. 10a – row 6), or ETP (Fig. 10a – row 7) alone have
similarly large effects on surrounding regions. In contrast, activating the feedback within higher-latitude regions
(AAZ, SAZ, NA, NP) neither significantly moderates export decreases in any individual regions nor globally (Fig.
10a – rows 1-2, 8-9). The AAZ uniquely undergoes near-zero decreases in export for all runs with the feedback on or
off; PSR feedback strength here is therefore negligible (Fig. 10a,b – row 1).

When the PSR feedback is turned on within a given region, the effect is typically felt most strongly within

that same region, as would reasonably be expected given that export production and resultant remineralization are
spatially co-occurring (Fig. 10a,b – diagonal going from upper left to lower right corner). However, depending on the
local magnitude of $\frac{d\beta_{sat}}{dE_{n,sat}}$ compared to that of neighboring regions, as well as the connectivity of nutrient supplies
between them, there can be substantial PSR feedback effects originating from afar. For example, in the Pacific basin,
switching on the PSR feedback in the ETP has a stronger buffering effect on export in the STP region than switching
on the feedback in the STP itself (Fig. 10b – row 7, column 4). This is because the relationship between β and export
is much stronger in the ETP (with a regional mean $\frac{d\beta_{sat}}{dE_{n,sat}}$ of -0.40; see Fig. 3c) than in the STP (with a regional mean
$\frac{d\beta_{sat}}{dE_{n,sat}}$ of -0.18; see Fig. 3c), and because remineralized surface nutrients in the ETP are readily carried into the STP
by wind-driven Ekman transport. In this way, PSR feedback-driven buffering of surface nutrient supply changes
within the ETP indirectly buffers surface nutrient supply changes in the STP as well. This indirect effect also operates
in the reverse direction, in that activating the PSR feedback in the STP also has a relatively strong impact back on the
ETP (Fig. 10b – row 4, column 7). In this case, nutrients remineralized shallower in the STP thermocline are directed
along sloping isopycnals that eventually upwell into the ETP surface, thus buffering decreases in export there. The
STP also has a relatively large PSR feedback effect on the subpolar NP (Fig. 10b – row 4, column 9), due to the intense
flow of the Kuroshio Current, which carries surface nutrients from the STP northward.

Similar relationships hold in the Atlantic basin between the tropics, subtropics, and subpolar regions.

However, the PSR feedback effect of the ETA on the STA is smaller (Fig. 10b – row 6, column 3), while the effect of
the STA on the ETA is larger (Fig. 10b – row 3, column 6) compared to their Pacific counterparts, presumably due to
less pronounced Ekman divergence along the equatorial Atlantic. The STA's PSR feedback effect on the subpolar NA
(Fig. 10b – row 3, column 8) is also substantially more pronounced than the STP's effect on the NP, indicating a
stronger nutrient supply pathway between subtropical and subpolar gyres in the Atlantic Ocean via the Gulf Stream.

An interesting phenomenon that arises in the Southern Ocean is the negative (dampening) overall PSR

feedback effect on the SAZ (Fig. 10a – row 10, column 2), despite a positive (amplifying) local feedback effect (Fig.
10a,b – row 2, column 2) and relationship between β and export here (regional mean $\frac{d\beta_{sat}}{dE_{n,sat}}$ of +0.13; see Fig. 3c).
Additive negative (dampening) PSR feedback effects from surrounding regions (STA, STP, IND, ETA, ETP) (Fig.
10a,b – rows 3-7, column 2) overcome the small positive (amplifying) local feedback effect here (Fig. 10a,b – row 2,
column 2), such that the total feedback effect still reduces the magnitude of the regional mean export decrease by
1.2% compared to the feedback-off case (Fig. 10a – last row, column 2). Because the SAZ spans the entire width of
the ocean and touches every other basin, additional remineralized surface nutrients collected in the many connected
regions are quickly and easily circulated into the SAZ when the global PSR feedback is active, thus buffering larger
would-be decreases in export here.
**3.5 Comparison with CMIP5 models**
In the absence of the PSR feedback, our model predicts a 100-year global mean export decrease of 8.1%
(0.29 molC m$^{-2}$ yr$^{-1}$). With the PSR feedback on, this export decrease is reduced to 7.0% (0.25 molC m$^{-2}$ yr$^{-1}$).
Meanwhile, CMIP5 models project global mean export decreases of around 7-18% between 2090-2099 and 1990-
1999 under a "business-as-usual" radiative forcing scenario (RCP8.5), with an ensemble mean of 13% (Bopp et al.,
2013). Assuming that none of the CMIP5 models are currently able to simulate the PSR feedback, our results
therefore suggest that accounting for the feedback would alter the CMIP5 range of projections from a 7-18% to a 6-
15.5% decline in export, and the CMIP5 ensemble mean projection from a 13% to an 11% decline in export. Many
of the CMIP5 models may be capable of capturing some semblance of a PSR feedback effect, however, thus
necessitating smaller corrections.
All CMIP5 models simulate various processes that have the potential to change POC export fluxes in a
warming future ocean, including zooplankton grazing/fecal pellet formation, phytoplankton aggregation,
phytoplankton/zooplankton mortality, and variations in phytoplankton community structure (which provide the
source material for sinking particles) based on changing nutrient, temperature, and light conditions. However, 7 of
17 total CMIP5 models with ocean biogeochemistry simulate only one class or type of particulate organic matter
(Ilyina et al., 2013; Tjiputra et al., 2013; Tsujino et al., 2010; Watanabe et al., 2011; Zahariev et al., 2008), and
therefore cannot capture changes in the *nature* of sinking POC with future warming. The other 10 CMIP5 models
*can* simulate changes in the nature of sinking particulate organic matter with future warming, either through the
amount of associated ballasting (3 models) (Dunne et al., 2013; Moore et al., 2004) or through particle size (7
models).
Of the 7 models that simulate changes in particle size and could thus potentially capture the PSR feedback
effect, 3 of these models resolve two particle sizes (small and large) with different sinking speeds (3 m day$^{-1}$ for the
small particles and 50 to 200 m day$^{-1}$ for the large particles) (Aumont and Bopp, 2006). Another 2 of these 7 models
also resolve two size-based particle types (diatoms and detritus) with different sinking speeds (1 m day$^{-1}$ for diatoms
and 10 m day$^{-1}$ for detritus) (Totterdell, 2019). The final 2 of these 7 models simulate 5 different organic particle
sinking speeds based on size, 1 for each different phytoplankton type in the model (for a total of 4) and 1 for carbon
detritus (Romanou et al., 2013). In sum, out of the 17 CMIP5 models described here, only 7 resolve particles of
more than one size that sink down the water column at different speeds and therefore have the potential to capture
some part of the PSR feedback effect.
If all the CMIP5 models differed only in their resolution of sinking particle sizes, then we would expect the
7 models with more than one particle size to project the smallest decreases in export with future warming. In reality,
however, the models differ in too many other ways to isolate potential impacts of the PSR feedback when comparing
between them. For example, three of the models that resolve more than one sinking particle size (IPSL-CM5A-LR,
IPSL-CM5A-MR, and HadGEM2-ES) predict some of the largest decreases in export production by 2100 (Fig. 9b in
Bopp et al., 2013), contrary to what would be expected given the potential presence of a PSR feedback in these
models. Indeed, the models that can simulate changes in the nature of sinking particles project changes in export that
span the entire range of CMIP5 model predictions (Fig. 9b in Bopp et al., 2013). The reasons for these differences in
projected export decreases is difficult to disentangle and would require examining the effects of one mechanism at a
time on export in each model.
Simply representing differently sized particles also does not ensure that a model will adequately represent
the negative PSR feedback quantified in this study. To adequately represent the negative PSR feedback, a given
model would need to contain mechanisms that give rise to the same strong, empirically-derived relationships
between POC export and particle size that we specify here. Within the models that resolve particle size to some
degree, the relative proportion of large and small particles is determined by internal model dynamics and are not
prescribed empirically. Furthermore, these models do not resolve a particle size spectrum over a wide range of sizes
as is done here. Therefore, CMIP5 models that dynamically resolve 2-5 different particle size classes with different
sinking speeds might qualitatively reproduce the same feedback, but it is not clear whether the magnitude or even
the sign of the feedback would be accurately captured. We argue that our model, which resolves particle size spectra
over a wide range of sizes and employs empirical export/particle-size relationship constraints, is most likely to
accurately capture the true magnitude of the PSR feedback. We thus suggest that our study provides a reasonable
baseline against which more complex Earth System Models can assess their ability to reproduce particle size-
remineralization feedbacks.
**3.6 Caveats and future work**
The exact strength of the PSR feedback hinges on the empirical relationship between carbon export and
particle size, which may differ depending on the datasets used to constrain it. To address this uncertainty, we correlated
$\beta$ against a range of different global export datasets and found that our results were relatively insensitive to the choice
of export dataset. Unfortunately, well-grounded alternative global and temporally-resolved datasets for $\beta$ were not
readily available, so uncertainty in the PSR feedback strength due to uncertainties in observed $\beta$ could not be quantified
here. Analysis of in situ Underwater Visual Profiler (UVP) data suggests that $\beta$ may actually be smaller (thus particles,
larger) and less variable (Cram et al., 2018) than the backscatter-derived values (Kostadinov et al., 2009) used in this
study. This would potentially imply less variability in particle size-driven remineralization depths, weakening the PSR
feedback strength calculated here. On the other hand, differences in remineralization depths are greater at smaller
values of $\beta$ (Fig. S1; Fig. 2 in Devries et al., 2014), such that any given increase in $\beta$ associated with a decrease in
export would lead to greater shoaling of remineralization depths and a larger PSR feedback effect than calculated here.
More in situ observations of $\beta$ are clearly needed to better resolve these competing effects. One potential explanation
for these $\beta$ discrepancies is that the algorithm used to derive $\beta$ from remotely-sensed particulate backscatter sometimes
misses the largest particles in high-productivity areas such as the Southern Ocean (Kostadinov et al., 2009). In addition
to the mechanisms proposed in Lam and Bishop (2007), this may partly explain why $\frac{d\beta_{sat}}{dE_{n,sat}}$ is weakly positive in the
Subantarctic Zone (SAZ); particles may actually get larger with increasing export here, but because they are already
quite large, the satellite $\beta$ sensor/algorithm may not be able to capture the particles becoming any larger. The result
would be an underestimation of the negative (dampening) PSR feedback effect in this region.
Another caveat of our study is that very simple phytoplankton biology and growth dependent on only one
macronutrient was assumed. Furthermore, particle fragmentation—via processes such as zooplankton grazing,
microbial degradation, or ocean turbulence (e.g., Cavan et al., 2017; Briggs et al., 2020)—was not included in our
model, nor was particle aggregation—via processes such as Transparent Exopolymer Particles production (e.g.,
Passow, 2002; Mari et al., 2017) or fecal pellet generation (e.g., Steinberg et al., 2012; Turner, 2015 and references
therein). Despite the aforementioned shortcomings, the results presented here represent a reasonable first attempt to
quantify the strength of the PSR feedback effect on export changes within a global model. Future work should test the
PSR feedback effect in more complex models that better resolve phytoplankton/zooplankton biology, particle
dynamics, and/or circulation changes. These models could include particle aggregation-disaggregation with
prognostic sinking speeds (Gehlen et al., 2006), empirically-driven food-webs (Siegel et al., 2014), explicit
phytoplankton and grazers of different sizes (Buesseler & Boyd, 2009), and/or spatiotemporally-resolved circulation
changes that respond directly to atmospheric forcing.
Additionally, future work should analyze the downstream effects of the PSR feedback on climate-driven
projections of fisheries productivity, dissolved oxygen availability, and carbon sequestration in the deep ocean over
centennial to millennial timescales. A smaller-than-currently-projected decrease in surface nutrient supply and export
rates would be beneficial for maintaining fisheries, for example. On the other hand, predicted increases in deep ocean
carbon sequestration may be reduced by the PSR feedback. In particular, a decrease in circulation rates should enable
enhanced carbon sequestration, as nutrients and $CO_2$ collect in the deep ocean (Fig. 8), but the PSR feedback may
potentially moderate this increased sequestration effect by shoaling remineralization and forcing a shorter carbon
sequestration timescale. We focused solely on 100-year changes in POC export, to the exclusion of potential longer-
term changes in deep ocean carbon storage, because it is a critical energy source energy to the mesopelagic twilight
zone and therefore determines the productivity of heterotrophic communities, including commercial fisheries.
Furthermore, while changes in the biological pump may also drive changes in ocean carbon storage, these will manifest
over longer timescales than changes in export, and will likely be overwhelmed on short timescales by the effects of
anthropogenic $CO_2$ uptake and solubility-driven outgassing. A detailed exploration of changes in carbon storage is
therefore beyond the scope of the current paper, but could be a fruitful avenue for future work.
Other remineralization depth-related feedbacks induced by changes in temperature, oxygen, particle density,
and mineral ballasting (among others) not studied here may also be important for modulation of future changes in
carbon export and its downstream effects. Ensuring that the PSR and other remineralization feedbacks are adequately
represented in ESMs should be a priority of the modeling community to enable robust predictions of carbon export
fluxes in the future ocean.
**4 Summary**
In this study, we used remotely sensed data to show that sinking particle size is empirically correlated with
the rate of particulate organic carbon export out of the euphotic zone across the global ocean, such that larger particles
tend to dominate when export is high. This empirical relationship between particle size and export likely emerges due
to the dependence of both variables on surface nutrient supply. Indeed, nutrient limitation both curtails productivity
and selects for smaller phytoplankton that likely aggregate into smaller sinking particles (Litchman et al., 2007; Guidi
et al., 2007; 2008; 2009). A reduction in surface nutrient supply stemming from increased water column stratification
in a warming ocean (Bopp et al., 2013; Cabré et al., 2015a; Capotondi et al., 2012) should thus lead to a decrease in
global export production (Fig 4, green arrows; Fig. 5, slower circulation solid lines/bars) *and* sinking particle size (Fig
4a, red arrow; Fig. 4c-d, red line; Fig. 6b). Smaller particles in turn drive shallower nutrient remineralization and thus
faster resupply of those nutrients to the surface, dampening the initial circulation-driven change in export (Fig. 4a,
blue arrow; Fig. 4d, blue line; Fig. 5, slower circulation dashed lines/hatched bars; Fig. 6b; Fig. 7c-f; Fig. 8b-c).
Regardless of the mechanism linking export and particle size, implementing the empirical relationships between the
two in an idealized global biogeochemical model revealed the presence of a negative particle size-remineralization
feedback effect that moderates circulation-driven changes in export.
Many Earth System Models ignore the effects of nutrient supply on particle size and/or the effects of particle
size on remineralization depths. Within our model, including these effects reduces the magnitude of predicted 100-
year changes in global export production by ~14% (Fig. 5). This implies that in isolation of other mechanisms, ESMs
without the PSR feedback may be projecting 100-year climate-driven export decreases that are ~1.16 times too large.
Under a relatively extreme ESM-projected decrease of 18% by 2100 (Bopp et al., 2013), absolute global export would
be reduced by ~0.7-2.9 GtC/yr, assuming a present rate in the range of 4-16 GtC/yr (Boyd & Trull, 2007; DeVries &
Weber, 2017; Dunne et al., 2005, 2007; Falkowski et al., 1998; Henson et al., 2011; Laws et al., 2000; Siegel et al.,
2014; Yamanaka & Tajika, 1996); with the PSR feedback in effect, this predicted decrease would be reduced by ~14%
to ~0.6-2.3 GtC/yr.
The PSR feedback is strongest in low-latitude tropical and subtropical regions (moderating export changes
by up to 20%; Fig. 7g,h; Fig. 10), where ESMs also predict some of the largest future export decreases (Bopp et al.,
2013; Cabré et al., 2015a). Within these regions, primary and export production are highly nutrient-limited, such that
a given stratification-induced decrease in nutrient supply leads to relatively large decreases in export and sinking
particle size (Fig. 3), with correspondingly large effects on remineralization depth (Fig. 6) and surface nutrient
recycling. Because these regions exhibit the greatest projected decreases in export as well as the strongest PSR
feedback effects, spatial variations in projected export decrease may also be less pronounced than currently expected.
The PSR feedback operates on increases in surface nutrient supply as well. Under surface nutrient supply
increases, phytoplankton/particles grow larger and remineralization depths deepen, which sends more nutrients out of
the shallow subsurface and thereby moderates initial circulation-driven increases in export. This PSR feedback reduces
the magnitude of predicted 100-year changes in global export production by about 18% when circulation rates are
increased by 10% (Fig. 5, faster circulation dashed lines/hatched bars). In scenarios of global cooling (resulting in
water column destratification, enhanced mixing, and increased surface nutrient supply), centennial-scale projections
of export increase in models lacking the PSR feedback would therefore be >1.2 times too big, again with the largest
overestimates in the low to mid-latitude regions. The PSR feedback thus moderates export changes in response to any
physical perturbation to surface nutrient supply, whether driven by increasing or decreasing circulation rates.
**Code availability**
The MATLAB code required to make the figures generated here can be found at
https://doi.org/10.5281/zenodo.4117382.
**Data availability**
Data in the form of *.mat files required to the make the figures generated here can be found at
http://doi.org/10.5281/zenodo.3785724.
**Author contribution**
SL and CD designed the model experiments. SL developed the model code and performed the simulations. SL
prepared the manuscript with contributions from all co-authors.

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

**Figure Captions**

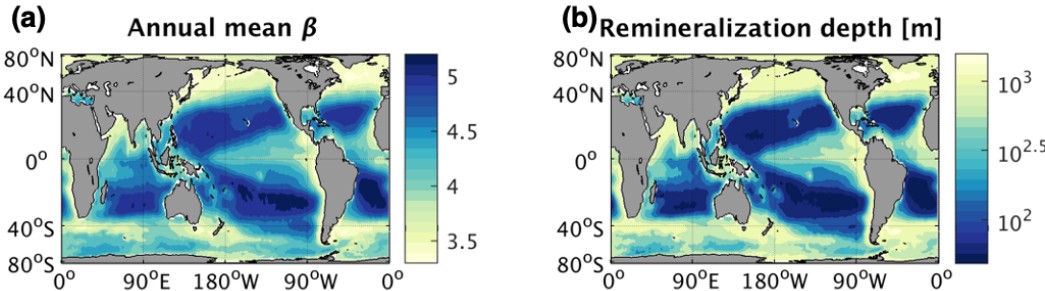


**Figure 1. Global maps of annual mean (a) particle size distribution slope (β) measured by remotely-sensed particulate**
**backscatter and reproduced from Kostadinov et al. (2009) and (b) remineralization depth, defined as the depth at which**
**particulate flux out of the euphotic zone is decreased by a factor of e assuming β in (a) at the surface, calculated using a**
**particle remineralization and sinking model (PRiSM, described in Section 2.1.1). Larger values of β are associated with**
**smaller particles, while smaller values of β are associated with larger particles.**

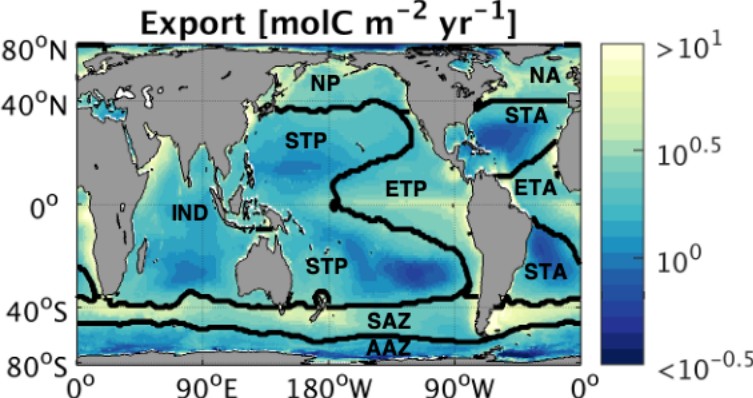

**Figure 2. Global map of regionally-weighted annual mean export, averaged over nine different export maps (detailed in**
**Section 2.2.2). Contours indicate biogeochemical regions used for weighting and spatial averaging.**

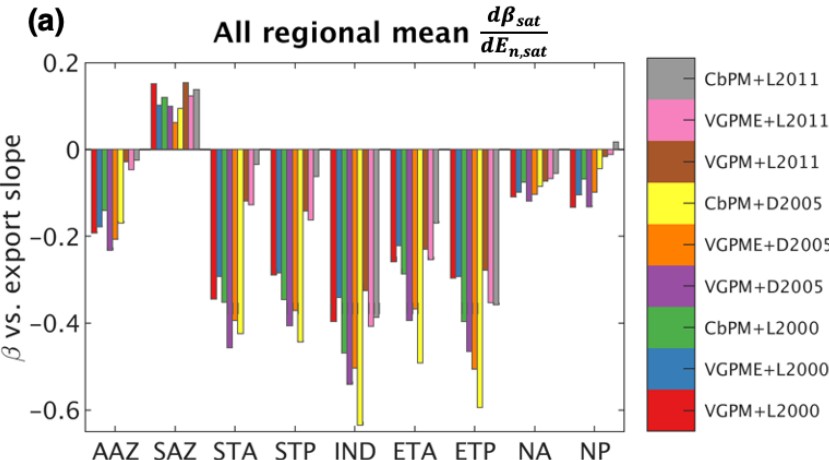

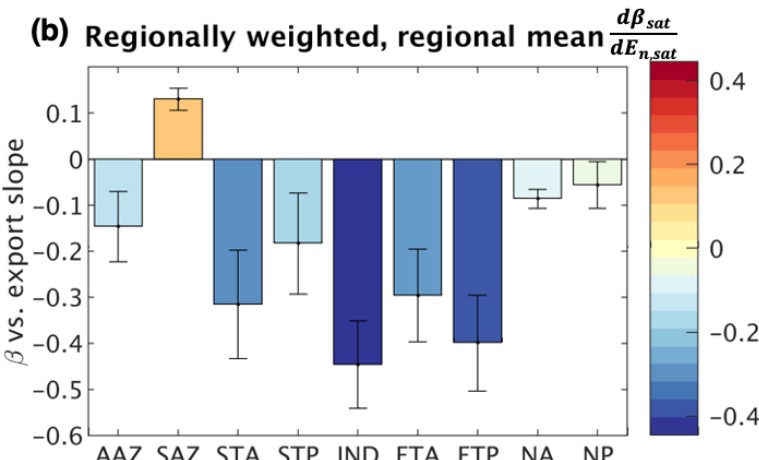

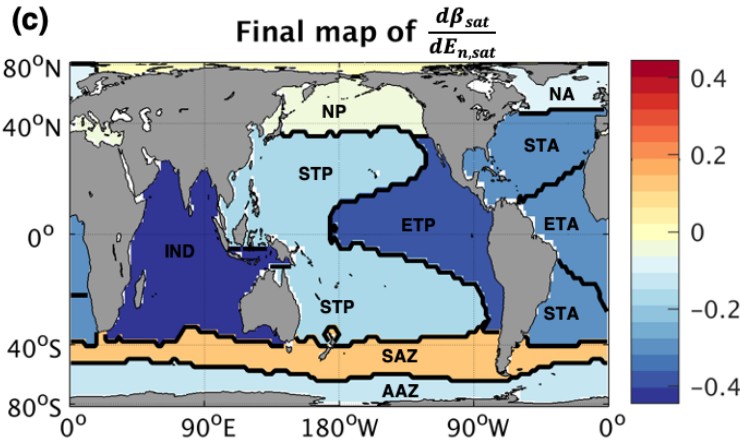


Figure 3. (a) All regional mean changes in particle size slope for a given change in time-mean normalized export, $\frac{d\beta_{sat}}{dE_{n,sat}}$
values (i.e., spatial averages of each map in Fig. S3 over regions shown in Fig. 2), colored by corresponding export map.
Colorbar labels indicate the NPP and e-ratio algorithms used to generate the given export map (see Section 2.2.2 for full
descriptions of the algorithms). NPP algorithm key: VGPM = the Vertically Generalized Production Model (VGPM)

**(Behrenfeld & Falkowski, 1997); VGPME = the Eppley-VGPM model (Carr et al., 2006); CbPM = the Carbon-based**
**Production Model (Behrenfeld et al., 2005). E-ratio algorithm key: L2000 = Laws et al. (2000); D2005 = Dunne et al. (2005);**
**L2011 = Laws et al. (2011). (b) Regionally-weighted mean $\frac{d\beta_{sat}}{dE_{n,sat}}$, averaged over the nine possibilities for each region shown**
**in Fig. 3a. Error bars represent one weighted standard deviation. (c) Global map of regionally variable $\frac{d\beta_{sat}}{dE_{n,sat}}$ used in model**
**runs with the PSR feedback on.**

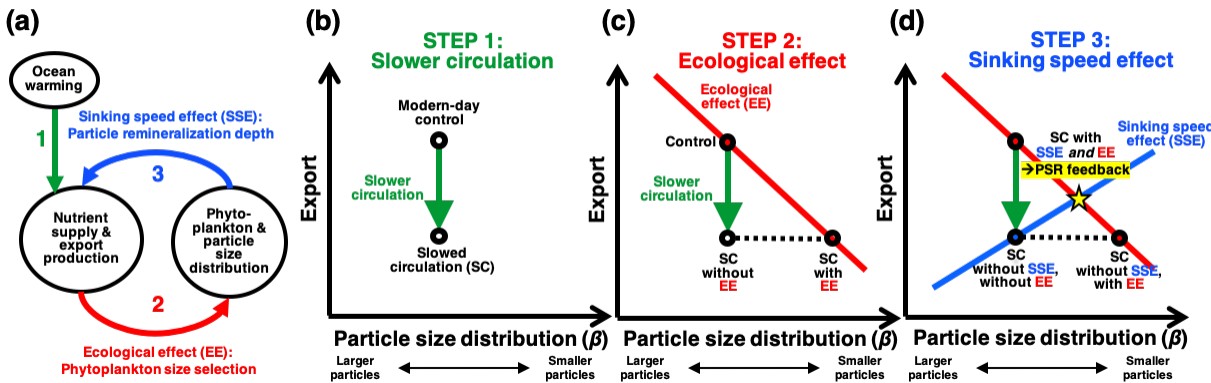


**Figure 4. (a) Schematic diagram of the particle size-remineralization (PSR) negative feedback on export production. A**
**change in circulation rates induced by climate change alters surface nutrient supply and subsequent export production**
**(green arrow). Changes in surface nutrient supply also drive changes in phytoplankton and resultant sinking particle sizes**
**(red arrow). Changes in sinking particle sizes in turn alter remineralization depth and consequently, surface nutrient supply**
**and export (blue arrow). (b) Schematic depicting decreased export production with decreases in circulation rates and**
**surface nutrient supply. (c) Schematic depicting a theoretical relationship between export and β, here termed the**
**phytoplankton size selection ecological effect (EE), in which smaller phytoplankton dominate in low-nutrient, low-export**
**conditions. (d) Schematic depicting all previous components of the PSR feedback, in addition to the crucial final component:**
**the particle remineralization depth sinking speed effect (SSE), in which smaller particles tend to get remineralized**
**shallower, leading to a greater recycled surface nutrient supply and therefore greater export.**

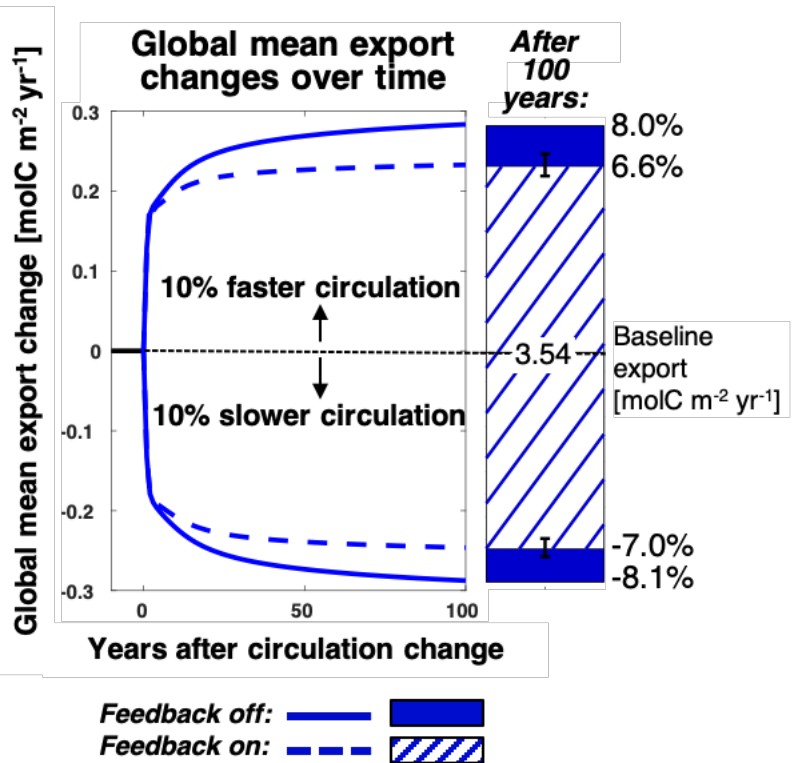


Figure 5. Changes in global mean export over time from baseline conditions (current-day circulation, ran to steady-state) after increasing or decreasing circulation rates by 10%. Dashed and solid lines represent runs with the PSR feedback turned off and on, respectively. The bars on the right show absolute changes in global mean export from the baseline case 100 years after changing circulation rates. Corresponding relative changes (calculated as absolute changes from the baseline over the baseline mean) are listed in black. Global mean export in the baseline case is listed on the zero line. Hatched and solid patterns represent runs with the PSR feedback turned off and on, respectively. The error bars represent export decreases generated when employing the upper and lower-bound $\frac{d\beta_{sat}}{dE_{n,sat}}$ maps described in Section 2.2.3.

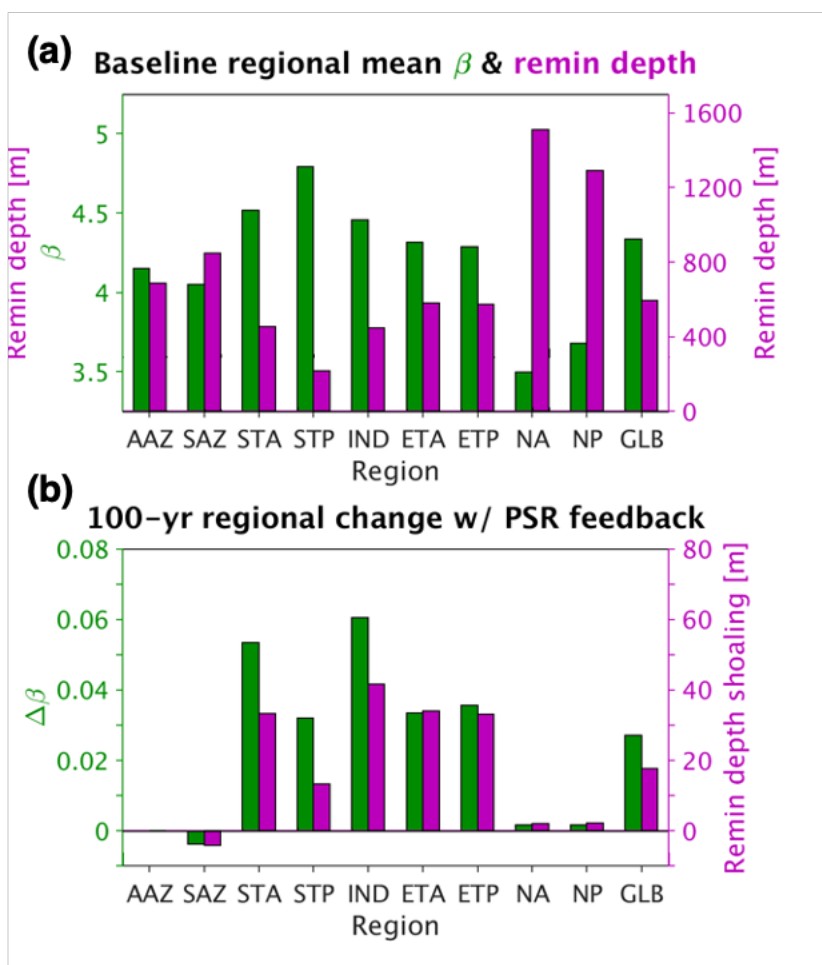

Figure 6. (a) Baseline (current-day circulation, ran to steady-state) regional mean β (shown in green) and e-folding remineralization depth (shown in purple). (b) Absolute change in regional mean β (shown in green) and absolute shoaling of regional mean remineralization depth (shown in purple) 100 years after decreasing circulation rates by 10% with the PSR feedback turned on.

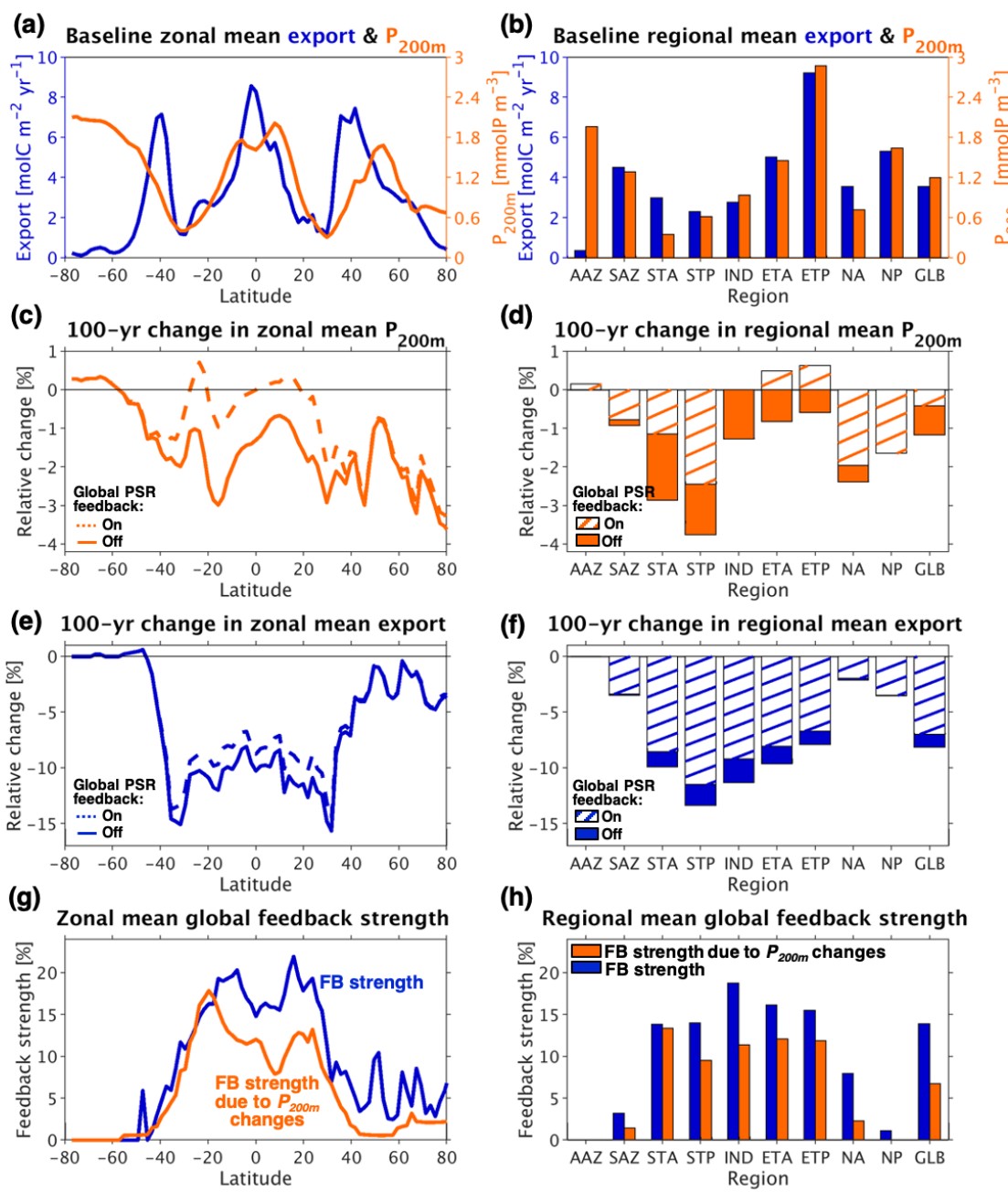

1073

Figure 7. (a) Baseline (current-day circulation, ran to steady-state) zonal mean export and shallow subsurface (200 m) phosphate concentration. (b) Same as (a), but with regional and global rather than zonal means. (c) Relative changes (calculated as absolute changes from the baseline over the baseline mean) in zonal mean phosphate concentration at 200 m depth 100 years after decreasing circulation rates by 10%. (d) Same as (c), but with regional and global means. (e) Relative changes in zonal mean export 100 years after decreasing circulation rates by 10%. (f) Same as (e), but with regional and global means. (g) Zonal mean PSR feedback strength, calculated as the difference in zonal mean export change from baseline between the feedback-off and on cases divided by the zonal mean export change in the feedback-off case alone (left-hand side of Eq. (4); shown in blue). Predicted zonal mean PSR feedback strength from changes in circulation and shallow subsurface phosphate concentration (right-hand side of Eq. (4); shown in orange). (h) Same as (g), but with regional and global means.

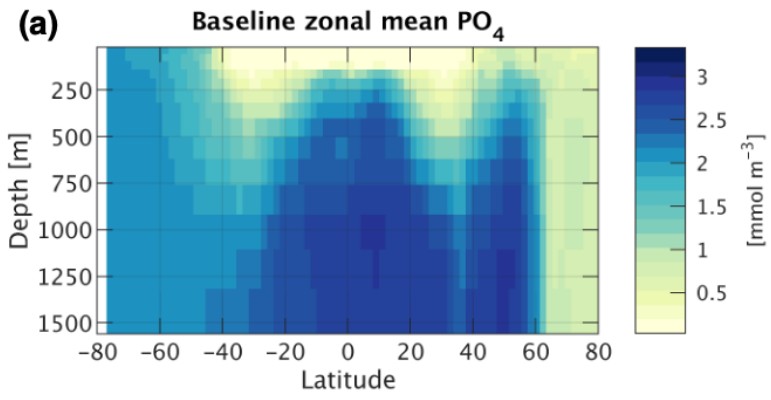

**(a)** Baseline zonal mean PO$_4$

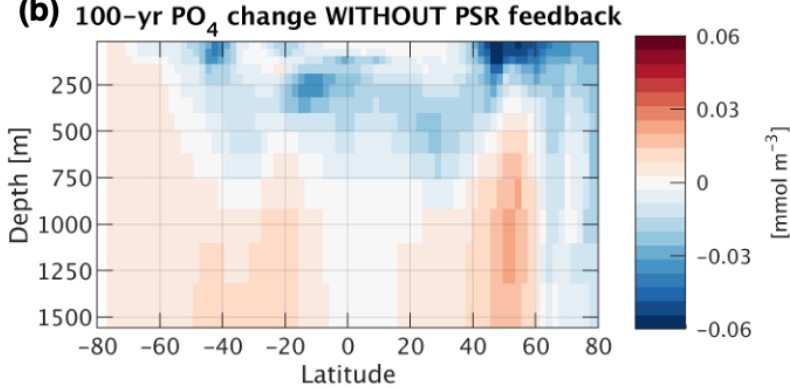

**(b)** 100-yr PO$_4$ change WITHOUT PSR feedback

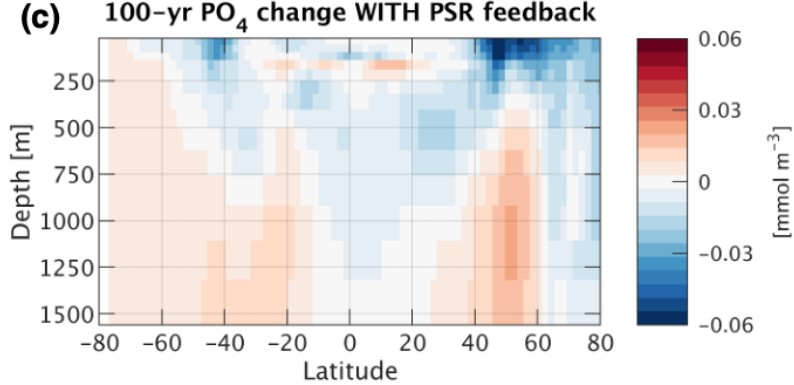

**(c)** 100-yr PO$_4$ change WITH PSR feedback

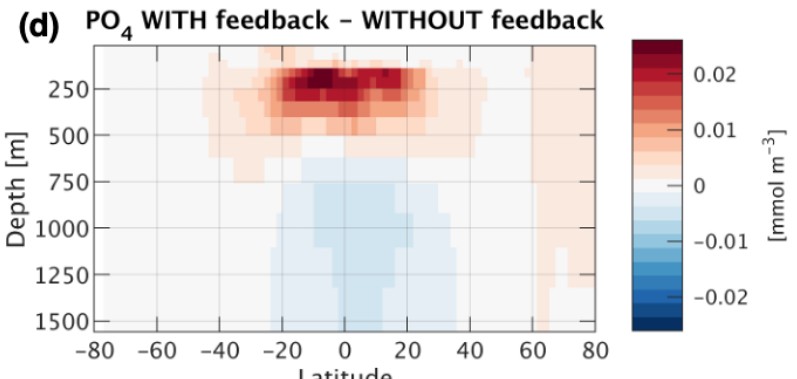

**(d)** PO$_4$ WITH feedback – WITHOUT feedback

1084

Figure 8. (a) Baseline (current-day circulation, ran to steady-state) zonal mean phosphate concentration. (b) Absolute change in zonal mean phosphate concentration 100 years after decreasing circulation rates by 10% with the PSR feedback turned off. (c) Same as (b), but with the PSR feedback turned on. (d) Difference in zonal mean phosphate concentration between PSR feedback-on and -off runs (i.e., (b) minus (c)).

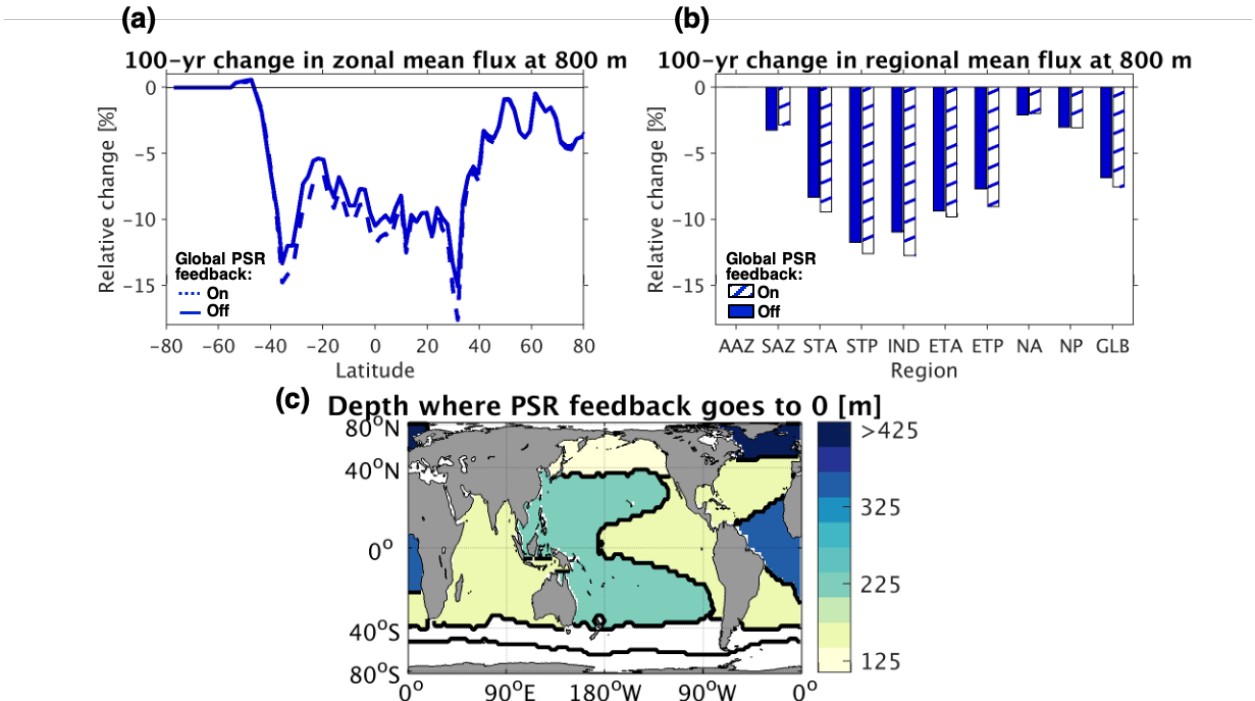

Figure 9. (a) Relative changes in zonal mean particulate organic carbon (POC) flux at 800 m depth (725 m below the bottom of the euphotic zone) 100 years after decreasing circulation rates by 10%. (b) Same as (a), but with regional and global means. (c) Map of regional mean depths below the surface at which the PSR feedback effect flips from negative to positive. In the North Atlantic, the regional mean PSR feedback effect flips from negative to positive at ~2100 m depth, which is off the color scale.

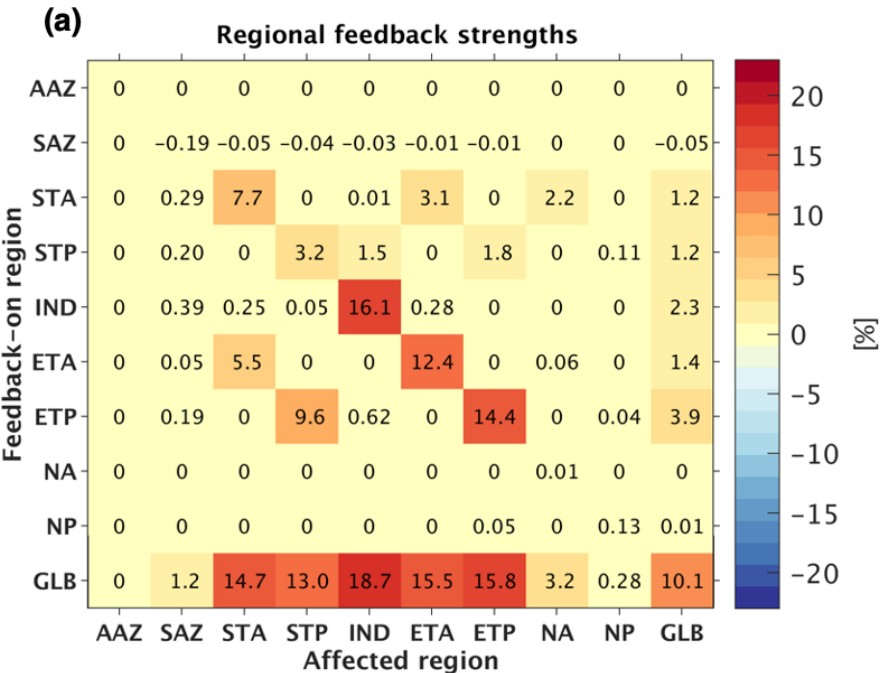

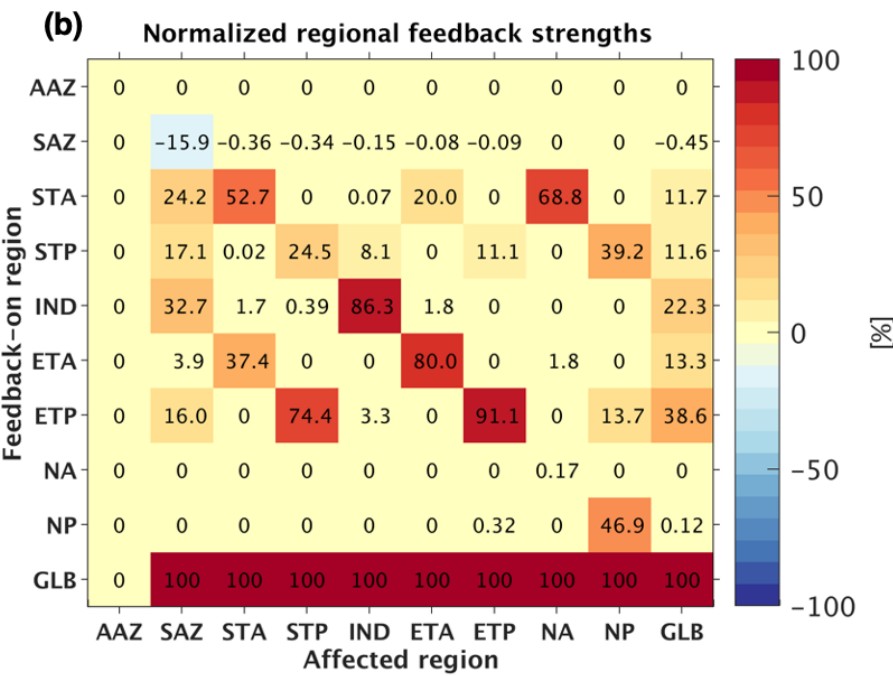

**Figure 10. (a) Regional mean feedback strength due to the PSR negative feedback effect within each individual region. The y-axis denotes the single region (or the entire ocean in the case of "GLB" or "global") within which the PSR feedback was turned on, while the x-axis denotes the region affected. A feedback strength above 0 indicates a negative PSR feedback effect (that is, a reduction in export change when the PSR feedback is applied). (b) Percent contribution of each individual region to each region's total PSR feedback strength, computed as the regionally-derived feedback strength within an affected region divided by the globally-derived feedback strength in the same affected region (i.e., each given grid cell in (a) is divided by the corresponding column's bottom-most grid cell).**