# Peer review of "Variable particle size distributions reduce the sensitivity of global"

_Biogeosciences, 2020_

## Referee Comment (RC1) · Anonymous Referee #1 · 15 Jun 2020

Review of Leung et al. " Variable phytoplankton size distributions reduce the sensitivity of global export flux to climate change"

This manuscript examines the influence of incorporating information on phytoplankton size into a biogeochemical model alters predicted carbon export in response to climate change. The overall result that the influence of climate change is damped by incorporating size information is intriguing and worthy of publication. Generally, the manuscript is clear and well-written., and represents a useful addition to the literature.

Abstract: Line 20: the altered export values are reported without any reference to what the baseline simulation is, i.e. report what is the predicted decline in export without

the size considerations, and what is it with the size included Line 22: "more robust predictions" – How do you know these predictions are more robust than the baseline? There is no model validation presented (which I don't mind for this manuscript, but "more robust" can't be asserted in this case).

Introduction: Line 41: how do the models cited here handle size or sinking speed (if they do at all)? i.e. do these models also already include a size-based parameterisation which means that the predictions are ∼ equivalent to yours? Line 86-87: a fundamental assumption in this study is that small phytoplankton = small particles. A critical assessment from observational data of whether this is true, and when this assumption might break down, should be included. For example, how might TEP production or fragmentation affect the size structure of particles? Line 91-92: references needed for the assertion that small picophytoplankton = small particles, and same for large. Line 97: there's a dawning realisation that Stokes law rarely holds for marine particles e.g. https://aslopubs.onlinelibrary.wiley.com/doi/abs/10.1002/lno.11388 This should be acknowledged here. Line 127: but wouldn't a shift to smaller particles also result in less C sequestration at depth as the C just goes round and round in the upper mesopelagic being readily recycled and re-entrained to surface? Also, Figure 2d – the caption acknowledges that smaller particles leads to greater recycled nutrient supply. This wouldn't increase C sequestration (or $CO_2$ drawdown) as that depends on the re-supply of preformed nutrients which isn't affected by the size considerations used here, at least on the timescales considered.

Methods: Line 217: just curious to know why Laws and Dunne estimates of export ratio, rather than others such as Henson et al. 2012 or Siegel et al. 2014 were used Line 227: I couldn't find where the in situ observations mentioned here had come from Line 238-240: is temporal autocorrelation accounted for here? I guess the seasonal cycle in beta and En are similar which will affect the linear regression. Also, are beta and En independent? How are pixels/regions with non statistically significant regressions dealt with? Line 266: in reality I suspect dbetaobs/dEnobs could vary seasonally. Might be

worth a caveat on that point in the discussion? Actually, I suspect that some of the strange behaviour in the SAZ might be due to a seasonal effect or a time lag between changes in phytoplankton size and particle export. It would be helpful to the reader to include some example annual time series of a region showing the PP, export and beta to illustrate how they interact. It would also inform on potential time lags between PP, export and beta.

Results and Discussion: A point that should be acknowledged somewhere is that the results presented here are of course still dependent on the details of the model parameterisation and choices. Line 301-303: references needed here Line 322: isn't this 21% rather than 18%? I found the use of the word "visually" here and on line 307 confusing. It made me think that you had estimated the values by eye rather than calculating them. I suggest just dropping 'visually'. Line 346: some C:P ratio must be used here too? Couldn't find where that was mentioned. Does this formulation also assume that all nutrients supplied are regenerated? I think it assumes that all nutrients supplied are turned into PP (which is fine in nutrient limited regions), and then are all exported i.e. and e-ratio of 1? Line 376-378: I'm not sure this "visually" statement helps the reader's understanding here Line 415-426: specify the direction of +ve/-ve changes in the caption. At the moment it's a bit confusing as +ve indicates a reduction

Figure 1a: specify in the caption that higher beta values = smaller plankton (or mark with arrows on the colour bar) Figure 6a: rather than having the right y-axis in remin depth x 100, just write out the numbers in full – it's clearer Figure S1: specify in the caption or legend that higher beta values = smaller plankton Figure S2: add the key to the PP and e-ratio model abbreviations into the caption here Table S1: Define the parameter names in this table.

---

## Referee Comment (RC2) · Anonymous Referee #2 · 17 Jun 2020

The authors apply a global biogeochemical model to examine the effect of variable particle (phytoplankton) size distribution on surface and subsurface nutrients, and their mutual feedbacks when nutrients are supplied under different physical forcings. The feedback effect of the (nutrient-dependent) size distribution and subsequent particle sinking and remineralisation dampens the model response to changes in physics. I find this manuscript generally well written. The authors do a great job in explaining the mechanisms involved. In general, the experimental design to disentangle the effects of circulation, ecology and sinking is clear and well justified. Thus, the manuscript provides valuable new insights into a potential negative feedback mechanism in global biogeochemical models. However, I have a few points that I think could be improved

with regard to model description and its critical discussion.

(1) Model description: I recommend to describe the biogeochemical model, particle sinking and remineralisation in detail (including equations), and also explain its basic assumptions. As far as I understand, the model assumes a power law size distribution of particles at the surface; particles then sink depending on their size, and remineralise with a size independent rate. Therefore, the particle size distribution changes with depth, favouring large particles as depth increases, similar to the 1D approach presented by Kriest and Oschlies (2008). (In fact, there seem to be only small differences between both models, in terms of formulation and results.) Both approaches make quite strong implicit assumptions about constant individual particle properties, which do not change with time or depth. In particular, the models neglect any processes besides sinking and remineralisation that might affect the particle size distribution below the euphotic zone, such as particle breakup, reworking by zooplankton (e.g., flux feeding, formation of fecal pellets), particles becoming more or less porous because of bacterial degradation, etc.. Of course, one cannot address all details and complications at once especially in global models; but describing the current implicit model assumptions in detail would help the reader to understand how the model works, and what its limitations and merits might be.

(2) Model description: The description of the model and its general setup is somehow unclear about how the phytoplankton size distribution might be related to larger particles, which likely contribute most to mesopelagic and deep particle flux. For example, the work by Kostadinov et al (2009), from which the observed size distribution at the surface is taken, is based on phytoplankton, i.e. extends only to a size of ca. 50 um. However, the present model applies a size range of 20-2000 um (Table S1). Moreover, the model parameters sometimes seem to relate to phytoplankton properties (e.g., the exponent of eta=1.17 relating cell diameter to sinking speed is based on phytoplankton data by Smayda, 1971), whereas other relate more to porous aggregates (e.g., the exponent relating particle mass to size of zeta=1.62; see also Kriest,

2002 http://dx.doi.org/10.1016/S0967-0637(02)00127-9, and citations therein). Again, here it would be useful to present and discuss these basic model assumptions. This is done partly on page 3, yet I think this subsection could be improved (see below, my comments Lines 80ff, 82ff, 95). In summary, I would suggest to more clearly distinguish between phytoplankton and particle size distribution, and to address potential connections between these more comprehensively.

(3) Experimental setup: To me it is not clear how the circulation was reduced (e.g., Lines 253-254 "To simulate increased water column stratification and reduced vertical exchange due to warming, we uniformly and instantaneously reduce circulation and diffusion rates by 10% throughout the ocean.") - I would appreciate a more in depth explanation.

(5) Discussion: The model shows a large response and differences between the two setups (with or without PSR) in the equatorial upwelling regions. However, especially models of coarser resolution tend to suffer from an insufficient representation of the equatorial current system, with possible consequences for the representation of nutrients and/or oxygen (e.g., Dietze and Loeptien, 2013, https://doi.org/10.1002/gbc.20029; Duteil et al., 2014, https://doi.org/10.1029/2011GL046877). I would suggest to add some discussion on these potential effects.

(4) Discussion: Section 3 is named "Results and discussion", yet it almost entirely presents the results. In contrast, Section 4 is named "Conclusions", but partly discusses the results before the background of other works, is quite long and partly repetitive. I would suggest to rename section 3 to "Results", add a "Discussion" section, that extends a bit on the comparison of results obtained here with other model studies and also includes a critical discussion of model processes and properties. The "Conclusions" section could then be shortened and more concise.

Specific comments:

- Lines 35-37: "Where sinking POC fluxes are particularly high, enhanced bacterial breakdown of particles can deplete available oxygen and create hypoxic or even suboxic conditions [...] " - there are many places in the ocean where sinking POC fluxes are high; another necessary condition for the development of OMZs is that supply of oxygen by physical transport is low.

- Line 66: Note that there are further global ocean models that address spatial and temporal variation of the size distribution (of marine aggregates) and sinking speed, e.g., Schwinger et al. (2016, www.geosci-model-dev.net/9/2589/2016/) and Niemeyer et al. (2019, https://doi.org/10.5194/bg-16-3095-2019). On a local (1D) scale, even very complex models of particle transformations have been developed (Jokulsdottir and Archer, 2016, www.geosci-model-dev.net/9/1455/2016/).

- Lines 80ff: "Large particles tend to exist in the ocean where larger microphytoplankton (>20 um in diameter) are dominant, while relatively small particles tend to exist where smaller picophytoplankton (<2 um in diameter) are dominant (Guidi et al., 2007; Guidi et al., 2008; Guidi et al., 2009). [...]" - The observations by Guidi et al. (2007, 2008), are based on UVP data of large particles (aggregates, fecal pellets, ...), of a size of at least 250 um. Therefore, I don't think that these observations can be used to justify the assumptions about the phytoplankton size distributions made in this paper.

- Lines 82ff: "The presence of large phytoplankton leads to the generation of larger particles perhaps because large phytoplankton are more likely to form aggregates and be transformed into large fecal pellets by large zooplankton, whereas small phytoplankton are more likely to be degraded by bacteria and consumed by smaller zooplankton (Bopp et al., 2005; Guidi et al., 2007; Guidi et al., 2009; Michaels and Silver, 1988). The exact mechanisms governing the processes by which smaller and larger phytoplankton become smaller and larger particles are not clearly known, however, and is an active area of research." - The global model study by Bopp et al. does not address aggregates; moreover, as a model study it is based on a priori assumptions, and does not provide insight into real in situ mechanisms. As noted above, the study by Guidi

et al. (2007) addresses the UVP size range and the study by Michaels et al. is also a (food web) model. While I tend to agree with the idea that large phytoplankton triggers large sinking particles, I would appreciate a more convincing reasoning why one can extend the phytoplankton size range up to particles 2 millimeters in diameter.

- Line 95: "Past work has also firmly established a strong positive relationship between particle size and sinking speed in the ocean (Alldredge and Gotschalk, 1988; Smayda, 1971) [...]" - The relationship between diameter and sinking speed in Alldredge and Gotschalk (1988) is $w=50\ d^{0.26}$, and shows considerable scatter. I would not call this a strong relationship. This weak relationship is possibly because of the fractal and variable nature of aggregates - indeed, single cells show a higher exponent (Smayda, 1971). Again, here I would suggest to more clearly distinguish between aggregates and single phytoplankton cells.

- Line 100: "by a factor of e" - e is 2.718, do you really mean a factor of e?

- Line 157: "physical relationships between particle size, mass, and sinking velocity" - I don't think that the relationship between particle size, mass and sinking velocity of organic particles is a purely physical one; at least the relationships by Smayda (1971) and Alldredge and Gotschalk (1988) are empirical. I suggest to skip "physical".

- Line 184-185: "time-mean export" - mean over what time? A year?

- Line 483-484: "This implies that global models without the PSR feedback may be overestimating 100-year climate-driven export decreases by $\sim$1.16 times." - What is meant with 1.16 times?

---

## Referee Comment (RC3) · Anonymous Referee #3 · 7 Jul 2020

Leung et al. present an interesting study in which they thoroughly analyse the "particle size remineralization" (PSR) feedback in a simple 3D biogeochemical model. The PSR feedback is described by decreased circulation leading to increased nutrient limitation at surface, which in turn leads to plankton communities with smaller sizes producing smaller sinking particles, leading to a slower sinking speed and hence remineralization at shallower depths. This chain of mechanisms increases nutrient concentration at these shallower depths, dampening the decreased nutrient supply to the surface. Their results show that the PSR feedback dampens projected decrease in export production by 14% globally and also details the dampening in different regions. In particular the detailed regional analysis is very informative, suggesting that changes in particle sinking speed in the Southern Ocean have only a small impact on carbon export. Overall I find this an important and interesting study, well written and clearly presented and I recommend a speedy publication after the following comments have been addressed.

Main comments:

- I like that the PSR feedback has been thoroughly analyzed, but it should be discussed or mentioned earlier that there are several other processes potentially affecting the nutrient supply to the surface (varying stoichiometry, temperature dependence of remineralization, ballasting, aggregation/fragmentation etc) that are ignored in this study. There is a good discussion of the caveats at the very end in the conclusions, but I think it would help to mention early on that this paper analyses the PSR feedback in isolation of the other feedbacks.

- The proposed PSR feedback has been analysed for carbon export through 100m depth, however I would expect that there is a decrease in POC flux at deeper layers in the ocean (as the PSR feedback mainly acts in the first few 100m? My interpretation of course). For instance for the question how much carbon is sequestered from the atmosphere or how much food supply is available for mesopelagic organisms, the carbon flux below 100m depth might be of higher relevance. It would be very interesting to see how much the carbon flux at deeper layers is dampened by the PSR feedback.

- There are models participating in CMIP5/CMIP6 that parameterize two different particle size classes with different sinking speeds (I have written a bit more in the line-by-line comments). Is the full particle size spectrum really needed for the PSR feedback, or is it also possible to use just two particle size classes?

- I think some aspects of the model should be described in more detail, in particular the PRiSM component used to estimate particle flux (see line-by-line comments)

Line-by-line comments:

L 9: Please write something like "This decline is mainly caused by...", as there are

other mechanisms projected as well, for instances stronger grazing pressure due to higher temperatures

L 11: But there are some Earth System Models that simulate changes in remineralization depth, due to e.g. changes in phytoplankton community composition? For instance CESM-BEC, IPSL-PISCES and GFDL models all simulate changes in formation and sinking of particles under climate change. It is true that most models don't have a dynamic particle size spectrum though. (I am also wondering, do the Earth System Models that already have particles with different size classes also predict lower export decrease? But this is probably impossible to disentangle from other effects such as a temperature-dependent remineralization or varying stoichiometry etc. )

L 45/46 Maybe cite Laufkotter et al. 2015 for drivers of future declines in primary production as well

Introduction: I miss a discussion or mentioning of other drivers of sinking speed/particle remineralization, such as particle density/porosity and ballasting, also fragmentation and aggregation which potentially change the particle size over time

L 67/68 "parameters and processes in most previous models are not constrained by observations of particle size distributions" - Yes! Nice!

L74/75 Are there more recent references for the particle size distribution?

L 99 Just a small comment: I think the "shallowest" can be removed?

L 117 I think the decreased nutrient supply causes the decrease in phytoplankton/particle size, not the decrease in export?

The introduction reads slightly redundant to me and could be streamlined a bit, particularly part 1.2 (PSR feedback). However, I acknowledge that other reviewers seem to particularly like it and find it "very clear", so maybe for readers with little background in particle export it is better the way it is.

L 164 Does the plankton not become small plankton, or does the small plankton not produce small particles? Oh I see, plankton community isn't explicitly represented, correct?

I don't understand where Prism has the microbial respiration rates from, and whether they increase due to temperature in a warmer ocean? Is there an oxygen dependence? Is there aggregation/fragmentation? Can the sinking speed change over depth? Given that these questions are essential to this study, please explain Prism in more detail.

L173/174 wouldn't it make sense to also increase the production of DOC when changing the beta of the particle size spectrum?

Fig 3: That's probably only my personal preference, but I generally prefer mapping data using a log scale, instead of plotting log(data) and using a linear scale.

L 233/234 The NPP estimates you are using are significantly biased in the Southern Ocean. This should probably be discussed here, or you could use an NPP estimate that has been specifically created for the Southern Ocean (for instance Johnsson et al. 2013).

L254 How sensitive are your results to this simplified representation of future ocean circulation?

L 282 In light of these "counterintuitive" relationships I think it would be really interesting to use an NPP algorithm that's been developed for the Southern Ocean, as mentioned above (the result might stay the same of course, and I like how it is discussed/explained with the Lam&Bishop findings)

L 346: Is this a conclusion from the correlation between P200m and E or something that is actively diagnosed in the model?

L 355 ff The considerations up to this point hold for the nutrient-limited low and mid latitudes. But here you discuss a global uniform decrease of circulation rates. I found the jump from regional to global a bit confusing. Wait, I see now that in line 349, eq 3

is meant to hold at any given location, including regions in which eq. 2 does not hold, yes? Is that mathematically sound??

L 358 "This decrease in P200m is .." - Is this your interpretation or has this been diagnosed in the model?

L 404 - Is eq. 2 a close approximation in the low latitudes or globally? Is it possible to show on a map how well Eq 2 holds regionally?

L 484 is this a typo "1.16 times"? Otherwise, I don't understand, why 1.16 times? 1 - 16%? But I wouldn't understand that, either. Sorry if I am missing something obvious.

L 485 I believe the Pisces models used in the Bopp et al. study implements two different particle sizes, so it's possible that the PSR feedback is at least partly included? Please check

L 507/508 "PSR Âăfeedback strength remains relatively constant whether circulation rates are increased/decreased by 10% or 50% " - That is very surprising to me. Why would that be? Are you reaching a maximum/minimum particle size distribution such that a further decrease in nutrient supply does not affect the size distribution anymore? Please discuss this a bit more. I also think this sensitivity test should be mentioned in the results already.

L 528 I think the study by Briggs et al. 2020 is good recent reference for fragmentation

L 545 The effect of temperature increases and oxygen decreases on future carbon export has been analyzed in Laufkotter et al 2017, maybe good to cite here

Conclusion: If I understand everything correctly, your results suggest that changes in particle size/sinking speed in the Southern Ocean have only a small impact on carbon export regionally and globally. This could be interpreted as a justification for modelers to give implementation of particle size a low priority in this region - would you agree?

Anyway. I hope this review helps, and my apologies that it took me so long!

[Figure]

Laufkötter, C., Vogt, M., Gruber, N., Aita-Noguchi, M., Aumont, O., Bopp, L., Buitenhuis, E., Doney, S. C., Dunne, J., Hashioka, T., Hauck, J., Hirata, T., John, J., Le Quéré, C., Lima, I. D., Nakano, H., Seferian, R., Totterdell, I., Vichi, M., and Völker, C.: Drivers and uncertainties of future global marine primary production in marine ecosystem models, Biogeosciences, 12, 6955–6984, https://doi.org/10.5194/bg-12-6955-2015, 2015.

Briggs et al., Science 367, 791–793 (2020), Major role of particle fragmentation in regulating biological sequestration of CO2 by the oceans

Johnson, R., P. G. Strutton, S. W. Wright, A. McMinn, and K. M. Meiners (2013), Three improved Satellite Chlorophyll algorithms for the Southern Ocean, J. Geophys. Res. Oceans, 118, 3694–3703

Laufkötter, C., J. G. John, C. A. Stock, and J. P. Dunne (2017), Temperature and oxygen dependence of the remineralization of organic matter, Global Biogeochem. Cycles, 31, 1038–1050,

---

## Author Comment (AC1) · 16 Aug 2020

For this journal's review process, authors are expected to post a response to all reviewer comments before revising the actual manuscript. Based on these author responses, the editor either invites the authors to submit a revised manuscript or directly rejects the manuscript. We therefore do not yet include a revised manuscript along with answers to the following comments. See more details on the process here: https://www.biogeosciences.net/peer_review/interactive_review_process.html.

In the paragraphs below, all reviewer comments will be italicized, while author responses will be in normal font.

[Figure]

*Leung et al. present an interesting study in which they thoroughly analyse the "particle size remineralization" (PSR) feedback in a simple 3D biogeochemical model. The PSR feedback is described by decreased circulation leading to increased nutrient limitation at surface, which in turn leads to plankton communities with smaller sizes producing smaller sinking particles, leading to a slower sinking speed and hence remineralization at shallower depths. This chain of mechanisms increases nutrient concentration at these shallower depths, dampening the decreased nutrient supply to the surface. Their results show that the PSR feedback dampens projected decrease in export production by 14% globally and also details the dampening in different regions. In particular the detailed regional analysis is very informative, suggesting that changes in particle sinking speed in the Southern Ocean have only a small impact on carbon export. Overall I find this an important and interesting study, well written and clearly presented and I recommend a speedy publication after the following comments have been addressed.*

We thank the reviewer for their constructive comments and appreciate the positive response to our manuscript.

*Main comments:*

*- I like that the PSR feedback has been thoroughly analyzed, but it should be discussed or mentioned earlier that there are several other processes potentially affecting the nutrient supply to the surface (varying stoichiometry, temperature dependence of remineralization, ballasting, aggregation/fragmentation etc) that are ignored in this study. There is a good discussion of the caveats at the very end in the conclusions, but I think it would help to mention early on that this paper analyses the PSR feedback in isolation of the other feedbacks.*

The reviewer makes a good point about structure. We will move this discussion to the introduction and then mention it once more succinctly in the conclusion.

*- The proposed PSR feedback has been analysed for carbon export through 100m depth, however I would expect that there is a decrease in POC flux at deeper layers in*

[Figure]

*the ocean (as the PSR feedback mainly acts in the first few 100m? My interpretation of course). For instance for the question how much carbon is sequestered from the atmosphere or how much food supply is available for mesopelagic organisms, the carbon flux below 100m depth might be of higher relevance. It would be very interesting to see how much the carbon flux at deeper layers is dampened by the PSR feedback.*

This is a great point, and something we discussed at length when we were writing this manuscript. We elected to constrain the scope of this manuscript to exploring export, rather than carbon sequestration. An analysis of sequestration is a clear direction for future work, either by our group or by another. We will expand our discussion of what this PSR feedback might mean for carbon sequestration in the conclusion.

The reason why we have focused on export rather than carbon storage/sequestration is because carbon storage responses to changes in biological pump strength will manifest on longer timescales than changes in export, and will likely be small compared to changes driven by anthropogenic $CO_2$ uptake or solubility. Furthermore, export is important in its own right, and is more directly related to the changes in nutrient cycling explored here.

The reason why we focus on export out of the bottom of the euphotic zone and not carbon flux at some deeper depth is because export, by definition, is a measure of the *total* food supply to subsurface heterotrophic communities, and so the PSR feedback does buffer the productivity of mesopelagic communities as a whole, by damping changes in export. It's true that the feedback has the opposite effect on deeper fluxes (say at 500m), and this results in a vertical re-organization of the food supply, with more food available in the upper mesopelagic and less in the lower mesopelagic. But the net effect is still a buffering of the *total* food supply from the surface ocean.

We will add substantial explanation of this vertical reorganization in our revised manuscript with the aid of a new figure. This figure will contain a repeat of Fig. 7e-f, but show change in 500 m flux (Fig. 1 shown below), rather than export, with the feedback on/off. One noteworthy finding is that it is at about 150 m below the bottom of the euphotic zone that the PSR feedback effect on POC flux changes sign across most ocean regions, so we will also include a global map showing the exact depths at which the PSR feedback turns from negative to positive across all of the ocean regions.

Though we want to keep our main focus on export, which determines the *total* food source to the mesopelagic, we thank the reviewer for pointing out this very cool vertical reorganization! We are excited to add this discussion into the paper and think it will make it quite a bit more interesting.

*- There are models participating in CMIP5/CMIP6 that parameterize two different particle size classes with different sinking speeds (I have written a bit more in the line-by-line comments). Is the full particle size spectrum really needed for the PSR feedback, or is it also possible to use just two particle size classes?*

This is a great question, but not one we can definitively answer in the current study. We believe that a model like ours that resolves a continuous size spectrum is most likely to accurately capture the feedback, especially given that the export/particle-size relationship can be directly constrained with remotely-sensed spectral observations. Clearly, models with 2 different particle size classes, with different sinking speeds, could qualitatively reproduce the same feedback, but it is not clear whether the magnitude of the feedback would be accurately captured. Furthermore, in the CMIP5 models, the relative proportion of large and small particles is determined by internal model dynamics, and not prescribed empirically as in our model. Their ability to reproduce the feedback would therefore depend on the degree to which the observed relationship between export and particle-size emerges from the ecosystem model. We hope that in future work, our study can provide a baseline against which ESMs can assess their ability to reproduce particle size feedbacks, and we will add a note on this to the discussion section of the revised manuscript.

*- I think some aspects of the model should be described in more detail, in particular*

*the PRiSM component used to estimate particle flux (see line-by-line comments)*

We thank the reviewer for pointing out that the model description falls short. The mathematical description of the PRiSM model has been fully laid out in previous references and cannot be repeated in full here. However, we agree that enough information needs to be provided to allow the reader to understand how the model "works", without extensive cross-referencing to previous papers. We will revise the methods section to expand the description of the PRiSM model. Specifically, at line 157 we will insert the following:

"In the PRiSM model (DeVries et al., 2014), particles are produced in the surface euphotic zone (<75m) following a power-law size spectrum, in which log10(particle number density) declines linearly with log10(diameter), and the relative abundance of large and small particles is controlled by the slope of the spectrum on a log-log scale ($\beta$). The simulated particle size spectrum then evolves through the water column due to remineralization and size-dependent sinking, which are each parameterized based on empirically derived relationships and observed particle properties. Remineralization is represented by first-order mass loss from each particle, such that particles shrink and sink more slowly with depth, resulting in attenuation of the particle flux. Because smaller, slower-sinking particles reside for longer within any given depth interval, and therefore have more time to remineralize, they are preferentially lost from the particle population over depth, resulting in a flattening of the size spectrum (reduced $\beta$) and thus increased average sinking speeds at deeper depths. A constant rate of microbial respiration is used, optimized to fit global in situ phosphate distributions (DeVries et al., 2014). There are therefore no temporal changes in bacterial respiration due to warming, for example, which allows us to isolate changes in export that stem from the PSR feedback alone. While PRiSM has recently been expanded to include temperature and oxygen effects on bacterial respiration and remineralization (Cram et al., 2018), as well as to represent particle disaggregation (Bianchi Weber et al., 2018), here we use the original version described in DeVries et al. (2014), which can be solved analytically

and has previously been optimized to best fit the global phosphate distribution. The recent expanded versions of PRiSM must be solved numerically, which is less computationally efficient and therefore not suitable for incorporation into our global three dimensional simulations."

Added reference:

Bianchi, D., Weber, T.S., Kiko, R. and Deutsch, C., 2018. Global niche of marine anaerobic metabolisms expanded by particle microenvironments. Nature Geoscience, 11(4), pp.263-268.

*Line-by-line comments:*

*L 9: Please write something like "This decline is mainly caused by. . .", as there are other mechanisms projected as well, for instances stronger grazing pressure due to higher temperatures*

We will change the statement from:

"This decline is caused by increased stratification..."

To:

"This decline **is mainly** caused by increased stratification..."

*L 11: But there are some Earth System Models that simulate changes in remineralization depth, due to e.g. changes in phytoplankton community composition? For instance CESM-BEC, IPSL-PISCES and GFDL models all simulate changes in formation and sinking of particles under climate change. It is true that most models don't have a dynamic particle size spectrum though. (I am also wondering, do the Earth System Models that already have particles with different size classes also predict lower export decrease? But this is probably impossible to disentangle from other effects such as a temperature-dependent remineralization or varying stoichiometry etc. )*

Good point. We will revise our language to better specify which ESMs include which

processes and feedbacks.

Regarding export changes in ESMs that already have particles with different size classes: Based on Fig. 9 in Bopp et al. (2013), we see that having different particle size classes does not seem to be a major determinant of the magnitude of projected export declines. Looking at Fig. 9b in particular, IPSL, CESM, and GFDL model projections span the entire range. As noted in a previous response, simply representing different sized particles does not ensure that a model will adequately represent the particle size feedback described here. They would also need to contain some mechanism that gives rise to the strong relationship between carbon export and particle size that we specify empirically in our model. We will mention this in our Discussion section.

*L 45/46 Maybe cite Laufkotter et al. 2015 for drivers of future declines in primary production as well*

Good point. An important reference! We will add this citation.

*Introduction: I miss a discussion or mentioning of other drivers of sinking speed/particle remineralization, such as particle density/porosity and ballasting, also fragmentation and aggregation which potentially change the particle size over time*

Good point. As stated above, some of this information is currently included in the conclusions section, but would be better placed in the introduction. In the revised manuscript, we will move this discussion to the introduction (and specifically mention the roles of density, ballasting, and fragmentation) and then reiterate it more succinctly in the conclusion.

*L 67/68 "parameters and processes in most previous models are not constrained by observations of particle size distributions" - Yes! Nice!*

Thanks!

*L74/75 Are there more recent references for the particle size distribution?*

[Figure]

Yes. We will add citations to the following references:

Cael, B.B. and White, A.E., 2020. Sinking versus suspended particle size distributions in the North Pacific Subtropical Gyre. Geophysical Research Letters, p.e2020GL087825.

White, A.E., Letelier, R.M., Whitmire, A.L., Barone, B., Bidigare, R.R., Church, M.J. and Karl, D.M., 2015. Phenology of particle size distributions and primary productivity in the North Pacific subtropical gyre (Station ALOHA). Journal of Geophysical Research: Oceans, 120(11), pp.7381-7399.

Buonassissi, C.J. and Dierssen, H.M., 2010. A regional comparison of particle size distributions and the power law approximation in oceanic and estuarine surface waters. Journal of Geophysical Research: Oceans, 115(C10).

*L 99 Just a small comment: I think the "shallowest" can be removed?*

We will change the statement from:

"Here we define remineralization depth as the shallowest depth at which POC flux. . ."

To:

"Here we define remineralization depth as **the depth** at which POC flux. . ."

*L 117 I think the decreased nutrient supply causes the decrease in phytoplankton/particle size, not the decrease in export?*

In our model, everything is linked by the empirical function/relationship between phytoplankton/particle size and export. Thus, the causal chain in our model is that the reduced nutrient supply results in reduced export, which (empirically) drives a shift towards smaller particles. In reality, this shift to smaller particles is likely driven by increased dominance of smaller phytoplankton, but this is not explicitly represented in our model, only the export/particle-size relationship is.

*The introduction reads slightly redundant to me and could be streamlined a bit, particularly part 1.2 (PSR feedback). However, I acknowledge that other reviewers seem to particularly like it and find it "very clear", so maybe for readers with little background in particle export it is better the way it is.*

We appreciate this observation. We will tighten this section, while maintaining the clarity that the other reviewers appreciated.

*L 164 Does the plankton not become small plankton, or does the small plankton not produce small particles? Oh I see, plankton community isn't explicitly represented, correct?*

Correct, the phytoplankton community isn't explicitly resolved - our model simulates phytoplankton growth (NPP) and organic matter export as a function of temperature, light and nutrients. The size spectrum of sinking particles ($\beta$) is then computed based on the empirically derived export/particle-size relationships.

*I don't understand where Prism has the microbial respiration rates from, and whether they increase due to temperature in a warmer ocean? Is there an oxygen dependence? Is there aggregation/fragmentation? Can the sinking speed change over depth? Given that these questions are essential to this study, please explain Prism in more detail.*

See our response above to the comment that reads "I think some aspects of the model should be described in more detail, in particular the PRiSM component used to estimate particle flux (see line-by-line comments)."

*L173/174 wouldn't it make sense to also increase the production of DOC when changing the beta of the particle size spectrum?*

In general, the cutoff between DOC and POC is arbitrary (i.e., based on filter sizes). Perhaps there should be more DOC production when particles are smaller (because there would be a relatively greater number of very small particles beneath the minimum filtration size), but we do not have sufficient information that we can use to parameterize

this in an empirical way.

*Fig 3: That's probably only my personal preference, but I generally prefer mapping data using a log scale, instead of plotting log(data) and using a linear scale.*

This preference is likely shared by many. We will change the scale to log rather than plotting log(data) and using a linear scale.

*L 233/234 The NPP estimates you are using are significantly biased in the Southern Ocean. This should probably be discussed here, or you could use an NPP estimate that has been specifically created for the Southern Ocean (for instance Johnsson et al. 2013).*

We thank the reviewer for pointing this out. At this stage, we cannot repeat our entire suite of analyses using a new NPP estimate, but we will certainly note this limitation of the estimates that we do use.

*L254 How sensitive are your results to this simplified representation of future ocean circulation?*

This is a great question, and not one that we can definitively answer. Because our model represents circulation using a mass-conserving transport matrix, we can only uniformly scale circulation rates up or down, rather than manipulating the patterns of circulation in more realistic ways. This limitation is already noted in our paper.

However, it is important to point to note that although we use a simplified representation of future changes in ocean circulation, the exact same simplified representation is used in both the PSR feedback-on and feedback-off cases.We are therefore isolating the effects of the particle size feedback from the effects of the circulation change itself. It is therefore not clear that our results would be significantly different if we used a model with a more complex representation of future circulation changes, as long as that model also applied the exact same circulation changes in feedback-on and feedback-off scenarios.

*L 282 In light of these "counterintuitive" relationships I think it would be really interesting to use an NPP algorithm that's been developed for the Southern Ocean, as mentioned above (the result might stay the same of course, and I like how it is discussed/explained with the LamBishop findings)*

Good point - we will mention this in the text as a suggested direction for future work.

*L 346: Is this a conclusion from the correlation between P200m and E or something that is actively diagnosed in the model?*

We answer this question and the following L355 question together below.

*L 355 ff The considerations up to this point hold for the nutrient-limited low and mid latitudes. But here you discuss a global uniform decrease of circulation rates. I found the jump from regional to global a bit confusing. Wait, I see now that in line 349, eq 3 is meant to hold at any given location, including regions in which eq. 2 does not hold, yes? Is that mathematically sound??*

This answer is in response to the above 2 comments.

Equation 2 is intended to provide a simplified interpretation of results in terms of changes in subsurface nutrient concentrations. It reflects the assumption that at steady-state, the export flux out of the euphotic zone must approximately balance the supply of nutrients into the euphotic zone by upwelling. Of course, because this is a simplification of reality, the relationship will never be 100% accurate, if this is what the reviewer means by "mathematically sound." However, it is a reasonable assumption for much of the ocean and is widely used in the biogeochemical literature (e.g. Ducklow et al., 2001; Passow and Carlson, 2012). We note that equations 3-4 are directly derived from Equation 2 via perturbation analysis. They therefore hold precisely to the same degree that Equation 2 holds.

In the revised manuscript, we will note more clearly that Equations 2-4 are only intended to help simplify our interpretations, and are not mathematically equivalent to

the full model solution. However, Fig 7 demonstrates that Eqs 2-4 do explain much of the full model behavior, and in doing so allow us to deconvolve and better understand the different mechanisms leading to export changes. Where these equations hold, they allow us to demonstrate how the PSR feedback operates to buffer export under slower circulation (i.e., by increasing nutrient concentrations in the shallow subsurface). We will clarify wording in the text to explain this.

Circulation rates are indeed decreased uniformly across the global ocean ($\Delta w / w_{baseline}$ = -10% in all regions in Equation 4), but the strength of the PSR feedback varies regionally depending on how much P200 changes in that region between the feedback on and off runs (see Equation 4).

Added reference:

Passow, U. and Carlson, C.A., 2012. The biological pump in a high $CO_2$ world. Marine Ecology Progress Series, 470, pp.249-271.

*L 358 "This decrease in P200m is .." - Is this your interpretation or has this been diagnosed in the model?*

This is our interpretation, backed up by an understanding of how Southern Ocean nutrient utilization drives low latitude nutrient supply from other model studies (e.g., Sarmiento et al., 2004; Marinov et al., 2006).

We will cite the above 2 studies and change the wording from:

"This decrease in P200m is largely driven by enhanced biological nutrient utilization..."

To:

"This decrease in P200m **is likely** largely driven by enhanced biological nutrient utilization..."

Added references:

Sarmiento, J.L., Gruber, N., Brzezinski, M.A. and Dunne, J.P., 2004. High-latitude controls of thermocline nutrients and low latitude biological productivity. Nature, 427(6969), pp.56-60.

Marinov, I., Gnanadesikan, A., Toggweiler, J.R. and Sarmiento, J.L., 2006. The southern ocean biogeochemical divide. Nature, 441(7096), pp.964-967.

*L 404 - Is eq. 2 a close approximation in the low latitudes or globally? Is it possible to show on a map how well Eq 2 holds regionally?*

Fig. 7g-h show how well this approximation holds both zonally and regionally. The orange lines/bars show feedback strength as approximated from the right hand side of Equation 4, which is directly derived from Equation 2. The blue lines/bars show actual feedback strength calculated from changes in export (left hand side of Equation 4). The closer the orange lines/bars are to the blue lines/bars, the better the approximation in Equation 2. We will add more explanation of this in the text.

*L 484 is this a typo "1.16 times"? Otherwise, I don't understand, why 1.16 times? 1 - 16%? But I wouldn't understand that, either. Sorry if I am missing something obvious.*

We will change the sentence from:

"Within our model, including these effects reduces the magnitude of predicted 100-year changes in global export production by ~14% when circulation rates are decreased by a conservative 10% (Fig. 5). This implies that global models without the PSR feedback may be overestimating 100-year climate-driven export decreases by ~1.16 times."

To:

"Within our model, including these effects reduces the magnitude of predicted 100-year changes in global export production by ~14% when circulation rates are decreased by a conservative 10% (Fig. 5). This implies that global models without the PSR feedback may be **projecting** 100-year climate-driven export decreases **that are ~1.16 times too large.**"

*L 485 I believe the Pisces models used in the Bopp et al. study implements two different particle sizes, so it's possible that the PSR feedback is at least partly included? Please check*

Yes, this is true. This is mentioned in Lines 64-65, but we will clarify this and add more details.

*L 507/508 "PSR feedback strength remains relatively constant whether circulation rates are increased/decreased by 10% or 50%" - That is very surprising to me. Why would that be? Are you reaching a maximum/minimum particle size distribution such that a further decrease in nutrient supply does not affect the size distribution anymore? Please discuss this a bit more. I also think this sensitivity test should be mentioned in the results already.*

Remember that the PSR feedback strength is calculated as the relative difference in projected export decrease between the feedback on and off cases, which are both run with exactly the same circulation rate change. It is this PSR feedback strength, NOT the amount of PSD slope ($\beta$) change, that stays the same when you amplify circulation rate changes. Thus, even though $\beta$ changes more in the 50% decreased circulation case when the feedback is on, the PSR feedback strength is relatively unchanged from the 10% decreased circulation case. In other words, the percentage difference in projected export change between the PSR feedback on and off cases is constant even as circulation rates slow more. To be sure, however, absolute export decreases are indeed greater in both the feedback on and off cases under the 50% reduced circulation rates. We will add this explanation into the text.

*L 528 I think the study by Briggs et al. 2020 is good recent reference for fragmentation*

Good point. We will add this reference.

*L 545 The effect of temperature increases and oxygen decreases on future carbon export has been analyzed in Laufkotter et al 2017, maybe good to cite here*

Good point. We will add this reference.

*Conclusion: If I understand everything correctly, your results suggest that changes in particle size/sinking speed in the Southern Ocean have only a small impact on carbon export regionally and globally. This could be interpreted as a justification for modelers to give implementation of particle size a low priority in this region - would you agree?*

Very interesting point! I suppose this would be true if modelers only really cared about better resolving the effects of this particular feedback in the Southern Ocean. However, correctly modeling particle size in this region may be important for accurate estimates of absolute export. Indeed some of our earlier work shows that large particles in cold temperatures in the Southern Ocean contribute to particularly high carbon sequestration in this region (Cram et al., 2018). Thus, I think modelers/observationalists should keep working on improving particle size representations in the Southern Ocean.

[Figure]

**Fig. 1.** Particulate organic carbon fluxes at 500 m with and without the particle size remineralization feedback

---

## Author Comment (AC2) · 16 Aug 2020

For this journal's review process, authors are expected to post a response to all reviewer comments before revising the actual manuscript. Based on these author responses, the editor either invites the authors to submit a revised manuscript or directly rejects the manuscript. We therefore do not yet include a revised manuscript along with answers to the following comments. See more details on the process here: https://www.biogeosciences.net/peer_review/interactive_review_process.html.

In the paragraphs below, all reviewer comments will be italicized, while author responses will be in normal font.

*The authors apply a global biogeochemical model to examine the effect of variable particle (phytoplankton) size distribution on surface and subsurface nutrients, and their mutual feedbacks when nutrients are supplied under different physical forcings. The feedback effect of the (nutrient-dependent) size distribution and subsequent particle sinking and remineralisation dampens the model response to changes in physics. I find this manuscript generally well written. The authors do a great job in explaining the mechanisms involved. In general, the experimental design to disentangle the effects of circulation, ecology and sinking is clear and well justified. Thus, the manuscript provides valuable new insights into a potential negative feedback mechanism in global biogeochemical models. However, I have a few points that I think could be improved with regard to model description and its critical discussion.*

We thank the reviewer for their positive comments about our experimental design, and for their constructive criticism below, which we believe will help greatly improve the manuscript.

*(1) Model description: I recommend to describe the biogeochemical model, particle sinking and remineralisation in detail (including equations), and also explain its basic assumptions. As far as I understand, the model assumes a power law size distribution of particles at the surface; particles then sink depending on their size, and remineralise with a size independent rate. Therefore, the particle size distribution changes with depth, favouring large particles as depth increases, similar to the 1D approach presented by Kriest and Oschlies (2008). (In fact, there seem to be only small differences between both models, in terms of formulation and results.) Both approaches make quite strong implicit assumptions about constant individual particle properties, which do not change with time or depth. In particular, the models neglect any processes besides sinking and remineralisation that might affect the particle size distribution below the euphotic zone, such as particle breakup, reworking by zooplankton (e.g., flux feeding, formation of fecal pellets), particles becoming more or less porous because of bacterial degradation, etc.. Of course, one cannot address all details and compli-*
*cations at once especially in global models; but describing the current implicit model assumptions in detail would help the reader to understand how the model works, and what its limitations and merits might be.*

We thank the reviewer for pointing out that the model description falls short. The mathematical description of the PRiSM model has been fully laid out in previous references and cannot be repeated in full here. However, we agree that enough information needs to be provided to allow the reader to understand how the model "works", without extensive cross-referencing to previous papers. We will revise the methods section to expand the description of the PRiSM model. Specifically, at line 157 we will insert the following:

"In the PRiSM model (DeVries et al., 2014), particles are produced in the surface euphotic zone (<75m) following a power-law size spectrum, in which log10(particle number density) declines linearly with log10(diameter), and the relative abundance of large and small particles is controlled by the slope of the spectrum on a log-log scale ($\beta$). The simulated particle size spectrum then evolves through the water column due to remineralization and size-dependent sinking, which are each parameterized based on empirically derived relationships and observed particle properties. Remineralization is represented by first-order mass loss from each particle, such that particles shrink and sink more slowly with depth, resulting in attenuation of the particle flux. Because smaller, slower-sinking particles reside for longer within any given depth interval, and therefore have more time to remineralize, they are preferentially lost from the particle population over depth, resulting in a flattening of the size spectrum (reduced $\beta$) and thus increased average sinking speeds at deeper depths. A constant rate of microbial respiration is used, optimized to fit global in situ phosphate distributions (DeVries et al., 2014). There are therefore no temporal changes in bacterial respiration due to warming, for example, which allows us to isolate changes in export that stem from the PSR feedback alone. While PRiSM has recently been expanded to include temperature and oxygen effects on bacterial respiration and remineralization (Cram et al., 2018), as well

as to represent particle disaggregation (Bianchi Weber et al., 2018), here we use the original version described in DeVries et al. (2014), which can be solved analytically and has previously been optimized to best fit the global phosphate distribution. The recent expanded versions of PRiSM must be solved numerically, which is less computationally efficient and therefore not suitable for incorporation into our global three dimensional simulations."

Added reference:

Bianchi, D., Weber, T.S., Kiko, R. and Deutsch, C., 2018. Global niche of marine anaerobic metabolisms expanded by particle microenvironments. Nature Geoscience, 11(4), pp.263-268.

*(2) Model description: The description of the model and its general setup is somehow unclear about how the phytoplankton size distribution might be related to larger particles, which likely contribute most to mesopelagic and deep particle flux. For example, the work by Kostadinov et al (2009), from which the observed size distribution at the surface is taken, is based on phytoplankton, i.e. extends only to a size of ca. 50 um. However, the present model applies a size range of 20-2000 um (Table S1). Moreover, the model parameters sometimes seem to relate to phytoplankton properties (e.g., the exponent of eta=1.17 relating cell diameter to sinking speed is based on phytoplankton data by Smayda, 1971), whereas other relate more to porous aggregates (e.g., the exponent relating particle mass to size of zeta=1.62; see also Kriest, 2002 http://dx.doi.org/10.1016/S0967-0637(02)00127-9, and citations therein). Again, here it would be useful to present and discuss these basic model assumptions. This is done partly on page 3, yet I think this subsection could be improved (see below, my comments Lines 80ff, 82ff, 95). In summary, I would suggest to more clearly distinguish between phytoplankton and particle size distribution, and to address potential connections between these more comprehensively.*

We assume that the particle size distribution slope computed by Kostadinov et al.

(2009) continues to hold for particles larger than those they explicitly compute the slope for. Prior research backs up this assumption (e.g., Durkin et al., 2015). We will add a note about this in addition to the just mentioned citations in the text.

We assume that phytoplankton simply behave as smaller particles. We will make a note of this in the text.

The driving mechanism behind the particle size feedback is the relationship between export production and particle size, which we determine empirically from remote sensing data in our study. Our model setup simply computes particle size based on this empirical relationship to export, and makes no explicit assumption about the root cause of the relationship. We hypothesize that the export/particle size relationship arises from plankton community structure simply because this seems like an intuitive mechanism, and is supported by correlative evidence: large particles and large phytoplankton taxa are both generally more dominant in regions of high productivity and export (Cram et al., 2018; Hirata et al., 2011), and we therefore find it reasonable to assume that large phytoplankton aggregate (either directly or by grazing) into large particles. However, this needn't be true for the particle size feedback to hold. Any other mechanism that gives rise to the observed export/particle-size relationship would give rise to the same feedback. In the revised manuscript, we will be more careful about distinguishing between the explicit assumptions and relationships "baked in" to our model, and the mechanisms that we are hypothesizing give rise to those relationships.

Added reference:

Durkin, C.A., Estapa, M.L. and Buesseler, K.O., 2015. Observations of carbon export by small sinking particles in the upper mesopelagic. Marine Chemistry, 175, pp.72-81.

*(3) Experimental setup: To me it is not clear how the circulation was reduced (e.g., Lines 253-254 "To simulate increased water column stratification and reduced vertical exchange due to warming, we uniformly and instantaneously reduce circulation and diffusion rates by 10% throughout the ocean.") - I would appreciate a more in depth*

*explanation.*

Our model uses the Transport Matrix Method, in which all physical fluxes (advection and diffusion) of element X are represented by the matrix-vector product A*X, in which A is the mass-conserving transport matrix that quantifies the mass exchanges between every gridcell in the model. We change circulation rates in an idealized way, simply by multiplying A by a factor of 0.9 (a 10% reduction in circulation rates) or 1.1 (a 10% increase in circulation rates). Therefore, the patterns of circulation remain unchanged, but the absolute exchange rates are scaled up or down. This explanation will be added into the revised manuscript.

*(5) Discussion: The model shows a large response and differences between the two setups (with or without PSR) in the equatorial upwelling regions. However, especially models of coarser resolution tend to suffer from an insufficient representation of the equatorial current system, with possible consequences for the representation of nutrients and/or oxygen (e.g., Dietze and Loeptien, 2013, https://doi.org/10.1002/gbc.20029; Duteil et al., 2014, https://doi.org/10.1029/2011GL046877). I would suggest to add some discussion on these potential effects.*

Great point. Coarse resolution dynamical models do tend to represent equatorial regions poorly. However, our transport matrix is derived from the observationally-constrained Ocean Circulation Inverse Model, which assimilates passive and transient water mass and ventilation tracers. Thus, even though the resolution does not allow accurate simulation of equatorial currents from a dynamical perspective, the data-assimilation ensures that the net effect of these currents on tracer transport is realistic. The model has been used successfully for simulation of nutrients (DeVries, 2014) and oxygen (DeVries and Weber, 2017), and does not suffer from the equatorial biases often evident in coarse resolution models. Nevertheless, we will point out the potential shortcomings of using a coarse resolution model in the discussion section of the revised manuscript.
*(4) Discussion: Section 3 is named "Results and discussion", yet it almost entirely presents the results. In contrast, Section 4 is named "Conclusions", but partly discusses the results before the background of other works, is quite long and partly repetitive. I would suggest to rename section 3 to "Results", add a "Discussion" section, that extends a bit on the comparison of results obtained here with other model studies and also includes a critical discussion of model processes and properties. The "Conclusions" section could then be shortened and more concise.*

We thank the reviewer for this helpful suggestion. We will take this suggestion seriously as we re-write the paper and will make sure that the final version is structured clearly.

*Specific comments:*

*- Lines 35-37: "Where sinking POC fluxes are particularly high, enhanced bacterial breakdown of particles can deplete available oxygen and create hypoxic or even suboxic conditions [...] " - there are many places in the ocean where sinking POC fluxes are high; another necessary condition for the development of OMZs is that supply of oxygen by physical transport is low.*

We will change the statement from:

"Where sinking POC fluxes are particularly high, enhanced bacterial breakdown of particles can deplete available oxygen and create hypoxic or even suboxic conditions..."

To:

"Where sinking POC fluxes are particularly high **and supply of oxygen via physical transport is low,** enhanced bacterial breakdown of particles can deplete available oxygen and create hypoxic or even suboxic conditions..."

*- Line 66: Note that there are further global ocean models that address spatial and temporal variation of the size distribution (of marine aggregates) and sinking speed, e.g., Schwinger et al. (2016, www.geosci-model-dev.net/9/2589/2016/) and Niemeyer et al. (2019, https://doi.org/10.5194/bg-16-3095-2019). On a local (1D) scale, even*

[Figure]

*very complex models of particle transformations have been developed (Jokulsdottir and Archer, 2016, www.geosci-model-dev.net/9/1455/2016/).*

We thank the reviewer for bringing these highly relevant and useful studies to our attention. We will add discussion and citation of these studies to our text.

*- Lines 80ff: "Large particles tend to exist in the ocean where larger microphytoplankton (>20 um in diameter) are dominant, while relatively small particles tend to exist where smaller picophytoplankton (<2 um in diameter) are dominant (Guidi et al., 2007; Guidi et al., 2008; Guidi et al., 2009). [...]" - The observations by Guidi et al. (2007, 2008), are based on UVP data of large particles (aggregates, fecal pellets, ...), of a size of at least 250 um. Therefore, I don't think that these observations can be used to justify the assumptions about the phytoplankton size distributions made in this paper.*

Yes, it is true that UVP data is measuring larger particles. However, our model does include some particles of these sizes, as the reviewer noted above. Furthermore, again, we assume that the particle size distribution slope holds throughout the entire range of particle sizes in the ocean. We will make a note of this in our text. See also our response to the "(2) Model description" comment above.

*- Lines 82ff: "The presence of large phytoplankton leads to the generation of larger particles perhaps because large phytoplankton are more likely to form aggregates and be transformed into large fecal pellets by large zooplankton, whereas small phytoplankton are more likely to be degraded by bacteria and consumed by smaller zooplankton (Bopp et al., 2005; Guidi et al., 2007; Guidi et al., 2009; Michaels and Silver, 1988). The exact mechanisms governing the processes by which smaller and larger phytoplankton become smaller and larger particles are not clearly known, however, and is an active area of research." - The global model study by Bopp et al. does not address aggregates; moreover, as a model study it is based on a priori assumptions, and does not provide insight into real in situ mechanisms. As noted above, the study by Guidi et al. (2007) addresses the UVP size range and the study by Michaels et al. is also a*

*(food web) model. While I tend to agree with the idea that large phytoplankton triggers large sinking particles, I would appreciate a more convincing reasoning why one can extend the phytoplankton size range up to particles 2 millimeters in diameter.*

As stated in response to the "Model Description (2)" comment above and repeated here:

"We assume that the particle size distribution slope computed by Kostadinov et al. (2009) continues to hold for particles larger than those they explicitly compute the slope for. Prior research backs up this assumption (e.g., Durkin et al., 2015). We will add a note about this in addition to the just mentioned citations in the text."

Furthermore, Kostadinov et al. (2009)'s particle size distribution slope is available globally and is also temporally and spatially resolved. Thus, this was really the best PSD slope dataset that we could find. In essence, it was what was available. Our method really requires this kind of global data available over long enough timescales and with enough spatiotemporal resolution to compute the necessary correlations.

Additionally, to reiterate our points above, though we do implicitly assume that small phytoplankton = small particles to explain/understand the results of our empirical analyses, our model setup and study in general do not require this relationship to be true. Our model setup simply computes particle size as an empirical function of export. The empirical positive relationship between export and particle size illuminated by our satellite data analysis showed that increasing productivity and export are associated with larger particle sizes. Increasing productivity is in turn associated with larger phytoplankton; thus, based on our empirical analysis, it seems that larger phytoplankton are associated with larger particle sizes. We bring up other studies that find or assume this merely to better explain the mechanistic underpinnings of this result and NOT to say that this must be the case for our model setup to hold. Our empirical analyses could have shown that export and particle size were negatively correlated, for example, in which case, our model would have demonstrated a positive particle size

remineralization feedback, rather than a negative one. Thus, our results/findings of a negative particle size remineralization feedback effect hinge entirely on the direction of the empirical relationship between export and particle size, rather than on our own assumptions. We will alter wording/discussion in the paper to clarify this point.

*- Line 95: "Past work has also firmly established a strong positive relationship between particle size and sinking speed in the ocean (Alldredge and Gotschalk, 1988; Smayda, 1971) [...]" - The relationship between diameter and sinking speed in Alldredge and Gotschalk (1988) is w=50 d ^0.26, and shows considerable scatter. I would not call this a strong relationship. This weak relationship is possibly because of the fractal and variable nature of aggregates - indeed, single cells show a higher exponent (Smayda, 1971). Again, here I would suggest to more clearly distinguish between aggregates and single phytoplankton cells.*

We will change the sentence from:

"Past work has also firmly established a strong positive relationship between particle size and sinking speed in the ocean (Alldredge and Gotschalk, 1988; Smayda, 1971) (although there are exceptions to this rule, particularly in the Southern Ocean – see McDonnell and Buesseler (2010))."

To:

"Past work has also **suggested a positive relationship** between particle size and sinking speed in the ocean (Alldredge and Gotschalk, 1988; Smayda, 1971), although **there appear to be complications and exceptions to these rules (Cael and White, 2020; Laurenceau-Cornec et al., 2019),** particularly in the Southern Ocean (McDonnell and Buesseler, 2010)."

Added references:

Cael, B.B. and White, A.E., 2020. Sinking versus suspended particle size distributions in the North Pacific Subtropical Gyre. Geophysical Research Letters,

p.e2020GL087825.

Laurenceau-Cornec, E.C., Le Moigne, F.A., Gallinari, M., Moriceau, B., Toullec, J., Iversen, M.H., Engel, A. and De La Rocha, C.L., 2020. New guidelines for the application of Stokes' models to the sinking velocity of marine aggregates. Limnology and Oceanography, 65(6), pp.1264-1285.

*- Line 100: "by a factor of e" - e is 2.718, do you really mean a factor of e?*

Yes, this is the e-folding length scale.

*- Line 157: "physical relationships between particle size, mass, and sinking velocity"*
*- I don't think that the relationship between particle size, mass and sinking velocity of organic particles is a purely physical one; at least the relationships by Smayda (1971) and Alldredge and Gotschalk (1988) are empirical. I suggest to skip "physical".*

We will change the sentence from:

"PRiSM computes particle flux profiles as a function of particle size distribution ($\beta$) at the surface, microbial remineralization rates, and physical relationships between particle size, mass, and sinking velocity."

To:

"PRiSM computes particle flux profiles as a function of particle size distribution ($\beta$) at the surface, microbial remineralization rates, **and relationships** between particle size, mass, and sinking velocity."

*- Line 184-185: "time-mean export" - mean over what time? A year?*

We will change the statement from:

"...time-mean normalized export ($E_{n,obs}$) (i.e., absolute export divided by time-mean export at a given grid point)."

To:

"...time-mean normalized export (En,obs) (i.e., absolute export divided by time-mean export **calculated over the SeaWiFS period** at a given grid point)."

*- Line 483-484: "This implies that global models without the PSR feedback may be overestimating 100-year climate-driven export decreases by ~1.16 times." - What is meant with 1.16 times?*

We will change the sentence from:

"Within our model, including these effects reduces the magnitude of predicted 100-year changes in global export production by ~14% when circulation rates are decreased by a conservative 10% (Fig. 5). This implies that global models without the PSR feedback may be overestimating 100-year climate-driven export decreases by ~1.16 times."

To:

"Within our model, including these effects reduces the magnitude of predicted 100-year changes in global export production by ~14% when circulation rates are decreased by a conservative 10% (Fig. 5). This implies that global models without the PSR feedback may be projecting 100-year climate-driven export decreases that are ~1.16 times too large."

---

## Author Comment (AC3) · 16 Aug 2020

Apologies - The author comment file titled "Response to Reviewer 1" should actually be "Response to Reviewer 3." This document is the actual author response to Reviewer 1.

For this journal's review process, authors are expected to post a response to all reviewer comments before revising the actual manuscript. Based on these author responses, the editor either invites the authors to submit a revised manuscript or directly rejects the manuscript. We therefore do not yet include a revised manuscript along with answers to the following comments. See more details on the process here:

In the paragraphs below, all reviewer comments will be italicized, while author responses will be in normal font.

*This manuscript examines the influence of incorporating information on phytoplankton size into a biogeochemical model alters predicted carbon export in response to climate change. The overall result that the influence of climate change is damped by incorporating size information is intriguing and worthy of publication. Generally, the manuscript is clear and well-written., and represents a useful addition to the literature.*

We thank the reviewer for recognizing the contribution our study makes and for their constructive comments, which we believe will improve the manuscript if a revised version is invited.

*Abstract:*

*Line 20: the altered export values are reported without any reference to what the baseline simulation is, i.e. report what is the predicted decline in export without the size considerations, and what is it with the size included*

Good point. We will change the sentence from:

"This negative feedback mechanism (termed the particle size-remineralization feedback) slows export decline over the next century by ~14% globally and by ~20% in the tropical and subtropical oceans, where export decreases are currently predicted to be greatest."

to:

"This negative feedback mechanism (termed the particle size-remineralization feedback) slows export decline over the next century by ~14% globally **(from -0.29 GtC/year to -0.25 GtC/year)** and by ~20% in the tropical and subtropical oceans, where export decreases are currently predicted to be greatest."

*Line 22: "more robust predictions" – How do you know these predictions are more robust than the baseline? There is no model validation presented (which I don't mind for this manuscript, but "more robust" can't be asserted in this case).*

Good point. We will change the sentence to recognize that including the feedback may be important for future predictions, without implying that the feedback is certain to make these predictions more accurate. We will thus change the sentence from:

"Thus, incorporating dynamic particle size dependent remineralization depths into Earth System Models will result in more robust predictions of changes in biological pump strength in a warming climate."

to:

"Thus, incorporating dynamic particle size dependent remineralization depths into Earth System Models **may be important when predicting** changes in biological pump strength in a warming climate."

*Introduction:*

*Line 41: how do the models cited here handle size or sinking speed (if they do at all)? i.e. do these models also already include a size-based parameterisation which means that the predictions are ~equivalent to yours?*

Discussion of how ESMs handle size/sinking speed is contained in Lines 59-68, but is relatively brief. In a revised manuscript, we will further expand on how these models work and the different approaches they take, as requested by other reviewers as well.

*Line 86-87: a fundamental assumption in this study is that small phytoplankton = small particles. A critical assessment from observational data of whether this is true, and when this assumption might break down, should be included. For example, how might TEP production or fragmentation affect the size structure of particles?*

We answer this question and the following Line 91-92 comment together below.

*Line 91-92: references needed for the assertion that small picophytoplankton = small particles, and same for large.*

The following is in response to both of the above comments.

The driving mechanism behind the particle size feedback is the relationship between export production and particle size, which we determine empirically from remote sensing data in our study. Our model setup simply computes particle size based on this empirical relationship to export, and makes no explicit assumption about the root cause of the relationship. We hypothesize that the export/particle size relationship arises from plankton community structure simply because this seems like an intuitive mechanism, and is supported by correlative evidence: large particles and large phytoplankton taxa are both generally more dominant in regions of high productivity and export (Cram et al., 2018; Hirata et al., 2011), and we therefore find it reasonable to assume that large phytoplankton aggregate (either directly or by grazing) into large particles. However, this needn't be true for the particle size feedback to hold. Any other mechanism that gives rise to the observed export/particle-size relationship would give rise to the same feedback. In the revised manuscript, we will be more careful about distinguishing between the explicit assumptions and relationships "baked in" to our model, and the mechanisms that we are hypothesizing give rise to those relationships.

We agree completely with the reviewer that TEP production or fragmentation would affect the size structure of particles. Lack of resolution of these processes is a limitation of our study. However, data to constrain these processes are limited and adding TEP and fragmentation would make for a substantially more complex model. Our group is currently working on modeling fragmentation in an ongoing project. We will add more emphasis on these limitations and look forward to future studies examining the importance of these other processes.

*Line 97: there's a dawning realisation that Stokes law rarely holds for marine particles e.g. https://aslopubs.onlinelibrary.wiley.com/doi/abs/10.1002/lno.11388 This should be*

*acknowledged here.*

We will change the sentence from:

"Past work has also firmly established a strong positive relationship between particle size and sinking speed in the ocean (Alldredge and Gotschalk, 1988; Smayda, 1971) (although there are exceptions to this rule, particularly in the Southern Ocean – see McDonnell and Buesseler (2010))."

To:

"Past work has also **suggested a positive relationship** between particle size and sinking speed in the ocean (Alldredge and Gotschalk, 1988; Smayda, 1971), although **there appear to be complications and exceptions to these rules (Cael and White, 2020; Laurenceau-Cornec et al., 2019),** particularly in the Southern Ocean (McDonnell and Buesseler, 2010)."

Added references:

Cael, B.B. and White, A.E., 2020. Sinking versus suspended particle size distributions in the North Pacific Subtropical Gyre. Geophysical Research Letters, p.e2020GL087825.

Laurenceau-Cornec, E.C., Le Moigne, F.A., Gallinari, M., Moriceau, B., Toullec, J., Iversen, M.H., Engel, A. and De La Rocha, C.L., 2020. New guidelines for the application of Stokes' models to the sinking velocity of marine aggregates. Limnology and Oceanography, 65(6), pp.1264-1285.

*Line 127: but wouldn't a shift to smaller particles also result in less C sequestration at depth as the C just goes round and round in the upper mesopelagic being readily recycled and re-entrained to surface? Also, Figure 2d – the caption acknowledges that smaller particles leads to greater recycled nutrient supply. This wouldn't increase C sequestration (or $CO_2$ drawdown) as that depends on the resupply of preformed nutrients which isn't affected by the size considerations used here, at least on the*

*timescales considered.*

We agree with the reviewer, and make a similar point in our conclusions section. Ocean carbon storage does not just depend on export, but on the sequestration timescale of the exported carbon (Boyd et al., 2019). The particle size feedback helps maintain export, but also results in shallow remineralization and therefore a shorter carbon sequestration timescale. However, our manuscript is solely focused on future changes in carbon export, not ocean carbon storage. Export is a critical process in its own right, even when decoupled from changes in carbon storage, because it is the source of nutrition to the mesopelagic twilight, and therefore determines the productivity of heterotrophic communities, including commercial fisheries. For this reason, carbon export is one of the key variables that is focused on in ocean biogeochemistry forecasts. While changes in the biological pump may also drive changes in ocean carbon storage, these will manifest over longer timescales than changes in export, and will likely be overwhelmed by the effects of anthropogenic $CO_2$ uptake and solubility-driven outgassing. A detailed exploration of changes in carbon storage is therefore beyond the scope of the current paper, and is only discussed in the introduction and conclusions. However, the reviewer's comment has highlighted the fact that we have not drawn the distinction between carbon export and storage clearly enough. We will revise the manuscript to make this clearer, and incorporate some more detailed discussion of the implications of the PSR feedback for carbon storage in the conclusions section.

*Methods:*

*Line 217: just curious to know why Laws and Dunne estimates of export ratio, rather than others such as Henson et al. 2012 or Siegel et al. 2014 were used*

Good question. We use the Laws and Dunne relationships because Weber et al. (2016) showed that when these algorithms, including Henson et al. (2012) were applied to satellite NPP, they gave the best matches to a range of in situ export estimates based on tracer/mass balance approaches. Seigel et al. (2014) was not used because

it provides no simple formula that can be applied to NPP to estimate export. Instead, they use an ecosystem model that makes its own assumptions about grazing, particle size, etc., and the export ratio is an emergent property of the model.

*Line 227: I couldn't find where the in situ observations mentioned here had come from*

We will change the sentence from:

"When reporting most-likely values, we weight the nine map sets according to how well each map set's annual mean export matches in situ observations within each region defined here (Table S3; see Weber et al. (2016) for derivation of weighting factors)."

To:

"When reporting most-likely values, we weight the nine map sets according to how well each map set's annual mean export matches in situ oxygen **and mass balance-based observations (Reuer et al., 2007; Emerson, 2014)** within each region defined here (Table S3; see Weber et al. (2016) for derivation of weighting factors)."

Added references:

Reuer, M.K., Barnett, B.A., Bender, M.L., Falkowski, P.G. and Hendricks, M.B., 2007. New estimates of Southern Ocean biological production rates from $O_2$/Ar ratios and the triple isotope composition of $O_2$. Deep Sea Research Part I: Oceanographic Research Papers, 54(6), pp.951-974.

Emerson, S., 2014. Annual net community production and the biological carbon flux in the ocean. Global Biogeochemical Cycles, 28(1), pp.14-28.

*Line 238-240: is temporal autocorrelation accounted for here? I guess the seasonal cycle in beta and En are similar which will affect the linear regression. Also, are beta and En independent? How are pixels/regions with non statistically significant regressions dealt with?*

For the purposes of this study, autocorrelation should not pose a problem. Autocorrelation only poses a problem if one is trying to determine how much of the relationship between two variables is because they are actually related and how much is just because both are seasonal or slowly varying over time. Seasonality and coincidental variation, from our perspective, both contribute to the relatedness of these two variables and so we want to include both in the calculation of their relationship. Indeed, the seasonal cycle in beta and En greatly affect the linear regression and are an important part of the measured effect/relationship. Their correlation with one another on the shortest available/reasonable timescales is what we were after here, which includes looking at how they vary with one another over months and seasons. We then assume that this relationship that holds on the monthly/seasonal timescale also holds on shorter timescales (i.e., on the timescale in which phytoplankton turn over/change community structure). We will make a note of this assumption. We also computed regression coefficients using monthly anomalies rather than raw monthly values and got similar values.

Beta and En are independent, but related measurements in that beta is derived from particulate backscattering spectra, while En is based on chlorophyll concentrations (as well as SST and euphotic zone depths). Though both particulate backscattering spectra and chl concentrations are ultimately derived from normalized water leaving radiances observed by SeaWiFS, the ways in which they glean information from these radiances are quite different. We will add mention of this in the revised manuscript.

The following further describes how beta is computed. First, the particulate backscattering spectra is computed from the slope connecting particulate backscattering coefficients at 490, 510, and 550 nm. These coefficients are in turn derived from normalized water leaving radiance at these same 3 wavelengths. To convert from a particulate backscattering spectra to beta, Mie modeling is used to establish a physical relationship/lookup table between the two variables ($\eta$ and $\xi$ in Kostadinov et al. (2009)).

Chlorophyll concentrations, on the other hand, are typically computed as follows: [chl] = $10^{(a + b*R + c*R^2 + d*R^3 + e*R^4)}$, where R = log10(maximum normalized water leaving radiance ratio out of 443 nm:555 nm, 490 nm:555 nm, and 510 nm:555

nm), and a-e are constants.

We average over the larger regions in order to avoid generating/using single grid cells with insignificant regressions. That is, avoidance of insignificant and spatially over-resolved/over-specified regression coefficients was the primary reason for averaging over regions.

*Line 266: in reality I suspect dbetaobs/dEnobs could vary seasonally. Might be worth a caveat on that point in the discussion? Actually, I suspect that some of the strange behaviour in the SAZ might be due to a seasonal effect or a time lag between changes in phytoplankton size and particle export. It would be helpful to the reader to include some example annual time series of a region showing the PP, export and beta to illustrate how they interact. It would also inform on potential time lags between PP, export and beta.*

As was discussed above, we in fact use the variability over the seasonal cycles of beta and En to see how they change in concert over the seasons. Again, this is part of their computed relationship, encapsulated in dbetaobs/dEnobs. We thus compute the relationship between beta and export by in effect exploiting their seasonal variations.

The relationship between beta and export could also be much more non-linear than assumed here. Furthermore, dbetaobs/dEnobs could also vary over time as ocean physics and nutrient limitation conditions are altered by climate warming, for example. Thus, our approach represents a tractable, but simplified approximation. We will make an additional note of this.

We have attached below figures of example beta and En time series over various regions, along with a map showing where the sample points were located within each region. dbetadobs/dEnobs values are calculated over the entire SeaWiFS period (Sep 1997 - Dec 2010), but we show only a random subset of these years for visual clarity. We will add a reduced version of these figures into the supplementary material. (In particular, we will add 1 supplementary figure of export and beta time series in the

SAZ over all export algorithms, and 1 supplementary figure of export and beta time series over all regions using only 1 example export algorithm.) One can see that there is likely not much of a time lag in the SAZ. Furthermore, from time series over all of the regions, one can indeed see that the vast majority of the variance in both beta and export occurs over seasonal (rather than interannual, say) timescales; thus, it is really their seasonalities that allow us to define dbetadobs/dEnobs in the first place and there is not enough data to get more granular than this (i.e., to subseasonal scales looking at differences between seasons).

*Results and Discussion:*

*A point that should be acknowledged somewhere is that the results presented here are of course still dependent on the details of the model parameterisation and choices.*

Good point. We will add this note.

*Line 301-303: references needed here*

We will add the missing reference to Bopp et al. (2013).

*Line 322: isn't this 21% rather than 18%? I found the use of the word "visually" here and on line 307 confusing. It made me think that you had estimated the values by eye rather than calculating them. I suggest just dropping 'visually'.*

The 18% reduction was calculated as follows: (export increase w/ feedback - export increase w/o feedback) / (export increase w/o feedback) = (0.23 - 0.28 molC/m ^2/yr) / (0.28 molC/m ^2/yr) = -17.9%.

We thank the reviewer for pointing out that this is confusing - we will drop the word "visually."

*Line 346: some C:P ratio must be used here too? Couldn't find where that was mentioned. Does this formulation also assume that all nutrients supplied are regenerated? I think it assumes that all nutrients supplied are turned into PP (which is fine in nutrient*

*limited regions), and then are all exported i.e. and e-ratio of 1?*

We do not assume that all nutrients supplied are regenerated. Nutrient concentrations are the sum of preformed and regenerated nutrients. We will add a note of this to our existing text in Section 2.1.1, which is quoted below:

"This scheme calculates phytoplankton growth rates as a function of observed annual-mean temperatures (Locarnini et al., 2010) and solar radiation levels (Rossow Schiffer, 1999), along with modeled PO4 3-. We thus explicitly model phytoplankton production in terms of phosphorus consumption and regeneration. We then use an empirical, spatially variable relationship between particulate C-to-P ratios and phosphate concentrations (Galbraith Martiny, 2015) to convert phytoplankton production into units of carbon. It is assumed that 10% of phytoplankton production is routed directly to dissolved organic matter in the euphotic zone, with the remainder becoming particulate organic matter (Thornton, 2013)."

*Line 376-378: I'm not sure this "visually" statement helps the reader's understanding here*

We will drop the word "visually."

*Line 415-426: specify the direction of +ve/-ve changes in the caption. At the moment it's a bit confusing as +ve indicates a reduction*

Great idea. We will add this specification.

*Figure 1a: specify in the caption that higher beta values = smaller plankton (or mark with arrows on the colour bar)*

Great idea. We will add this specification.

*Figure 6a: rather than having the right y-axis in remin depth x 100, just write out the numbers in full – it's clearer*

We thank the reviewer for this helpful attention to detail. We will change the labels as

requested.

*Figure S1: specify in the caption or legend that higher beta values = smaller plankton*

We will add this specification.

*Figure S2: add the key to the PP and e-ratio model abbreviations into the caption here*

Good point. We will add the key.

*Table S1: Define the parameter names in this table.*

We will add the parameter names.
* * *
**Fig. 1.** Map of example time series locations and their associated regions. Each different color on the map delineates a different region as defined in the paper. Corresponding time series are in Figs. 2-8.

[Figure]

**Fig. 2.** Example beta and time-mean normalized export time series in the given region.

[Figure]

**Fig. 3.** Example beta and time-mean normalized export time series in the given region.

[Figure]

**Fig. 4.** Example beta and time-mean normalized export time series in the given region.

[Figure]

**Fig. 5.** Example beta and time-mean normalized export time series in the given region.

[Figure]

**Fig. 6.** Example beta and time-mean normalized export time series in the given region.

[Figure]

**Fig. 7.** Example beta and time-mean normalized export time series in the given region.

[Figure]

---

## Author Comment (AC4) · 16 Aug 2020

Apologies - The author comment file titled "Response to Reviewer 1" is actually the author response to Reviewer 3. Please see the file "Response to Reviewer 1" instead.

---

## Author Response (AR1)

**Response to Reviewer 1**

In the paragraphs below, all reviewer comments will be italicized, while author responses will be in normal font. All changes to the manuscript described below are highlighted in yellow in the corresponding marked up manuscript.

*This manuscript examines the influence of incorporating information on phytoplankton size into a biogeochemical model alters predicted carbon export in response to climate change. The overall result that the influence of climate change is damped by incorporating size information is intriguing and worthy of publication. Generally, the manuscript is clear and well-written., and represents a useful addition to the literature.*

We thank the reviewer for recognizing the contribution our study makes and for their constructive comments, which we believe have improved the manuscript.

*Abstract:*

*Line 20: the altered export values are reported without any reference to what the baseline simulation is, i.e. report what is the predicted decline in export without the size considerations, and what is it with the size included*

Good point. We changed the sentence from:

"This negative feedback mechanism (termed the particle size-remineralization feedback) slows export decline over the next century by ~14% globally and by ~20% in the tropical and subtropical oceans, where export decreases are currently predicted to be greatest."

to:

"This negative feedback mechanism (termed the particle size-remineralization feedback) slows export decline over the next century by ~14% globally **(from -0.29 GtC/year to -0.25 GtC/year)** and by ~20% in the tropical and subtropical oceans, where export decreases are currently predicted to be greatest."

*Line 22: "more robust predictions" – How do you know these predictions are more robust than the baseline? There is no model validation presented (which I don't mind for this manuscript, but "more robust" can't be asserted in this case).*

Good point. We changed the sentence from:

"Thus, incorporating dynamic particle size-dependent remineralization depths into Earth System Models will result in more robust predictions of changes in biological pump strength in a warming climate."

to:

"Our findings suggest that to more accurately predict changes in biological pump strength under a warming climate, Earth System Models should include dynamic particle size-dependent remineralization depths."

Introduction:

*Line 41: how do the models cited here handle size or sinking speed (if they do at all)? i.e. do these models also already include a size-based parameterisation which means that the predictions are ~ equivalent to yours?*

Good question. We added Section 3.5: Comparison with CMIP5 models to our text to discuss this, as follows:

"3.5 Comparison with CMIP5 models

In the absence of the PSR feedback, our model predicts a 100-year global mean export decrease of 8.1% (0.29 molC m-2 yr-1). With the PSR feedback on, this export decrease is reduced to 7.0% (0.25 molC m-2 yr-1). Meanwhile, CMIP5 models project global mean export decreases of around 7-18% between 2090-2099 and 1990-1999 under a "business-as-usual" radiative forcing scenario (RCP8.5), with an ensemble mean of 13% (Bopp et al., 2013). Assuming that none of the CMIP5 models are currently able to simulate the PSR feedback, our results therefore suggest that accounting for the feedback would alter the CMIP5 range of projections from a 7-18% to a 6-15.5% decline in export, and the CMIP5 ensemble mean projection from a 13% to an 11% decline in export. Many of the CMIP5 models may be capable of capturing some semblance of a PSR feedback effect, however, thus necessitating smaller corrections.

All CMIP5 models simulate various processes that have the potential to change POC export fluxes in a warming future ocean, including zooplankton grazing/fecal pellet formation, phytoplankton aggregation, phytoplankton/zooplankton mortality, and variations in phytoplankton community structure (which provide the source material for sinking particles) based on changing nutrient, temperature, and light conditions. However, 7 of 17 total CMIP5 models with ocean biogeochemistry simulate only one class or type of particulate organic matter (Ilyina et al., 2013; Tjiputra et al., 2013; Tsujino et al., 2010; Watanabe et al., 2011; Zahariev et al., 2008), and therefore cannot capture changes in the nature of sinking POC with future warming. The other 10 CMIP5 models can simulate changes in the nature of sinking particulate organic matter with future warming, either through the amount of associated ballasting (3 models) (Dunne et al., 2013; Moore et al., 2004) or through particle size (7 models).

Of the 7 models that simulate changes in particle size and could thus potentially capture the PSR feedback effect, 3 of these models resolve two particle sizes (small and large) with different sinking speeds (3 m day-1 for the small particles and 50 to 200 m day-1 for the large particles) (Aumont and Bopp, 2006). Another 2 of these 7 models also resolve two size-based particle types (diatoms and detritus) with different sinking speeds (1 m day-1 for diatoms and 10 m day-1 for detritus) (Totterdell, 2019). The final 2 of these 7 models simulate 5 different organic particle sinking speeds based on size, 1 for each different phytoplankton type in the model (for a total of 4) and 1 for carbon detritus (Romanou et al., 2013). In sum, out of the 17 CMIP5 models described here, only 7 resolve particles of more than one size that sink down the water column at different speeds and therefore have the potential to capture some part of the PSR feedback effect.

If all the CMIP5 models differed only in their resolution of sinking particle sizes, then we would expect the 7 models with more than one particle size to project the smallest decreases in export with future warming. In reality, however, the models differ in too many other ways to isolate potential impacts of the PSR feedback when comparing between them. For example, three of the models that resolve more than one sinking particle size (IPSL-CM5A-LR, IPSL-CM5A-MR, and HadGEM2-ES) predict some of the largest decreases in export production by 2100 (Fig. 9b in Bopp et al., 2013), contrary to what would be expected given the potential presence of a PSR feedback in these models. Indeed, the models that can simulate changes in the nature of sinking particles project changes in export that span the entire range of CMIP5 model predictions (Fig. 9b in Bopp et al., 2013). The reasons for these differences in projected export decreases is difficult to disentangle and would require examining the effects of one mechanism at a time on export in each model.

Simply representing differently sized particles also does not ensure that a model will adequately represent the negative PSR feedback quantified in this study. To adequately represent the negative PSR feedback, a given model would need to contain mechanisms that give rise to the same strong, empirically-derived relationships between POC export and particle size that we specify here. Within the models that resolve particle size to some degree, the relative proportion of large and small particles is determined by internal model dynamics and are not prescribed empirically. Furthermore, these models do not resolve a particle size spectrum over a wide range of sizes as is done here. Therefore, CMIP5 models that dynamically resolve 2-5 different particle size classes with different sinking speeds might qualitatively reproduce the same feedback, but it is not clear whether the magnitude or even the sign of the feedback would be accurately captured. We argue that our model, which resolves particle size spectra over a wide range of sizes and employs empirical export/particle-size relationship constraints, is most likely to accurately capture the true magnitude of the PSR feedback. We thus suggest that our study provides a reasonable baseline against which more complex Earth System Models can assess their ability to reproduce particle size-remineralization feedbacks."

*Line 86-87: a fundamental assumption in this study is that small phytoplankton = small particles. A critical assessment from observational data of whether this is true, and when this assumption might break down, should be included. For example, how might TEP production or fragmentation affect the size structure of particles?*

We answer this question and the following Line 91-92 comment together below.

*Line 91-92: references needed for the assertion that small picophytoplankton = small particles, and same for large.*

The following is in response to both of the above comments.

The driving mechanism behind the particle size feedback is the relationship between export production and particle size, which we determine empirically from remote sensing data in our study. Our model setup simply computes particle size based on this empirical relationship to export, and makes no explicit assumption about the root cause of the relationship. We hypothesize that the export/particle size relationship arises from plankton community structure simply because this seems like an intuitive mechanism, and is supported by correlative evidence: large particles and large phytoplankton taxa are both generally more dominant in regions of high productivity and export (e.g., Cram et al., 2018), and we therefore find it reasonable to assume that large phytoplankton aggregate (either directly or by grazing) into large particles. However, this needn't be true for the particle size feedback to hold. Any other mechanism that gives rise to the observed export/particle-size relationship would give rise to the same feedback. In the revised manuscript, we are more careful about distinguishing between the explicit assumptions and relationships "baked in" to our model, and the mechanisms that we are hypothesizing give rise to those relationships.
We added the following text to address these points:

Line 74:
"Potential mechanisms that could drive changing particle sizes include changes in underlying phytoplankton community structure and organic matter packaging processes at higher trophic levels. Whatever the mechanism, the direction and magnitude of the particle size-remineralization feedback in a warming ocean will depend on how particle sizes change as export declines in the future. If the export decline is associated with a shift towards larger organic particles that sink more quickly, remineralization depths will deepen and further reduce surface nutrient supply and export in a positive feedback. If, on the other hand, export decreases are associated with a shift towards smaller sinking particles, shallower remineralization will allow faster nutrient recirculation to the surface and dampen stratification-driven decreases in nutrient supply and export in a negative feedback."

Line 159:
"When the PSR feedback is disabled within our model, circulation-driven changes in the nutrient supply to the euphotic zone (see Section 2.3) will lead to changes in POC export, but $\beta$ (and therefore particle remineralization depths) remains constant over time. With the PSR feedback enabled, any change in POC export is accompanied by a change in $\beta$, the direction and magnitude of which is specified using the empirical relationships discussed in Section 2.2. We note that by design, this modeling approach makes no assumptions about the mechanisms driving shifts in the particle size distribution, only that $\beta$ changes in tandem with POC export, in a manner that is consistent with observations."

Line 278:
"Because $\beta$ and export are negatively correlated, export tends to be high when $\beta$ is small (particles are large) and low when $\beta$ is large (particles are small). These empirical findings are in agreement with Cram et al. (2018), who observed that large particles tend to comprise a larger fraction of the sinking flux where productivity and carbon export are high.

While our analysis does not provide mechanistic insights into the roots of the negative correlation between $\beta$ and export, a plausible explanation for the direction of this relationship is as follows. Low-nutrient conditions select for small phytoplankton with high surface area-to-volume ratios, such that smaller phytoplankton are more abundant in low-nutrient conditions (Litchman et al., 2007). In these nutrient-limited regions of the ocean, productivity and export are also suppressed. Thus, nutrient availability controls both the export rate and the size structure of the phytoplankton community over much of the ocean. Assuming that phytoplankton size in turn controls the size of sinking particles, as suggested by past research (e.g., Guidi et al., 2007; Guidi et al., 2008; Guidi et al., 2009), the availability of nutrients then ultimately controls sinking particle size as well. This potentially explains why small particles (large $\beta$) are associated with reduced export rates and low-nutrient conditions, while large particles (small $\beta$) are associated with increased export rates and high-nutrient conditions."

Line 346:
"...the net effect of phytoplankton selection and particle size-dependent remineralization depths provide a negative feedback on, or dampening of, changes in export, due to the empirically-derived negative relationship between β and export. While we have assumed that phytoplankton community structure is the underlying mechanism linking POC export and particle size, the PSR feedback would operate in the same direction discussed here if another mechanism were ultimately responsible for the empirical negative relationship between these two factors."

Line 673:
"In this study, we used remotely sensed data to show that sinking particle size is empirically correlated with the rate of particulate organic carbon export out of the euphotic zone across the global ocean, such that larger particles tend to dominate when export is high. This empirical relationship between particle size and export likely emerges due to the dependence of both variables on surface nutrient supply. Indeed, nutrient limitation both curtails productivity and selects for smaller phytoplankton that likely aggregate into smaller sinking particles (Litchman et al., 2007; Guidi et al., 2007; 2008; 2009)."

Line 684:
"Regardless of the mechanism linking export and particle size, implementing the empirical relationships between the two in an idealized global biogeochemical model revealed the presence of a negative particle size-remineralization feedback effect that moderates circulation-driven changes in export."

We also moved explanation of the possible linkage between small phytoplankton and small particles from the introduction to the discussion section, where it is now explained as a hypothesized possible explanation for the negative relationships between beta and export, rather than as fact.

Lastly, we changed our title from "Variable phytoplankton size distributions reduce the sensitivity of global export flux to climate change" to "Variable **particle** size distributions reduce the sensitivity of global export flux to climate change."

We agree completely with the reviewer that TEP production or fragmentation would affect the size structure of particles. Lack of resolution of these processes is a limitation of our study. However, data to constrain these processes are limited and adding TEP and fragmentation would make for a substantially more complex model. Our group is currently working on modeling fragmentation in an ongoing project. We added the following caveats to note these limitations at Line 642:

"Furthermore, particle fragmentation—via processes such as zooplankton grazing, microbial degradation, or ocean turbulence (e.g., Cavan et al., 2017; Briggs et al., 2020)—was not included in our model, nor was particle aggregation—via processes such as Transparent Exopolymer Particles production (e.g., Passow, 2002; Mari et al., 2017) or fecal pellet generation (e.g., Steinberg et al., 2012; Turner, 2015 and references therein)."

*Line 97: there's a dawning realisation that Stokes law rarely holds for marine particles e.g. https://aslopubs.onlinelibrary.wiley.com/doi/abs/10.1002/lno.11388 This should be acknowledged here.*

We changed the sentence from:

"Past work has also firmly established a strong positive relationship between particle size and sinking speed in the ocean (Alldredge and Gotschalk, 1988; Smayda, 1971) (although there are exceptions to this rule, particularly in the Southern Ocean – see McDonnell and Buesseler (2010))."

To:

"Past work has **broadly** established **a positive** relationship between particle size and sinking speed in the ocean (Alldredge and Gotschalk, 1988; Smayda, 1971; Iversen and Ploug, 2010)—although there are exceptions to these rules **(Cael and White, 2020; Laurenceau-Cornec et al., 2019)**, particularly in the Southern Ocean (McDonnell and Buesseler, 2010)."

*Line 127: but wouldn't a shift to smaller particles also result in less C sequestration at depth as the C just goes round and round in the upper mesopelagic being readily recycled and re-entrained to surface? Also, Figure 2d – the caption acknowledges that smaller particles leads to greater recycled nutrient supply. This wouldn't increase C sequestration (or CO2 drawdown) as that depends on the resupply of preformed nutrients which isn't affected by the size considerations used here, at least on the timescales considered.*

We agree with the reviewer, and make a similar point in our previous conclusions section. Ocean carbon storage does not just depend on export, but on the sequestration timescale of the exported carbon (Boyd et al., 2019). The particle size feedback helps maintain export, but also results in shallow remineralization and therefore a shorter carbon sequestration timescale. However, our manuscript is solely focused on future changes in carbon export, not ocean carbon storage. Export is a critical process in its own right, even when decoupled from changes in carbon storage, because it is the source of nutrition to the mesopelagic twilight, and therefore determines the productivity of heterotrophic communities, including commercial fisheries. For this reason, carbon export is one of the key variables that is focused on in ocean biogeochemistry forecasts. While changes in the biological pump may also drive changes in ocean carbon storage, these will manifest over longer timescales than changes in export, and will likely be overwhelmed by the effects of anthropogenic CO2 uptake and solubility-driven outgassing. A detailed exploration of changes in carbon storage is therefore beyond the scope of the current paper, and is only discussed in the introduction and conclusions. However, the reviewer's comment has highlighted the fact that we have not drawn the distinction between carbon export, flux at depth, and storage clearly enough. We have revised the manuscript to make this clearer, and incorporated some more detailed discussion of the implications of the PSR feedback for carbon storage in the conclusions section.

Specifically, we added a new Section 3.3.4: Vertical reorganization of POC fluxes induced by the PSR feedback and a new figure (Fig. 9), as follows:

"3.3.4 Vertical reorganization of POC fluxes induced by the PSR feedback

Up to this point, the effect of the PSR feedback effect has been analyzed solely for carbon exported out of the surface euphotic zone (<75 m depth here). We largely focus on export out of the bottom of the euphotic zone rather than on POC fluxes at greater depths because export, by definition, is a measure of the total organic carbon supply that feeds subsurface heterotrophic communities. Thus, the PSR feedback effect buffers the productivity of mesopelagic communities as a whole by damping changes in export.

This buffering of the food supply does not occur uniformly through the water column, however. While shallower particle remineralization helps maintain the nutrient supply to the surface and buffers the POC export rate, it also means that fewer particles persist at depth. Indeed, the PSR feedback on export has the opposite effect on POC fluxes in the lower mesopelagic zone (at 800 m depth, equivalent to 725 m below the bottom of the euphotic zone, for example – Fig. 9a-b). This results in a vertical re-organization of particle fluxes, with more food available in the upper mesopelagic and less in the lower mesopelagic when the PSR feedback is activated. In particular, by 350 m depth (225 m below the bottom of the euphotic zone), the PSR feedback effect on POC flux has changed sign from negative to positive across all ocean regions except for the SAZ, AAZ, and NA (Fig. 9c). In regions where the negative PSR feedback effect is strongest, vertical reorganization is more extreme and the sign of the PSR feedback flips from negative to positive shallower in the water column. The PSR feedback effect becomes positive below depths as shallow as 125, 148, 158, and 165 m (50, 73, 83, and 90 m below the euphotic zone) in the NP, ETP, STA, and IND regions, respectively, for example (Fig. 9c).

A negative PSR feedback effect dampens predicted circulation-driven changes in global export out of the euphotic zone, while a positive PSR feedback effect amplifies predicted changes in lower mesopelagic global POC fluxes. It follows that global models without the PSR feedback effect would overestimate changes in export, but underestimate changes in POC fluxes at deeper depths in a future warming/slowed circulation scenario. This vertical reorganization of POC flux changes brought about by the PSR feedback effect, leading to greater-than-expected fluxes in the upper mesopelagic and lower-than-expected fluxes in the lower mesopelagic under future warming, has the potential to alter follow-on predictions of vertical ecological community organization as well. Importantly, the net effect of the PSR feedback is still a buffering of the total food supply to subsurface heterotrophic communities, however."

We also added the following to Section 3.6 Caveats and future work at Line 657:

"In particular, a decrease in circulation rates should enable enhanced carbon sequestration, as nutrients and CO2 collect in the deep ocean (Fig. 8), but the PSR feedback may potentially moderate this increased sequestration effect by shoaling remineralization and forcing a shorter carbon sequestration timescale. We focused solely on 100-year changes in POC export, to the exclusion of potential longer-term changes in deep ocean carbon storage, because it is a critical energy source energy to the mesopelagic twilight zone and therefore determines the productivity of heterotrophic communities, including commercial fisheries. Furthermore, while changes in the biological pump may also drive changes in ocean carbon storage, these will manifest over longer timescales than changes in export, and will likely be overwhelmed on short timescales by the effects of anthropogenic CO2 uptake and solubility-driven outgassing. A detailed exploration of changes in carbon storage is therefore beyond the scope of the current paper, but could be a fruitful avenue for future work."

*Methods:*

*Line 217: just curious to know why Laws and Dunne estimates of export ratio, rather than others such as Henson et al. 2012 or Siegel et al. 2014 were used*

Good question. We use the Laws and Dunne relationships because Weber et al. (2016) showed that when these algorithms, including Henson et al. (2012) were applied to satellite NPP, they gave the best matches to a range of in situ export estimates based on tracer/mass balance approaches. Seigel et al. (2014) was not used because it provides no simple formula that can be applied to NPP to estimate export. Instead, they use an ecosystem model that makes its own assumptions about grazing, particle size, etc., and the export ratio is an emergent property of the model.

*Line 227: I couldn't find where the in situ observations mentioned here had come from*

We changed the sentence from:

"When reporting most-likely values, we weight the nine map sets according to how well each map set's annual mean export matches in situ observations within each region defined here (Table S3; see Weber et al. (2016) for derivation of weighting factors)."

To:

"When reporting most-likely values, we weight the nine map sets according to how well each map set's annual mean export matches in situ **oxygen and mass balance-based observations (Reuer et al., 2007; Emerson, 2014)** within each region defined here (Table S3; see Weber et al. (2016) for derivation of weighting factors)."

*Line 238-240: is temporal autocorrelation accounted for here? I guess the seasonal cycle in beta and En are similar which will affect the linear regression. Also, are beta and En independent? How are pixels/regions with non statistically significant regressions dealt with?*

For the purposes of this study, autocorrelation should not pose a problem. Autocorrelation only poses a problem if one is trying to determine how much of the relationship between two variables is because they are actually related and how much is just because both are seasonal or slowly varying over time. Seasonality and coincidental variation, from our perspective, both contribute to the relatedness of these two variables and so we want to include both in the calculation of their relationship. Indeed, the seasonal cycle in beta and En greatly affect the linear regression and are an important part of the measured effect/relationship. Their correlation with one another on the shortest available/reasonable timescales is what we were after here, which includes looking at how they vary with one another over months and seasons. We then assume that this relationship that holds on the monthly/seasonal timescale also holds on shorter timescales (i.e., on the timescale in which phytoplankton turn over/change community structure). We also computed regression coefficients using monthly anomalies rather than raw monthly values and got similar values.

Beta and En are independent, but related measurements in that beta is derived from particulate backscattering spectra, while En is based on chlorophyll concentrations (as well as SST and euphotic zone depths). Though both particulate backscattering spectra and chl concentrations are ultimately derived from normalized water leaving radiances observed by SeaWiFS, the ways in which they glean information from these radiances are quite different.

The following further describes how beta is computed. First, the particulate backscattering spectra is computed from the slope connecting particulate backscattering coefficients at 490, 510, and 550 nm. These coefficients are in turn derived from normalized water leaving radiance at these same 3 wavelengths. To convert from a particulate backscattering spectra to beta, Mie modeling is used to establish a physical relationship/lookup table between the two variables ($\eta$ and $\xi$ in Kostadinov et al. (2009)).

Chlorophyll concentrations, on the other hand, are typically computed as follows: [chl] = $10^{\wedge}$(a + b*R + c*R^2 + d*R^3 + e*R^4), where R = log10(maximum normalized water leaving radiance ratio out of 443 nm:555 nm, 490 nm:555 nm, and 510 nm:555 nm), and a-e are constants.

We average over the larger regions in order to avoid generating/using only single grid cells with insignificant regressions. That is, avoidance of insignificant and spatially over-resolved/over-specified regression coefficients was the primary reason for averaging over regions.

*Line 266: in reality I suspect dbetaobs/dEnobs could vary seasonally. Might be worth a caveat on that point in the discussion? Actually, I suspect that some of the strange behaviour in the SAZ might be due to a seasonal effect or a time lag between changes in phytoplankton size and particle export. It would be helpful to the reader to include some example annual time series of a region showing the PP, export and beta to illustrate how they interact. It would also inform on potential time lags between PP, export and beta.*

As was discussed above, we in fact use the variability over the seasonal cycles of beta and En to see how they change in concert over the seasons. Again, this is part of their computed relationship, encapsulated in dbetaobs/dEnobs. We thus compute the relationship between beta and export by in effect exploiting their seasonal variations.

We added the following text at Line 276, as well as Supplementary Figures 4 and 5, to illustrate this:

"The vast majority of variance in both β and export occurs over seasonal (rather than interannual or longer) timescales, and therefore the coincident seasonal cycles of β and export account for much of the relationship between the two variables (Fig. S3-4)."

*Results and Discussion:*

*A point that should be acknowledged somewhere is that the results presented here are of course still dependent on the details of the model parameterisation and choices.*

Good point.

We added the following section of text to illustrate the many parameterizations and choices that were part of creating our model:

"2.1.1 Model setup

[revised manuscript text omitted]

We also broke out discussion of caveats into a longer, more detailed separate section of text: Section 3.6: Caveats and future work.

In this section at Line 642, we added:
"Furthermore, particle fragmentation—via processes such as zooplankton grazing, microbial degradation, or ocean turbulence (e.g., Cavan et al., 2017; Briggs et al., 2020)—was not included in our model, nor was particle aggregation—via processes such as Transparent Exopolymer Particles production (e.g., Passow, 2002; Mari et al., 2017) or fecal pellet generation (e.g., Steinberg et al., 2012; Turner, 2015 and references therein)."

We also added the following at Line 667:
"Other remineralization depth-related feedbacks induced by changes in temperature, oxygen, particle density, and mineral ballasting (among others) not studied here may also be important for modulation of future changes in carbon export and its downstream effects. Ensuring that the PSR and other remineralization feedbacks are adequately represented in ESMs should be a priority of the modeling community to enable robust predictions of carbon export fluxes in the future ocean."

*Line 301-303: references needed here*

We added the missing reference to Bopp et al. (2013).

*Line 322: isn't this 21% rather than 18%? I found the use of the word "visually" here and on line 307 confusing. It made me think that you had estimated the values by eye rather than calculating them. I suggest just dropping 'visually'.*

The 18% reduction was calculated as follows: (export increase w/ feedback - export increase w/o feedback) / (export increase w/o feedback) = (0.23 - 0.28 molC/m^2/yr) / (0.28 molC/m^2/yr) = -17.9%.

We thank the reviewer for pointing out that this is confusing - we dropped the word "visually."

*Line 346: some C:P ratio must be used here too? Couldn't find where that was mentioned. Does this formulation also assume that all nutrients supplied are regenerated? I think it assumes that all nutrients supplied are turned into PP (which is fine in nutrient limited regions), and then are all exported i.e. and e-ratio of 1?*

We do not assume that all nutrients supplied are regenerated. Nutrient concentrations are the sum of preformed and regenerated nutrients. We added a note of this at Line 111, as follows:

"Nutrient concentrations in the ocean interior represent the sum of preformed nutrients, transported from regions of incomplete utilization in the ocean surface, and the accumulated product of particulate and dissolved organic matter remineralization."

We also added the following at Line 148:

"This formulation successfully reproduces the broad spatial patterns of surface [PO43-] (Weber and Deutsch, 2012), suggesting that our model accurately captures the balance between preformed and remineralized nutrients in the ocean interior."

Our existing text in Section 2.1.1 discussed the other points, as follows:

"This scheme calculates phytoplankton growth rates as a function of observed annual-mean temperatures (Locarnini et al., 2010) and solar radiation levels (Rossow & Schiffer, 1999), along with modeled $PO_4^{3-}$. We thus explicitly model phytoplankton production in terms of phosphorus consumption and regeneration. We then use an empirical, spatially variable relationship between particulate C-to-P ratios and phosphate concentrations (Galbraith & Martiny, 2015) to convert phytoplankton production into units of carbon. It is assumed that 10% of phytoplankton production is routed directly to dissolved organic matter in the euphotic zone, with the remainder becoming particulate organic matter (Thornton, 2013)."

*Line 376-378: I'm not sure this "visually" statement helps the reader's understanding here*

We dropped the word "visually."

*Line 415-426: specify the direction of +ve/-ve changes in the caption. At the moment it's a bit confusing as +ve indicates a reduction*

Great idea. We added the following to the caption of now Figure 10:

"A feedback strength above 0 indicates a negative PSR feedback effect (that is, a reduction in export change when the PSR feedback is applied)."

*Figure 1a: specify in the caption that higher beta values = smaller plankton (or mark with arrows on the colour bar)*

Great idea. We added the following to the caption of Figure 1:

"Larger values of β are associated with smaller particles, while smaller values of β are associated with larger particles."

*Figure 6a: rather than having the right y-axis in remin depth x 100, just write out the numbers in full – it's clearer*

We thank the reviewer for this helpful attention to detail. We changed the labels as requested.

*Figure S1: specify in the caption or legend that higher beta values = smaller plankton*

We added the following to the caption of Figure S1:

"Larger β values correspond with smaller particles."

*Figure S2: add the key to the PP and e-ratio model abbreviations into the caption here*

Good point. We added the following to the caption of Figure S2:

"NPP algorithm key: VGPM = the Vertically Generalized Production Model (VGPM) (Behrenfeld & Falkowski, 1997); VGPME = the Eppley-VGPM model (Carr et al., 2006); CbPM = the Carbon-based Production Model (Behrenfeld et al., 2005). E-ratio algorithm key: L2000 = Laws et al. (2000); D2005 = Dunne et al. (2005); L2011 =  Laws et al. (2011)."

*Table S1: Define the parameter names in this table.*

We added the parameter names.

**Response to Reviewer 2**

In the paragraphs below, all reviewer comments will be italicized, while author responses will be in normal font. All changes to the manuscript described below are highlighted in yellow in the corresponding marked up manuscript.

*The authors apply a global biogeochemical model to examine the effect of variable particle (phytoplankton) size distribution on surface and subsurface nutrients, and their mutual feedbacks when nutrients are supplied under different physical forcings. The feedback effect of the (nutrient-dependent) size distribution and subsequent particle sinking and remineralisation dampens the model response to changes in physics. I find this manuscript generally well written. The authors do a great job in explaining the mechanisms involved. In general, the experimental design to disentangle the effects of circulation, ecology and sinking is clear and well justified. Thus, the manuscript provides valuable new insights into a potential negative feedback mechanism in global biogeochemical models. However, I have a few points that I think could be improved with regard to model description and its critical discussion.*

We thank the reviewer for their positive comments about our experimental design, and for their constructive criticism below, which we believe helped greatly improve the manuscript.

*(1) Model description: I recommend to describe the biogeochemical model, particle sinking and remineralisation in detail (including equations), and also explain its basic assumptions. As far as I understand, the model assumes a power law size distribution of particles at the surface; particles then sink depending on their size, and remineralise with a size independent rate. Therefore, the particle size distribution changes with depth, favouring large particles as depth increases, similar to the 1D approach presented by Kriest and Oschlies (2008). (In fact, there seem to be only small differences between both models, in terms of formulation and results.) Both approaches make quite strong implicit assumptions about constant individual particle properties, which do not change with time or depth. In particular, the models neglect any processes besides sinking and remineralisation that might affect the particle size distribution below the euphotic zone, such as particle breakup, reworking by zooplankton (e.g., flux feeding, formation of fecal pellets), particles becoming more or less porous because of bacterial degradation, etc.. Of course, one cannot address all details and complications at once especially in global models; but describing the current implicit model assumptions in detail would help the reader to understand how the model works, and what its limitations and merits might be.*

We thank the reviewer for pointing out that the model description falls short. The mathematical description of the PRiSM model has been fully laid out in previous references and cannot be repeated in full here. However, we agree that enough information needs to be provided to allow the reader to understand how the model "works", without extensive cross-referencing to previous papers. We revise the methods section to expand the description of the PRiSM model, as follows:

"2.1.1 Model setup

[revised manuscript text omitted]

*(2) Model description: The description of the model and its general setup is somehow unclear about how the phytoplankton size distribution might be related to larger particles, which likely contribute most to mesopelagic and deep particle flux. For example, the work by Kostadinov et al (2009), from which the observed size distribution at the surface is taken, is based on phytoplankton, i.e. extends only to a size of ca. 50 um. However, the present model applies a size range of 20-2000 um (Table S1). Moreover, the model parameters sometimes seem to relate to phytoplankton properties (e.g., the exponent of eta=1.17 relating cell diameter to sinking speed is based on phytoplankton data by Smayda, 1971), whereas other relate more to porous aggregates (e.g., the exponent relating particle mass to size of zeta=1.62; see also Kriest, 2002 http://dx.doi.org/10.1016/S0967-0637(02)00127-9, and citations therein). Again, here it would be useful to present and discuss these basic model assumptions. This is done partly on page 3, yet I think this subsection could be improved (see below, my comments Lines 80ff, 82ff, 95). In summary, I would suggest to more clearly distinguish between phytoplankton and particle size distribution, and to address potential connections between these more comprehensively.*

We assume that the particle size distribution slope computed by Kostadinov et al. (2009) continues to hold for particles larger than those they explicitly compute the slope for. Prior research backs up this assumption (e.g., Durkin et al., 2015). We added the following note at Line 189 to discuss this:

"Although $\beta$ from Kostadinov et al. (2009) is computed only over particle sizes ranging from 0.002 to 63 um, we assume that the same $\beta$ continues to hold for larger particles up to 2000 um (the largest particle size in PRiSM), as supported by prior research (e.g., Durkin et al., 2015). Ideally, measurements of $\beta$ would be computed over the same particle size range as simulated in PRiSM (20−2000 um); however, such a dataset was not readily available. Indeed, the Kostadinov et al. (2009) observations of $\beta$ were the only readily available measurements spanning long enough timescales, with high enough spatiotemporal resolution to compute the relationships between $\beta$ and POC export needed for this study."

We assume that phytoplankton simply behave as smaller particles. We added a note of this in the text at Line 115 (also quoted above from Section 2.1.1):

"PRiSM computes particle flux profiles as a function of particle size distribution at the surface, microbial remineralization rate, and empirical relationships between particle size, mass, and sinking velocity. These empirical relationships are in some cases derived from measurements of sinking phytoplankton and in other cases from those of sinking particles or porous aggregates. PRiSM therefore implicitly assumes that phytoplankton and smaller particles behave similarly as they sink down the water column."

The driving mechanism behind the particle size feedback is the relationship between export production and particle size, which we determine empirically from remote sensing data in our study. Our model setup simply computes particle size based on this empirical relationship to export, and makes no explicit assumption about the root cause of the relationship. We hypothesize that the export/particle size relationship arises from plankton community structure simply because this seems like an intuitive mechanism, and is supported by correlative evidence: large particles and large phytoplankton taxa are both generally more dominant in regions of high productivity and export (e.g., Cram et al., 2018), and we therefore find it reasonable to assume that large phytoplankton aggregate (either directly or by grazing) into large particles. However, this needn't be true for the particle size feedback to hold. Any other mechanism that gives rise to the observed export/particle-size relationship would give rise to the same feedback. In the revised manuscript, we are more careful about distinguishing between the explicit assumptions and relationships "baked in" to our model, and the mechanisms that we are hypothesizing give rise to those relationships.
We added the following text to address these points:

Line 74:
"Potential mechanisms that could drive changing particle sizes include changes in underlying phytoplankton community structure and organic matter packaging processes at higher trophic levels. Whatever the mechanism, the direction and magnitude of the particle size-remineralization feedback in a warming ocean will depend on how particle sizes change as export declines in the future. If the export decline is associated with a shift towards larger organic particles that sink more quickly, remineralization depths will deepen and further reduce surface nutrient supply and export in a positive feedback. If, on the other hand, export decreases are associated with a shift towards smaller sinking particles, shallower remineralization will allow faster nutrient recirculation to the surface and dampen stratification-driven decreases in nutrient supply and export in a negative feedback."

Line 159:
"When the PSR feedback is disabled within our model, circulation-driven changes in the nutrient supply to the euphotic zone (see Section 2.3) will lead to changes in POC export, but β (and therefore particle remineralization depths) remains constant over time. With the PSR feedback enabled, any change in POC export is accompanied by a change in β, the direction and magnitude of which is specified using the empirical relationships discussed in Section 2.2. We note that by design, this modeling approach makes no assumptions about the mechanisms driving shifts in the particle size distribution, only that β changes in tandem with POC export, in a manner that is consistent with observations."

Line 278:
"Because β and export are negatively correlated, export tends to be high when β is small (particles are large) and low when β is large (particles are small). These empirical findings are in agreement with Cram et al. (2018), who observed that large particles tend to comprise a larger fraction of the sinking flux where productivity and carbon export are high.
        While our analysis does not provide mechanistic insights into the roots of the negative correlation between β and export, a plausible explanation for the direction of this relationship is as follows. Low-nutrient conditions select for small phytoplankton with high surface area-to-volume ratios, such that smaller phytoplankton are more abundant in low-nutrient conditions (Litchman et al., 2007). In these nutrient-limited regions of the ocean, productivity and export are also suppressed. Thus, nutrient availability controls both the export rate and the size structure of the phytoplankton community over much of the ocean. Assuming that phytoplankton size in turn controls the size of sinking particles, as suggested by past research (e.g., Guidi et al., 2007; Guidi et al., 2008; Guidi et al., 2009), the availability of nutrients then ultimately controls sinking particle size as well. This potentially explains why small particles (large β) are associated with reduced export rates and low-nutrient conditions, while large particles (small β) are associated with increased export rates and high-nutrient conditions."

Line 346:

"...the net effect of phytoplankton selection and particle size-dependent remineralization depths provide a negative feedback on, or dampening of, changes in export, due to the empirically-derived negative relationship between β and export. While we have assumed that phytoplankton community structure is the underlying mechanism linking POC export and particle size, the PSR feedback would operate in the same direction discussed here if another mechanism were ultimately responsible for the empirical negative relationship between these two factors."

Line 673:
"In this study, we used remotely sensed data to show that sinking particle size is empirically correlated with the rate of particulate organic carbon export out of the euphotic zone across the global ocean, such that larger particles tend to dominate when export is high. This empirical relationship between particle size and export likely emerges due to the dependence of both variables on surface nutrient supply. Indeed, nutrient limitation both curtails productivity and selects for smaller phytoplankton that likely aggregate into smaller sinking particles (Litchman et al., 2007; Guidi et al., 2007; 2008; 2009)."

Line 684:
"Regardless of the mechanism linking export and particle size, implementing the empirical relationships between the two in an idealized global biogeochemical model revealed the presence of a negative particle size-remineralization feedback effect that moderates circulation-driven changes in export."

We also moved explanation of the possible linkage between small phytoplankton and small particles from the introduction to the discussion section, where it is now explained as a hypothesized possible explanation for the negative relationships between beta and export, rather than as fact.

Lastly, we changed our title from "Variable phytoplankton size distributions reduce the sensitivity of global export flux to climate change" to "Variable **particle** size distributions reduce the sensitivity of global export flux to climate change."

*(3) Experimental setup: To me it is not clear how the circulation was reduced (e.g., Lines 253-254 "To simulate increased water column stratification and reduced vertical exchange due to warming, we uniformly and instantaneously reduce circulation and diffusion rates by 10% throughout the ocean.") - I would appreciate a more in depth explanation.*

Our model uses the Transport Matrix Method, in which all physical fluxes (advection and diffusion) of element X are represented by the matrix-vector product A*X, in which A is the mass-conserving transport matrix that quantifies the mass exchanges between every gridcell in the model. We change circulation rates in an idealized way, simply by multiplying A by a factor of 0.9 (a 10% reduction in circulation rates) or 1.1 (a 10% increase in circulation rates). Therefore, the patterns of circulation remain unchanged, but the absolute exchange rates are scaled up or down.

We added the following explanations at Line 102 (also quoted above from Section 2.1.1):

"OCIM assimilates passive and transient tracer data to generate an annual-mean circulation that realistically reproduces water mass distributions and ventilation rates at 2-degree horizontal resolution on 24 vertical layers. The circulation rates are stored in a transport matrix (A), that quantifies physical exchanges between every grid cell in our model. Thus, all physical (advective and diffusive) fluxes of tracer X in our model are represented by the matrix-vector product A*[X]."

And at line 244:

"To simulate increased water column stratification and reduced vertical exchange due to warming in an idealized way, we uniformly and instantaneously reduce circulation and diffusion rates by 10% throughout the ocean (i.e. we multiply the tracer transport matrix A by 0.9, such that circulation patterns remain unchanged but the absolute exchange rates between all grid cells are scaled down by 10%)."

*(5) Discussion: The model shows a large response and differences between the two setups (with or without PSR) in the equatorial upwelling regions. However, especially models of coarser resolution tend to suffer from an insufficient representation of the equatorial current system, with possible consequences for the representation of nutrients and/or oxygen (e.g., Dietze and Loeptien, 2013, https://doi.org/10.1002/gbc.20029; Duteil et al., 2014, https://doi.org/10.1029/2011GL046877). I would suggest to add some discussion on these potential effects.*

Great point. Coarse resolution dynamical models do tend to represent equatorial regions poorly. However, our transport matrix is derived from the observationally-constrained Ocean Circulation Inverse Model, which assimilates passive and transient water mass and ventilation tracers. Thus, even though the resolution does not allow accurate simulation of equatorial currents from a dynamical perspective, the data-assimilation ensures that the net effect of these currents on tracer transport is realistic. The model has been used successfully for simulation of nutrients (DeVries, 2014) and oxygen (DeVries and Weber, 2017), and does not suffer from the equatorial biases often evident in coarse resolution models.

We added the following at Line 106 (also quoted above from Section 2.1.1):

"OCIM has previously been used successfully for high-fidelity simulation of nutrients (DeVries, 2014) and oxygen (DeVries and Weber, 2017) and does not suffer from the equatorial biases often evident in dynamical models with the same resolution."

*(4) Discussion: Section 3 is named "Results and discussion", yet it almost entirely presents the results. In contrast, Section 4 is named "Conclusions", but partly discusses the results before the background of other works, is quite long and partly repetitive. I would suggest to rename section 3 to "Results", add a "Discussion" section, that extends a bit on the comparison of results obtained here with other model studies and also includes a critical discussion of model processes and properties. The "Conclusions" section could then be shortened and more concise.*

We thank the reviewer for this helpful suggestion. We made major changes to the structure of the manuscript as follows:

- Moved previous introductory material discussing potential linkages between phytoplankton and particle size to Section 3.2: An empirical negative global particle size-remineralization (PSR) feedback (a subsection under Section 3 Results and Discussion)
- Added a Section 3.5: Comparison with CMIP5 models
- Created a new section just for discussion of caveats and future work (Section 3.6: Caveats and future work)
- Shortened/removed discussion from the conclusions section and renamed it "Summary" (Section 4: Summary)

*Specific comments:*

*- Lines 35-37: "Where sinking POC fluxes are particularly high, enhanced bacterial breakdown of particles can deplete available oxygen and create hypoxic or even suboxic conditions [...] " - there are many places in the ocean where sinking POC fluxes are high; another necessary condition for the development of OMZs is that supply of oxygen by physical transport is low.*

We changed the statement from:

"Where sinking POC fluxes are particularly high, enhanced bacterial breakdown of particles can deplete available oxygen and create hypoxic or even suboxic conditions…"

To:

"Where sinking POC fluxes are particularly high **and supply of oxygen via physical transport is low,** enhanced bacterial breakdown of particles can deplete available oxygen and create hypoxic or even suboxic conditions…"

*- Line 66: Note that there are further global ocean models that address spatial and temporal variation of the size distribution (of marine aggregates) and sinking speed, e.g., Schwinger et al. (2016, www.geosci-model-dev.net/9/2589/2016/) and Niemeyer et al. (2019, https://doi.org/10.5194/bg-16-3095-2019). On a local (1D) scale, even very complex models of particle transformations have been developed (Jokulsdottir and Archer, 2016, www.geosci-model-dev.net/9/1455/2016/).*

We thank the reviewer for bringing these highly relevant and useful studies to our attention. We added citations of these studies at Line 85 as follows:

"More complex models that resolve aggregation-disaggregation transformations and/or particle size distributions have been developed (Gehlen et al., 2006; Jokulsdottir and Archer, 2016; Kriest & Oschlies, 2008; Niemeyer et al., 2019; Schwinger et al., 2016), but have not been used to examine the interactions between climate change, particle size, and export production."

*- Lines 80ff: "Large particles tend to exist in the ocean where larger microphytoplankton (>20 um in diameter) are dominant, while relatively small particles tend to exist where smaller picophytoplankton (<2 um in diameter) are dominant (Guidi et al., 2007; Guidi et al., 2008; Guidi et al., 2009). [...]" - The observations by Guidi et al. (2007, 2008), are based on UVP data of large particles (aggregates, fecal pellets, ...), of a size of at least 250 um. Therefore, I don't think that*

*these observations can be used to justify the assumptions about the phytoplankton size distributions made in this paper.*

Yes, it is true that UVP data is measuring larger particles. However, our model does include some particles of these sizes, as the reviewer noted above. Furthermore, again, we assume that the particle size distribution slope holds throughout the entire range of particle sizes in the ocean. See our response to the "(2) Model description" comment above for the clarifications we added to our manuscript regarding the linkage between small phytoplankton and small particles.

*- Lines 82ff: "The presence of large phytoplankton leads to the generation of larger particles perhaps because large phytoplankton are more likely to form aggregates and be transformed into large fecal pellets by large zooplankton, whereas small phytoplankton are more likely to be degraded by bacteria and consumed by smaller zooplankton (Bopp et al., 2005; Guidi et al., 2007; Guidi et al., 2009; Michaels and Silver, 1988). The exact mechanisms governing the processes by which smaller and larger phytoplankton become smaller and larger particles are not clearly known, however, and is an active area of research." - The global model study by Bopp et al. does not address aggregates; moreover, as a model study it is based on a priori assumptions, and does not provide insight into real in situ mechanisms. As noted above, the study by Guidi et al. (2007) addresses the UVP size range and the study by Michaels et al. is also a (food web) model. While I tend to agree with the idea that large phytoplankton triggers large sinking particles, I would appreciate a more convincing reasoning why one can extend the phytoplankton size range up to particles 2 millimeters in diameter.*

We assume that the particle size distribution slope computed by Kostadinov et al. (2009) continues to hold for particles larger than those they explicitly compute the slope for. Prior research backs up this assumption (e.g., Durkin et al., 2015). Furthermore, Kostadinov et al. (2009)'s particle size distribution slope is available globally and is also temporally and spatially resolved. Thus, this was really the best PSD slope dataset that we could find. In essence, it was what was available. Our method really requires this kind of global data available over long enough timescales and with enough spatiotemporal resolution to compute the necessary correlations.

We added the following note at Line 189 to discuss this:

"Although $\beta$ from Kostadinov et al. (2009) is computed only over particle sizes ranging from 0.002 to 63 um, we assume that the same $\beta$ continues to hold for larger particles up to 2000 um (the largest particle size in PRiSM), as supported by prior research (e.g., Durkin et al., 2015). Ideally, measurements of $\beta$ would be computed over the same particle size range as simulated in PRiSM (20–2000 um); however, such a dataset was not readily available. Indeed, the Kostadinov et al. (2009) observations of $\beta$ were the only readily available measurements spanning long enough timescales, with high enough spatiotemporal resolution to compute the relationships between $\beta$ and POC export needed for this study."

Additionally, to reiterate our points above, though we do implicitly assume that small phytoplankton = small particles to explain/understand the results of our empirical analyses, our model setup and study in general do not require this relationship to be true. Our model setup simply computes particle size as an empirical function of export. The empirical positive relationship between export and particle size illuminated by our satellite data analysis showed that increasing productivity and export are associated with larger particle sizes. Increasing productivity is in turn associated with larger phytoplankton; thus, based on our empirical analysis, it seems that larger phytoplankton are associated with larger particle sizes. We bring up other studies that find or assume this merely to better explain the mechanistic underpinnings of this result and NOT to say that this must be the case for our model setup to hold. Our empirical analyses could have shown that export and particle size were negatively correlated, for example, in which case, our model would have demonstrated a positive particle size remineralization feedback, rather than a negative one. Thus, our results/findings of a negative particle size remineralization feedback effect hinge entirely on the direction of the empirical relationship between export and particle size, rather than on our own assumptions. See our response to the "(2) Model description" comment above for the clarifications we added to our manuscript regarding the linkage between small phytoplankton and small particles.

*- Line 95: "Past work has also firmly established a strong positive relationship between particle size and sinking speed in the ocean (Alldredge and Gotschalk, 1988; Smayda, 1971) [...]" - The relationship between diameter and sinking speed in Alldredge and Gotschalk (1988) is w=50 dˆ0.26, and shows considerable scatter. I would not call this a strong relationship. This weak relationship is possibly because of the fractal and variable nature of aggregates - indeed, single cells show a higher exponent (Smayda, 1971). Again, here I would suggest to more clearly distinguish between aggregates and single phytoplankton cells.*

We changed the sentence from:

"Past work has also firmly established a strong positive relationship between particle size and sinking speed in the ocean (Alldredge and Gotschalk, 1988; Smayda, 1971) (although there are exceptions to this rule, particularly in the Southern Ocean – see McDonnell and Buesseler (2010))."

To:

"Past work has **broadly** established **a positive** relationship between particle size and sinking speed in the ocean (Alldredge and Gotschalk, 1988; Smayda, 1971; Iversen and Ploug, 2010)—although there are exceptions to these rules **(Cael and White, 2020; Laurenceau-Cornec et al., 2019)**, particularly in the Southern Ocean (McDonnell and Buesseler, 2010)."

*- Line 100: "by a factor of e" - e is 2.718, do you really mean a factor of e?*

Yes, this is the e-folding length scale.

*- Line 157: "physical relationships between particle size, mass, and sinking velocity"*
*- I don't think that the relationship between particle size, mass and sinking velocity of organic particles is a purely physical one; at least the relationships by Smayda (1971) and Alldredge and Gotschalk (1988) are empirical. I suggest to skip "physical".*

We changed the sentence from:

"PRiSM computes particle flux profiles as a function of particle size distribution (β) at the surface, microbial remineralization rates, and physical relationships between particle size, mass, and sinking velocity."

To:

"PRiSM computes particle flux profiles as a function of particle size distribution (β) at the surface, microbial remineralization rate, **and empirical relationships** between particle size, mass, and sinking velocity."

*- Line 184-185: "time-mean export" - mean over what time? A year?*

We changed the statement from:

"...time-mean normalized export (En,obs) (i.e., absolute export divided by time-mean export at a given grid point)."

To:

"...time-mean normalized export (En,sat, **defined** as absolute export divided by time-mean export **calculated between 1997 and 2010** at a given grid point)."

*- Line 483-484: "This implies that global models without the PSR feedback may be overestimating 100-year climate-driven export decreases by ~1.16 times." - What is meant with 1.16 times?*

We changed the sentence from:

"This implies that global models without the PSR feedback may be overestimating 100-year climate-driven export decreases by ~1.16 times."

To:

"This implies that **in isolation of other mechanisms**, **ESMs** without the PSR feedback may be **projecting 100-year climate-driven export decreases that are ~1.16 times too large.**"

In the paragraphs below, all reviewer comments will be italicized, while author responses will be in normal font. All changes to the manuscript described below are highlighted in yellow in the corresponding marked up manuscript.

*Leung et al. present an interesting study in which they thoroughly analyse the "particle size remineralization" (PSR) feedback in a simple 3D biogeochemical model. The PSR feedback is described by decreased circulation leading to increased nutrient limitation at surface, which in turn leads to plankton communities with smaller sizes producing smaller sinking particles, leading to a slower sinking speed and hence remineralization at shallower depths. This chain of mechanisms increases nutrient concentration at these shallower depths, dampening the decreased nutrient supply to the surface. Their results show that the PSR feedback dampens projected decrease in export production by 14% globally and also details the dampening in different regions. In particular the detailed regional analysis is very informative, suggesting that changes in particle sinking speed in the Southern Ocean have only a small impact on carbon export. Overall I find this an important and interesting study, well written and clearly presented and I recommend a speedy publication after the following comments have been addressed.*

We thank the reviewer for their constructive comments and appreciate the positive response to our manuscript.

*Main comments:*

*- I like that the PSR feedback has been thoroughly analyzed, but it should be discussed or mentioned earlier that there are several other processes potentially affecting the nutrient supply to the surface (varying stoichiometry, temperature dependence of remineralization, ballasting, aggregation/fragmentation etc) that are ignored in this study. There is a good discussion of the caveats at the very end in the conclusions, but I think it would help to mention early on that this paper analyses the PSR feedback in isolation of the other feedbacks.*

The reviewer makes a good point about structure.

We added the following to the introduction:

"This raises the possibility of feedback loops in which changes in particle remineralization depth might either dampen (negative feedback) or enhance (positive feedback) circulation-driven decreases in primary production and export. For instance, increasing ocean temperatures may speed up bacterial remineralization rates (Cavan et al., 2019; Cram et al., 2018; John et al., 2014; Laufkötter et al., 2017; Marsay et al., 2015; Matsumoto, 2007) and enhance recycling of nutrients near the surface, which would dampen physically-driven decreases in surface nutrient concentrations and result in a negative feedback on export. Oxygen concentrations, on the other hand, are predicted to decrease with future warming (Bopp et al., 2002; Cabré et al., 2015b; Keeling et al., 2010; Long et al., 2016; Matear & Hirst, 2003; Schmidtko et al., 2017) and slow bacterial remineralization and zooplankton-mediated particle disaggregation rates (Cavan et al., 2017; Devol & Hartnett, 2001; Hartnett & Devol, 2003; Laufkötter et al., 2017; Van Mooy et al., 2002); this would result in deeper particle remineralization and further exacerbate circulation-driven nutrient supply decreases, leading to a positive feedback on export production. A decrease in mineral ballasting and protection of particles with ocean acidification may also feedback negatively on export decreases by shoaling remineralization depths (Hofmann and Schellnhuber, 2009). Future changes in sinking particle size may also lead to strong feedbacks on export."

We added the following to Section 3.6: Caveats and future work:

"Other remineralization depth-related feedbacks induced by changes in temperature, oxygen, particle density, and mineral ballasting (among others) not studied here may also be important for modulation of future changes in carbon export and its downstream effects. Ensuring that the PSR and other remineralization feedbacks are adequately represented in ESMs should be a priority of the modeling community to enable robust predictions of carbon export fluxes in the future ocean."

*- The proposed PSR feedback has been analysed for carbon export through 100m depth, however I would expect that there is a decrease in POC flux at deeper layers in the ocean (as the PSR feedback mainly acts in the first few 100m? My interpretation of course). For instance for the question how much carbon is sequestered from the atmosphere or how much food supply is available for mesopelagic organisms, the carbon flux below 100m depth might be of higher relevance. It would be very interesting to see how much the carbon flux at deeper layers is dampened by the PSR feedback.*

This is a great point, and something we discussed at length when we were writing this manuscript. We elected to constrain the scope of this manuscript to exploring export, rather than carbon sequestration. An analysis of sequestration is a clear direction for future work, either by our group or by another. We expanded our discussion of what this PSR feedback might mean for carbon sequestration in the conclusion as follows:

"In particular, a decrease in circulation rates should enable enhanced carbon sequestration, as nutrients and CO2 collect in the deep ocean (Fig. 8), but the PSR feedback may potentially moderate this increased sequestration effect by shoaling remineralization and forcing a shorter carbon sequestration timescale. We focused solely on 100-year changes in POC export, to the exclusion of potential longer-term changes in deep ocean carbon storage, because it is a critical energy source energy to the mesopelagic twilight zone and therefore determines the productivity of heterotrophic communities, including commercial fisheries. Furthermore, while changes in the biological pump may also drive changes in ocean carbon storage, these will manifest over longer timescales than changes in export, and will likely be overwhelmed on short timescales by the effects of anthropogenic CO2 uptake and solubility-driven outgassing. A detailed exploration of changes in carbon storage is therefore beyond the scope of the current paper, but could be a fruitful avenue for future work."

The reason why we focus on export out of the bottom of the euphotic zone and not carbon flux at some deeper depth is because export, by definition, is a measure of the total food supply to subsurface heterotrophic communities, and so the PSR feedback does buffer the productivity of mesopelagic communities as a whole, by damping changes in export. It's true that the feedback has the opposite effect on deeper fluxes (say at 500m), and this results in a vertical re-organization of the food supply, with more food available in the upper mesopelagic and less in the lower mesopelagic. But the net effect is still a buffering of the total food supply from the surface ocean. We added explanation of this with a new figure (Fig. 9) and a new section of text, Section 3.3.4: Vertical reorganization of POC fluxes induced by the PSR feedback, as follows:

"3.3.4 Vertical reorganization of POC fluxes induced by the PSR feedback

Up to this point, the effect of the PSR feedback effect has been analyzed solely for carbon exported out of the surface euphotic zone (<75 m depth here). We largely focus on export out of the bottom of the euphotic zone rather than on POC fluxes at greater depths because export, by definition, is a measure of the total organic carbon supply that feeds subsurface heterotrophic communities. Thus, the PSR feedback effect buffers the productivity of mesopelagic communities as a whole by damping changes in export.

This buffering of the food supply does not occur uniformly through the water column, however. While shallower particle remineralization helps maintain the nutrient supply to the surface and buffers the POC export rate, it also means that fewer particles persist at depth. Indeed, the PSR feedback on export has the opposite effect on POC fluxes in the lower mesopelagic zone (at 800 m depth, equivalent to 725 m below the bottom of the euphotic zone, for example – Fig. 9a-b). This results in a vertical re-organization of particle fluxes, with more food available in the upper mesopelagic and less in the lower mesopelagic when the PSR feedback is activated. In particular, by 350 m depth (225 m below the bottom of the euphotic zone), the PSR feedback effect on POC flux has changed sign from negative to positive across all ocean regions except for the SAZ, AAZ, and NA (Fig. 9c). In regions where the negative PSR feedback effect is strongest, vertical reorganization is more extreme and the sign of the PSR feedback flips from negative to positive shallower in the water column. The PSR feedback effect becomes positive below depths as shallow as 125, 148, 158, and 165 m (50, 73, 83, and 90 m below the euphotic zone) in the NP, ETP, STA, and IND regions, respectively, for example (Fig. 9c).

A negative PSR feedback effect dampens predicted circulation-driven changes in global export out of the euphotic zone, while a positive PSR feedback effect amplifies predicted changes in lower mesopelagic global POC fluxes. It follows that global models without the PSR feedback effect would overestimate changes in export, but underestimate changes in POC fluxes at deeper depths in a future warming/slowed circulation scenario. This vertical reorganization of POC flux changes brought about by the PSR feedback effect, leading to greater-than-expected fluxes in the upper mesopelagic and lower-than-expected fluxes in the lower mesopelagic under future warming, has the potential to alter follow-on predictions of vertical ecological community organization as well. Importantly, the net effect of the PSR feedback is still a buffering of the total food supply to subsurface heterotrophic communities, however."

*- There are models participating in CMIP5/CMIP6 that parameterize two different particle size classes with different sinking speeds (I have written a bit more in the line-by-line comments). Is the full particle size spectrum really needed for the PSR feedback, or is it also possible to use just two particle size classes?*

This is a great question, but not one we can definitively answer in the current study. We believe that a model like ours that resolves a wide-ranging particle size spectrum is most likely to accurately capture the feedback, especially given that the export/particle-size relationship can be directly constrained with remotely-sensed spectral observations. Models with different particle size classes, parameterized with different sinking speeds, could potentially qualitatively reproduce the same feedback (given that the other pieces of the feedback are also in place), but it is not clear whether the magnitude of the feedback would be accurately captured. Furthermore, in the CMIP5 models, the relative proportion of large and small particles is determined by internal model dynamics, and not prescribed empirically as in our model. Their ability to reproduce the feedback would therefore depend on the degree to which the observed relationship between export and particle-size emerges from the ecosystem model. We hope that in future work, our study can provide a baseline against which ESMs can assess their ability to reproduce particle size feedbacks. We added a new section, Section 3.5: Comparison with CMIP5 models, to discuss these points, as well as to better describe the CMIP5 models, as follows:

"3.5 Comparison with CMIP5 models

In the absence of the PSR feedback, our model predicts a 100-year global mean export decrease of 8.1% (0.29 molC m-2 yr-1). With the PSR feedback on, this export decrease is reduced to 7.0% (0.25 molC m-2 yr-1). Meanwhile, CMIP5 models project global mean export decreases of around 7-18% between 2090-2099 and 1990-1999 under a "business-as-usual" radiative forcing scenario (RCP8.5), with an ensemble mean of 13% (Bopp et al., 2013). Assuming that none of the CMIP5 models are currently able to simulate the PSR feedback, our results therefore suggest that accounting for the feedback would alter the CMIP5 range of projections from a 7-18% to a 6-15.5% decline in export, and the CMIP5 ensemble mean projection from a 13% to an 11% decline in export. Many of the CMIP5 models may be capable of capturing some semblance of a PSR feedback effect, however, thus necessitating smaller corrections.

All CMIP5 models simulate various processes that have the potential to change POC export fluxes in a warming future ocean, including zooplankton grazing/fecal pellet formation, phytoplankton aggregation, phytoplankton/zooplankton mortality, and variations in phytoplankton community structure (which provide the source material for sinking particles) based on changing nutrient, temperature, and light conditions. However, 7 of 17 total CMIP5 models with ocean biogeochemistry simulate only one class or type of particulate organic matter (Ilyina et al., 2013; Tjiputra et al., 2013; Tsujino et al., 2010; Watanabe et al., 2011; Zahariev et al., 2008), and therefore cannot capture changes in the nature of sinking POC with future warming. The other 10 CMIP5 models can simulate changes in the nature of sinking particulate organic matter with future warming, either through the amount of associated ballasting (3 models) (Dunne et al., 2013; Moore et al., 2004) or through particle size (7 models).

Of the 7 models that simulate changes in particle size and could thus potentially capture the PSR feedback effect, 3 of these models resolve two particle sizes (small and large) with different sinking speeds (3 m day-1 for the small particles and 50 to 200 m day-1 for the large particles) (Aumont and Bopp, 2006). Another 2 of these 7 models also resolve two size-based particle types (diatoms and detritus) with different sinking speeds (1 m day-1 for diatoms and 10 m day-1 for detritus) (Totterdell, 2019). The final 2 of these 7 models simulate 5 different organic particle sinking speeds based on size, 1 for each different phytoplankton type in the model (for a total of 4) and 1 for carbon detritus (Romanou et al., 2013). In sum, out of the 17 CMIP5 models described here, only 7 resolve particles of more than one size that sink down the water column at different speeds and therefore have the potential to capture some part of the PSR feedback effect.

If all the CMIP5 models differed only in their resolution of sinking particle sizes, then we would expect the 7 models with more than one particle size to project the smallest decreases in export with future warming. In reality, however, the models differ in too many other ways to isolate potential impacts of the PSR feedback when comparing between them. For example, three of the models that resolve more than one sinking particle size (IPSL-CM5A-LR, IPSL-CM5A-MR, and HadGEM2-ES) predict some of the largest decreases in export production by 2100 (Fig. 9b in Bopp et al., 2013), contrary to what would be expected given the potential presence of a PSR feedback in these models. Indeed, the models that can simulate changes in the nature of sinking particles project changes in export that span the entire range of CMIP5 model predictions (Fig. 9b in Bopp et al., 2013). The reasons for these differences in projected export decreases is difficult to disentangle and would require examining the effects of one mechanism at a time on export in each model.

Simply representing differently sized particles also does not ensure that a model will adequately represent the negative PSR feedback quantified in this study. To adequately represent the negative PSR feedback, a given model would need to contain mechanisms that give rise to the same strong, empirically-derived relationships between POC export and particle size that we specify here. Within the models that resolve particle size to some degree, the relative proportion of large and small particles is determined by internal model dynamics and are not prescribed empirically. Furthermore, these models do not resolve a particle size spectrum over a wide range of sizes as is done here. Therefore, CMIP5 models that dynamically resolve 2-5 different particle size classes with different sinking speeds might qualitatively reproduce the same feedback, but it is not clear whether the magnitude or even the sign of the feedback would be accurately captured. We argue that our model, which resolves particle size spectra over a wide range of sizes and employs empirical export/particle-size relationship constraints, is most likely to accurately capture the true magnitude of the PSR feedback. We thus suggest that our study provides a reasonable baseline against which more complex Earth System Models can assess their ability to reproduce particle size-remineralization feedbacks."

*- I think some aspects of the model should be described in more detail, in particular the PRiSM component used to estimate particle flux (see line-by-line comments)*

We thank the reviewer for pointing out that the model description falls short. The mathematical description of the PRiSM model has been fully laid out in previous references and cannot be repeated in full here. However, we agree that enough information needs to be provided to allow the reader to understand how the model "works", without extensive cross-referencing to previous papers. We added the following text to Section 2.1.1: Model setup:

"2.1.1 Model setup

[revised manuscript text omitted]

*Line-by-line comments:*

*L 9: Please write something like "This decline is mainly caused by. . .", as there are other mechanisms projected as well, for instances stronger grazing pressure due to higher temperatures*

We changed the statement from:

"This decline is caused by increased stratification..."

To:

"This decline **is mainly** caused by increased stratification..."

*L 11: But there are some Earth System Models that simulate changes in remineralization depth, due to e.g. changes in phytoplankton community composition? For instance CESM-BEC, IPSL-PISCES and GFDL models all simulate changes in formation and sinking of particles under climate change. It is true that most models don't have a dynamic particle size spectrum though. (I am also wondering, do the Earth System Models that already have particles with different size classes also predict lower export decrease? But this is probably impossible to disentangle from other effects such as a temperature-dependent remineralization or varying stoichiometry etc. )*

Good point. We added Section 3.5: Comparison with CMIP5 models to better discuss this, quoted above.

*L 45/46 Maybe cite Laufkotter et al. 2015 for drivers of future declines in primary production as well*

An important reference! We added this citation.

*Introduction: I miss a discussion or mentioning of other drivers of sinking speed/particle remineralization, such as particle density/porosity and ballasting, also fragmentation and aggregation which potentially change the particle size over time*

We added the following to the introduction:

"This raises the possibility of feedback loops in which changes in particle remineralization depth might either dampen (negative feedback) or enhance (positive feedback) circulation-driven decreases in primary production and export. For instance, increasing ocean temperatures may speed up bacterial remineralization rates (Cavan et al., 2019; Cram et al., 2018; John et al., 2014; Laufkötter et al., 2017; Marsay et al., 2015; Matsumoto, 2007) and enhance recycling of nutrients near the surface, which would dampen physically-driven decreases in surface nutrient concentrations and result in a negative feedback on export. Oxygen concentrations, on the other hand, are predicted to decrease with future warming (Bopp et al., 2002; Cabré et al., 2015b; Keeling et al., 2010; Long et al., 2016; Matear & Hirst, 2003; Schmidtko et al., 2017) and slow bacterial remineralization and zooplankton-mediated particle disaggregation rates (Cavan et al., 2017; Devol & Hartnett, 2001; Hartnett & Devol, 2003; Laufkötter et al., 2017; Van Mooy et al., 2002); this would result in deeper particle remineralization and further exacerbate circulation-driven nutrient supply decreases, leading to a positive feedback on export production. A decrease in mineral ballasting and protection of particles with ocean acidification may also feedback negatively on export decreases by shoaling remineralization depths (Hofmann and Schellnhuber, 2009). Future changes in sinking particle size may also lead to strong feedbacks on export."

We also added the following two segments to Section 3.6: Caveats and future work:

"Furthermore, particle fragmentation—via processes such as zooplankton grazing, microbial degradation, or ocean turbulence (e.g., Cavan et al., 2017; Briggs et al., 2020)—was not included in our model, nor was particle aggregation—via processes such as Transparent Exopolymer Particles production (e.g., Passow, 2002; Mari et al., 2017) or fecal pellet generation (e.g., Steinberg et al., 2012; Turner, 2015 and references therein)."

"Other remineralization depth-related feedbacks induced by changes in temperature, oxygen, particle density, and mineral ballasting (among others) not studied here may also be important for modulation of future changes in carbon export and its downstream effects. Ensuring that the PSR and other remineralization feedbacks are adequately represented in ESMs should be a priority of the modeling community to enable robust predictions of carbon export fluxes in the future ocean."

*L 67/68 "parameters and processes in most previous models are not constrained by observations of particle size distributions" - Yes! Nice!*

Thanks!

*L74/75 Are there more recent references for the particle size distribution?*

Yes. We added citations to the following references:

Cael, B.B. and White, A.E., 2020. Sinking versus suspended particle size distributions in the North Pacific Subtropical Gyre. Geophysical Research Letters, p.e2020GL087825.

White, A.E., Letelier, R.M., Whitmire, A.L., Barone, B., Bidigare, R.R., Church, M.J. and Karl, D.M., 2015. Phenology of particle size distributions and primary productivity in the North Pacific subtropical gyre (Station ALOHA). Journal of Geophysical Research: Oceans, 120(11), pp.7381-7399.

Buonassissi, C.J. and Dierssen, H.M., 2010. A regional comparison of particle size distributions and the power law approximation in oceanic and estuarine surface waters. Journal of Geophysical Research: Oceans, 115(C10).

*L 99 Just a small comment: I think the "shallowest" can be removed?*

We changed the statement from:

"Here we define remineralization depth as the shallowest depth at which POC flux…"

To:

"Here we define remineralization depth as **the depth** at which POC flux…"

*L 117 I think the decreased nutrient supply causes the decrease in phytoplankton/particle size, not the decrease in export?*

In our model, everything is linked by the empirical function/relationship between phytoplankton/particle size and export. Thus, the causal chain in our model is that the reduced nutrient supply results in reduced export, which (empirically) drives a shift towards smaller particles. In reality, this shift to smaller particles is likely driven by increased dominance of smaller phytoplankton, but this is not explicitly represented in our model, only the export/particle-size relationship is.

To clarify this, we added the following at the beginning of Section 2.1.2: Model representation of the PSR feedback:

"When the PSR feedback is disabled within our model, circulation-driven changes in the nutrient supply to the euphotic zone (see Section 2.3) will lead to changes in POC export, but β (and therefore particle remineralization depths) remains constant over time. With the PSR feedback enabled, any change in POC export is accompanied by a change in β, the direction and magnitude of which is specified using the empirical relationships discussed in Section 2.2. We note that by design, this modeling approach makes no assumptions about the mechanisms driving shifts in the particle size distribution, only that β changes in tandem with POC export, in a manner that is consistent with observations."

*The introduction reads slightly redundant to me and could be streamlined a bit, particularly part 1.2 (PSR feedback). However, I acknowledge that other reviewers seem to particularly like it and find it "very clear", so maybe for readers with little background in particle export it is better the way it is.*

We moved this text out of the introduction and into the discussion section, which is now called Section 3.2: An empirical negative global particle size-remineralization (PSR) feedback. We shortened it a bit as well.

*L 164 Does the plankton not become small plankton, or does the small plankton not produce small particles? Oh I see, plankton community isn't explicitly represented, correct?*

Correct, the phytoplankton community isn't explicitly resolved - our model simulates phytoplankton growth (NPP) and organic matter export as a function of temperature, light and nutrients. The size spectrum of sinking particles (beta) is then computed based on the empirically derived export/particle-size relationships.

*I don't understand where Prism has the microbial respiration rates from, and whether they increase due to temperature in a warmer ocean? Is there an oxygen dependence? Is there aggregation/fragmentation? Can the sinking speed change over depth? Given that these questions are essential to this study, please explain Prism in more detail.*

We added these pieces of information to Section 2.1.1: Model setup, quoted above.

*L173/174 wouldn't it make sense to also increase the production of DOC when changing the beta of the particle size spectrum?*

In general, the cutoff between DOC and POC is arbitrary (i.e., based on filter sizes). Perhaps there should be more DOC production when particles are smaller (because there would be a relatively greater number of very small particles beneath the minimum filtration size), but we do not have sufficient information that we can use to parameterize this in an empirical way.

*Fig 3: That's probably only my personal preference, but I generally prefer mapping data using a log scale, instead of plotting log(data) and using a linear scale.*

We kept the linear scale, but relabeled it with 10^x for clarity.

*L 233/234 The NPP estimates you are using are significantly biased in the Southern Ocean. This should probably be discussed here, or you could use an NPP estimate that has been specifically created for the Southern Ocean (for instance Johnsson et al. 2013).*

We thank the reviewer for pointing this out. At this stage, we cannot repeat our entire suite of analyses using a new NPP estimate, but we added a note about this limitation at Line 301 as follows: "The unique relationship between β and export in the SAZ is worth further exploration, and may be further elucidated by NPP datasets that are specifically calibrated for the Southern Ocean (e.g. Johnson et al., 2013), but is beyond the scope of the current study."

*L254 How sensitive are your results to this simplified representation of future ocean circulation?*

This is a great question, and not one that we can definitively answer. Because our model represents circulation using a mass-conserving transport matrix, we can only uniformly scale circulation rates up or down, rather than manipulating the patterns of circulation in more realistic ways. This limitation is already noted in our paper.

However, it is important to point to note that although we use a simplified representation of future changes in ocean circulation, the exact same simplified representation is used in both the PSR feedback-on and feedback-off cases. We are therefore isolating the effects of the particle size feedback from the effects of the circulation change itself. It is therefore not clear that our results would be significantly different if we used a model with a more complex representation of future circulation changes, as long as that model also applied the exact same circulation changes in feedback-on and feedback-off scenarios.

We added an explanation of this at Line 255 as follows:

"Although we use a simplified representation of future changes in ocean circulation, the exact same simplified representation is implemented in both PSR feedback-on and -off simulations. We are thus isolating the effects of the PSR feedback from the effects of the circulation change. It is therefore not unreasonable to assume that our calculated PSR feedback strength would be comparable to that computed from a physical model with a more complex representation of future circulation changes, as long as that model also applied identical circulation changes in PSR feedback-on and -off scenarios."

*L 282 In light of these "counterintuitive" relationships I think it would be really interesting to use an NPP algorithm that's been developed for the Southern Ocean, as mentioned above (the result might stay the same of course, and I like how it is discussed/explained with the Lam&Bishop findings)*

Good point - we mention this in the text as a suggested direction for future work at Line 301 as follows: "The unique relationship between β and export in the SAZ is worth further exploration, and may be further elucidated by NPP datasets that are specifically calibrated for the Southern Ocean (e.g. Johnson et al., 2013), but is beyond the scope of the current study."

*L 346: Is this a conclusion from the correlation between P200m and E or something that is actively diagnosed in the model?*

We answer this question and the following L355 question together below.

*L 355 ff The considerations up to this point hold for the nutrient-limited low and mid latitudes. But here you discuss a global uniform decrease of circulation rates. I found the jump from regional to global a bit confusing. Wait, I see now that in line 349, eq 3 is meant to hold at any given location, including regions in which eq. 2 does not hold, yes? Is that mathematically sound??*

This answer is in response to the above 2 comments.

Equation 2 is intended to provide a simplified interpretation of results in terms of changes in subsurface nutrient concentrations. It reflects the assumption that at steady-state, the export flux out of the euphotic zone must approximately balance the supply of nutrients into the euphotic zone by upwelling. Of course, because this is a simplification of reality, the relationship will never be 100% accurate, if this is what the reviewer means by "mathematically sound." However, it is a reasonable assumption for much of the ocean and is widely used in the biogeochemical literature (e.g. Ducklow et al., 2001; Passow and Carlson, 2012). We note that equations 3-4 are directly derived from Equation 2 via perturbation analysis. They therefore hold precisely to the same degree that Equation 2 holds.

In the revised manuscript, we note more clearly that Equations 2-4 are only intended to help simplify our interpretations, and are not mathematically equivalent to the full model solution. However, Fig 7 demonstrates that Eqs 2-4 do explain much of the full model behavior, and in doing so allow us to deconvolve and better understand the different mechanisms leading to export changes. Where these equations hold, they allow us to demonstrate how the PSR feedback operates to buffer export under slower circulation (i.e., by increasing nutrient concentrations in the shallow subsurface). We clarified wording in the text to explain this as follows:

Line 414:
"This relationship [referring to Eq. (2)] between export, upwelling, and subsurface nutrient concentrations reflects the common assumption that at steady-state, export flux out of the euphotic zone must approximately balance the supply of nutrients into the euphotic zone by upwelling (e.g. Ducklow et al., 2001; Passow and Carlson, 2012). This balance can in turn be used to derive (via perturbation analysis) a simple, approximate diagnostic [referring to Eq. (3)] for understanding changes in export under altered circulation rates at any given location."

Line 423:
"Though Eqs. (2-3) are not mathematically equivalent to the full model solution, they explain much of the full model's behavior and provide us with a tool to simplify, deconvolve, and better understand the different mechanisms leading to export changes."

Line 453:
"To understand this spatial pattern, we combine Eq. (3) with our definition of PSR feedback strength to yield the following diagnostic, which can help separate out the various determinants of PSR feedback strength: ..."

Line 471:
"Our simple diagnostic (Eq. (4), derived from Eqs. (2-3)) can explain PSR feedback strengths quite well over much of the global ocean, as can be seen by comparing total feedback strengths (blue lines/bars in Fig. 7g,h) with our diagnostic-derived feedback strengths (right-hand side of Eq. (4), represented by orange lines/bars in Fig. 7g,h), which were calculated under the assumption that export is nearly equal to the supply of nutrients into the euphotic zone via upwelling (Eq. (2)). In other words, Eqs. (2-4) are a good approximation to the full model solution where the orange lines/bars lie relatively close to the blue lines/bars in Fig. 7g-h. However, new production can be fed by local upwelling as well as lateral advection, such that changes in $P200m$ and vertical exchange rates alone (orange lines/bars in Fig. 7g,h) cannot perfectly predict all changes in export (blue lines/bars in Fig. 7g,h), especially in regions where lateral advection plays a relatively large role in supplying nutrients to the surface."

Circulation rates are indeed decreased uniformly across the global ocean ($\Delta w/w\_baseline$ = -10% in all regions in Equation 4), but the strength of the PSR feedback varies regionally depending on how much P200 changes in that region between the feedback on and off runs (see Equation 4).

*L 358 "This decrease in P200m is .." - Is this your interpretation or has this been diagnosed in the model?*

This is our interpretation, backed up by an understanding of how Southern Ocean nutrient utilization drives low latitude nutrient supply from other model studies (e.g., Sarmiento et al., 2004; Marinov et al., 2006).

We changed the wording from:

"This decrease in P200m is largely driven by enhanced biological nutrient utilization…"

To:

"This decrease in P200m **is likely** largely driven by enhanced biological nutrient utilization…"

And added references to Sarmiento et al. (2004) and Marinov et al. (2006).

*L 404 - Is eq. 2 a close approximation in the low latitudes or globally? Is it possible to show on a map how well Eq 2 holds regionally?*

Fig. 7g-h show how well this approximation holds both zonally and regionally. The orange lines/bars show feedback strength as approximated from the right hand side of Equation 4, which is directly derived from Equation 2. The blue lines/bars show actual feedback strength calculated from changes in export (left hand side of Equation 4). The closer the orange lines/bars are to the blue lines/bars, the better the approximation in Equation 2.

We added explanation of this in the text at Line 471 as follows:

"Our simple diagnostic (Eq. (4), derived from Eqs. (2-3)) can explain PSR feedback strengths quite well over much of the global ocean, as can be seen by comparing total feedback strengths (blue lines/bars in Fig. 7g,h) with our diagnostic-derived feedback strengths (right-hand side of Eq. (4), represented by orange lines/bars in Fig. 7g,h), which were calculated under the assumption that export is nearly equal to the supply of nutrients into the euphotic zone via upwelling (Eq. (2)). In other words, Eqs. (2-4) are a good approximation to the full model solution where the orange lines/bars lie relatively close to the blue lines/bars in Fig. 7g-h. However, new production can be fed by local upwelling as well as lateral advection, such that changes in P200m and vertical exchange rates alone (orange lines/bars in Fig. 7g,h) cannot perfectly predict all changes in export (blue lines/bars in Fig. 7g,h), especially in regions where lateral advection plays a relatively large role in supplying nutrients to the surface."

*L 484 is this a typo "1.16 times"? Otherwise, I don't understand, why 1.16 times? 1 - 16%? But I wouldn't understand that, either. Sorry if I am missing something obvious.*

We changed the sentence from:

"This implies that global models without the PSR feedback may be overestimating 100-year climate-driven export decreases by ~1.16 times."

To:

"This implies that **in isolation of other mechanisms, ESMs** without the PSR feedback may be **projecting 100-year climate-driven export decreases that are ~1.16 times too large.**"

*L 485 I believe the Pisces models used in the Bopp et al. study implements two different particle sizes, so it's possible that the PSR feedback is at least partly included? Please check*

Yes, this is true. We clarified this in our new Section 3.5: Comparison with CMIP5 models, quoted above.

*L 507/508 "PSR feedback strength remains relatively constant whether circulation rates are increased/decreased by 10% or 50% " - That is very surprising to me. Why would that be? Are you reaching a maximum/minimum particle size distribution such that a further decrease in nutrient supply does not affect the size distribution anymore? Please discuss this a bit more. I also think this sensitivity test should be mentioned in the results already.*

Remember that the PSR feedback strength is calculated as the relative difference in projected export decrease between the feedback on and off cases, which are both run with exactly the same circulation rate change. It is this PSR feedback strength, NOT the amount of PSD slope ($\beta$) change, that stays the same when you amplify circulation rate changes. Thus, even though $\beta$ changes more in the 50% decreased circulation case when the feedback is on, the PSR feedback strength is relatively unchanged from the 10% decreased circulation case. In other words, the percentage difference in projected export change between the PSR feedback on and off cases is constant even as circulation rates slow more. To be sure, however, absolute export decreases are indeed greater in both the feedback on and off cases under the 50% reduced circulation rates. We will added this explanation into the text at Line 400 as follows:

"The strength of the PSR feedback also does not depend on the size of circulation rate changes. Indeed, we observed that PSR feedback strength remains constant whether circulation rates are increased/decreased by 10% or 50%. Thus, the percentage difference in projected export change between PSR feedback on and off cases is relatively uniform even under quite different changes in circulation rates."

*L 528 I think the study by Briggs et al. 2020 is good recent reference for fragmentation*

Good point. We added this reference.

*L 545 The effect of temperature increases and oxygen decreases on future carbon export has been analyzed in Laufkotter et al 2017, maybe good to cite here*

Good point. We added this reference.

Conclusion: If I understand everything correctly, your results suggest that changes in particle size/sinking speed in the Southern Ocean have only a small impact on carbon export regionally and globally. This could be interpreted as a justification for modelers to give implementation of particle size a low priority in this region - would you agree?

Very interesting point! I suppose this would be true if modelers only really cared about better resolving the effects of this particular feedback in the Southern Ocean. However, correctly modeling particle size in this region may be important for accurate estimates of absolute export. Indeed some of our earlier work shows that large particles in cold temperatures in the Southern Ocean contribute to particularly high carbon sequestration in this region (Cram et al., 2018). Thus, I think modelers/observationalists should keep working on improving particle size representations in the Southern Ocean.

*Anyway. I hope this review helps, and my apologies that it took me so long!*

*Laufkötter, C., Vogt, M., Gruber, N., Aita-Noguchi, M., Aumont, O., Bopp, L., Buitenhuis, E., Doney, S. C., Dunne, J., Hashioka, T., Hauck, J., Hirata, T., John, J., Le Quéré, C., Lima, I. D., Nakano, H., Seferian, R., Totterdell, I., Vichi, M., and Völker, C.: Drivers and uncertainties of future global marine primary production in marine ecosystem models, Biogeosciences, 12, 6955–6984, https://doi.org/10.5194/bg-12-6955-2015, 2015.*

*Briggs et al., Science 367, 791–793 (2020), Major role of particle fragmentation in regulating biological sequestration of CO2 by the oceans*

*Johnson, R., P. G. Strutton, S. W. Wright, A. McMinn, and K. M. Meiners (2013), Three improved Satellite Chlorophyll algorithms for the Southern Ocean, J. Geophys. Res. Oceans, 118, 3694–3703*

[revised manuscript text omitted]

**(a)** Baseline zonal mean PO₄

[Figure]

**(b)** 100-yr PO₄ change WITHOUT PSR feedback

[Figure]

**(c)** 100-yr PO₄ change WITH PSR feedback

[Figure]

**(d)** PO₄ WITH feedback − WITHOUT feedback

[Figure]

Figure 9. (a) Relative changes in zonal mean particulate organic carbon (POC) flux at 800 m depth (725 m below the bottom of the euphotic zone) 100 years after decreasing circulation rates by 10%. (b) Same as (a), but with regional and global means. (c) Map of regional mean depths below the surface at which the PSR feedback effect flips from negative to positive. In the North Atlantic, the regional mean PSR feedback effect flips from negative to positive at ~2100 m depth, which is off the color scale.

[Figure]

[Figure]

**Figure 10. (a)** Regional mean feedback strength due to the PSR negative feedback effect within each individual region. The y-axis denotes the single region (or the entire ocean in the case of "GLB" or "global") within which the PSR feedback was turned on, while the x-axis denotes the region affected. A feedback strength above 0 indicates a negative PSR feedback effect (that is, a reduction in export change when the PSR feedback is applied). **(b)** Percent contribution of each individual region to each region's total PSR feedback strength, computed as the regionally-derived feedback strength within an affected region divided by the globally-derived feedback strength in the same affected region (i.e., each given grid cell in (a) is divided by the corresponding column's bottom-most grid cell).

**Supplementary Information**

[Figure]

**Figure S1: PRiSM-calculated particle flux profiles with varying surface $\beta$ values. Larger $\beta$ values correspond with smaller particles.**

[Figure]

**Figure S2: Annual means of all nine monthly time series of global export considered here, computed from all possible permutations of three net primary productivity (NPP) and three e-ratio (export/NPP) algorithms (described in Section 2.2.2). Units are molC m⁻² yr⁻¹.** NPP algorithm key: VGPM = the Vertically Generalized Production Model (VGPM) (Behrenfeld & Falkowski, 1997); VGPME = the Eppley-VGPM model (Carr et al., 2006); CbPM = the Carbon-based Production Model (Behrenfeld et al., 2005). E-ratio algorithm key: L2000 = Laws et al. (2000); D2005 = Dunne et al. (2005); L2011 = Laws et al. (2011).

[Figure]

**Figure S3: All nine monthly $\beta$ versus time-mean normalized export ($\frac{d\beta_{sat}}{dE_{n,sat}}$, unitless) maps considered here. Title colors correspond to the NPP and e-ratio export combinations in Fig. 4a.**

[Figure]

[Figure]

Figure S4: Example $\beta$ and time-mean normalized export ($E_n$) time series at the randomly chosen grid points within each ocean region shown in the map. The first letter of each ocean region name denotes the location of the chosen grid point. All $E_n$ time series shown here are derived from combining VGPM NPP and D2005 e-ratio. $\frac{d\beta_{sat}}{dE_{n,sat}}$ values are calculated over the entire SeaWiFS period (September 1997 – December 2010), but we show only a random subset of these years for visual clarity.

[Figure]

**Figure S5: Example β and time-mean normalized export ($E_n$) time series at the randomly chosen grid point within the SAZ (Subantarctic Zone) denoted in the map in Fig. S4. The NPP and e-ratio algorithms used to derive the $E_n$ time series are denoted above each subplot. $\frac{d\beta_{sat}}{dE_{n,sat}}$ values are calculated over the entire SeaWiFS period (September 1997 – December 2010), but we show only a random subset of these years for visual clarity.**

| Particle parameters | Definition | Units | Value |
|---|---|---|---|
| $D_L(z' = 0)$ | Largest particle diameter at surface | um | 2000 |
| $D_S(z' = 0)$ | Smallest particle diameter at surface | um | 20 |
| $c_w$ | Coefficient in the relationship between particle sinking velocity and particle size | $m^{(1-\eta)}\,day^{-1}$ | 2.2e5 |
| $\eta$ | Exponent in the relationship between particle sinking velocity and particle size | Unitless | 1.17 |
| $c_r$ | Degradation rate of sinking particles | $day^{-1}$ | 1/29 |
| $\zeta$ | Exponent in the relationship between particle mass and particle size | Unitless | 1.62 |

| Biogeochemical parameters | Definition | Units | Value |
|---|---|---|---|
| $\tau$ | Nutrient restoring timescale | days | 30 |
| $\kappa$ | DOP to $PO_4$ first-order decay rate | $year^{-1}$ | 0.5 |
| $\sigma$ | Fraction of production routed directly to DOP in the euphotic zone | Unitless | 0.1 |
| $z_s$ | Nominal mixing depth | m | 115 |

**Table S1: PRiSM parameter values (reproduced from Table 1 in DeVries et al., 2014 – see DeVries et al., 2014 for the equations in which the parameters are used)**

| Parameter | Definition | Units | Value |
|---|---|---|---|
| $T_o$ | Reference temperature | °C | 25 |
| $\mu_{max}$ | Maximum growth rate at reference temperature | $year^{-1}$ | 365.25 |
| $K_p$ | Half-saturation coefficient for $PO_4$ uptake | $mmol\ m^{-3}$ | 0.1 |
| $K_I$ | Saturating light level | $W\ m^{-2}$ | 40 |
| $k_T$ | Temperature sensitivity of growth | Unitless | 0.03 |
| $m_1$ | Linear mortality rate | $year^{-1}$ | 36.525 |
| $m_2$ | Quadratic mortality rate | $year^{-1}\,mmol^{-1}\,m^3$ | 3652.5 |

**Table S2: Prognostic production scheme parameter values, with minor differences from those used in Weber and Deutsch (2012). These parameter values were re-derived by matching model surface $PO_4$ values with World Ocean Atlas observations on a 2-degree horizontal grid, in contrast with the 4-degree grid used in Weber and Deutsch (2012).**

| Export algorithms | AAZ region | SAZ region | STA region | STP region | ETA region | ETP region | NA region | NP region |
|---|---|---|---|---|---|---|---|---|
| **VGPM NPP** **+ e-ratio from:** | | | | | | | | |
| Laws 2000 | 0.1139 | 0.3207 | 0.2308 | 0.0504 | 0.0656 | 0.0656 | 0.0478 | 0.0000 |
| Dunne 2005 | 0.1508 | 0.2328 | 0.1677 | 0.0300 | 0.0729 | 0.0729 | 0.0697 | 0.0026 |
| Laws 2011 | 0.0927 | 0.0454 | 0.0975 | 0.0208 | 0.0445 | 0.0445 | 0.1169 | 0.1855 |
| **VGPM-Eppley NPP** **+ e-ratio from:** | | | | | | | | |
| Laws 2000 | 0.1507 | 0.0420 | 0.1419 | 0.0663 | 0.1213 | 0.1213 | 0.1184 | 0.1197 |
| Dunne 2005 | 0.1349 | 0.0212 | 0.0993 | 0.0435 | 0.1516 | 0.1516 | 0.1294 | 0.2379 |
| Laws 2011 | 0.0622 | 0.0036 | 0.0636 | 0.0292 | 0.1080 | 0.1080 | 0.1308 | 0.1211 |
| **CbPM NPP** **+ e-ratio from:** | | | | | | | | |
| Laws 2000 | 0.0478 | 0.2014 | 0.0900 | 0.2688 | 0.1667 | 0.1667 | 0.1263 | 0.0107 |
| Dunne 2005 | 0.1215 | 0.1141 | 0.0640 | 0.2695 | 0.1047 | 0.1047 | 0.1322 | 0.0978 |
| Laws 2011 | 0.1255 | 0.0188 | 0.0451 | 0.2216 | 0.1648 | 0.1648 | 0.1286 | 0.2247 |
| **Sum** | 1.00 | 1.00 | 1.00 | 1.00 | 1.00 | 1.00 | 1.00 | 1.00 |

**Table S3: Regional weights for export map calculation (reproduced from Table S2 in Weber et al., 2016)**